# Parallel Tempering With a Variational Reference

**Nikola Surjanovic**
Department of Statistics
University of British Columbia
nikola.surjanovic@stat.ubc.ca

**Saifuddin Syed**
Department of Statistics
University of Oxford
saifuddin.syed@stats.ox.ac.uk

**Alexandre Bouchard-Côté**
Department of Statistics
University of British Columbia
bouchard@stat.ubc.ca

**Trevor Campbell**
Department of Statistics
University of British Columbia
trevor@stat.ubc.ca

## Abstract

Sampling from complex target distributions is a challenging task fundamental to Bayesian inference. Parallel tempering (PT) addresses this problem by constructing a Markov chain on the expanded state space of a sequence of distributions interpolating between the posterior distribution and a fixed reference distribution, which is typically chosen to be the prior. However, in the typical case where the prior and posterior are nearly mutually singular, PT methods are computationally prohibitive. In this work we address this challenge by constructing a generalized annealing path connecting the posterior to an adaptively tuned variational reference. The reference distribution is tuned to minimize the forward (inclusive) KL divergence to the posterior distribution using a simple, gradient-free moment-matching procedure. We show that our adaptive procedure converges to the forward KL minimizer, and that the forward KL divergence serves as a good proxy to a previously developed measure of PT performance. We also show that in the large-data limit in typical Bayesian models, the proposed method improves in performance, while traditional PT deteriorates arbitrarily. Finally, we introduce PT with two references—one fixed, one variational—with a novel split annealing path that ensures stable variational reference adaptation. The paper concludes with experiments that demonstrate the large empirical gains achieved by our method in a wide range of realistic Bayesian inference scenarios.

## 1 Introduction

Parallel tempering (PT) is a popular approach to sampling from challenging probability distributions used in many scientific disciplines [1, 2, 10, 5]. PT methods involve running Markov chain Monte Carlo (MCMC) on the expanded state space of a sequence of distributions that connect the target distribution of interest, $\pi_1$, to a simple reference distribution, $\pi_0$, for which i.i.d. sampling is tractable. The key innovation in PT is that the MCMC chain enables distributions along the path to swap states (or *communicate*). This communication enables i.i.d. draws from the reference $\pi_0$ to aid in exploration of the challenging target $\pi_1$. Indeed, recent work has shown that the effectiveness of a PT method is essentially characterized by the efficiency of this communication via the *global communication barrier* (GCB) from $\pi_0$ to $\pi_1$ [36]. Intuitively, the GCB is low when the reference $\pi_0$ is similar to the target $\pi_1$; in this case, the distributions along the path have substantial overlap and proposed swaps are generally accepted. The GCB is also inversely related to the *restart rate*, which quantifies how frequently i.i.d. samples from $\pi_0$ traverse the path to $\pi_1$ (a *restart*) [36].

36th Conference on Neural Information Processing Systems (NeurIPS 2022).

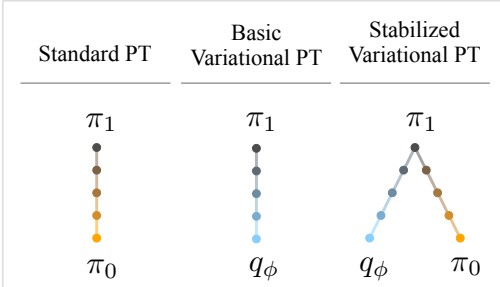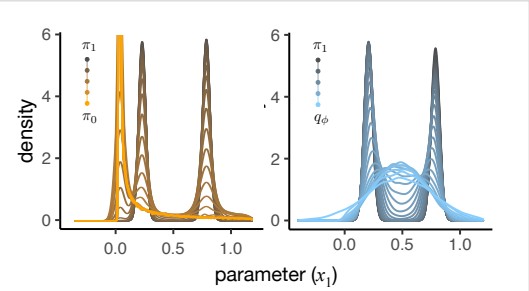

Figure 1: **Left box:** visualization of the three main PT algorithms considered in this work. Nodes represent distributions interpolating between tractable reference distributions (bottom, either fixed $\pi_0$ or variational $q_\phi$), and an intractable distribution (top, $\pi_1$, typically a Bayesian posterior). Edges encode the structure of the possible swaps performed by the various PT algorithms. **Right box:** examples of a path of marginal distributions obtained on a Bayesian ODE parameter estimation problem with more than one latent variable for an mRNA transfection dataset [5, page 8]. The two modes are non-trivial to switch as they require changing other parameters (not shown) simultaneously. In this example, the marginal of the prior places a small amount of mass on the second mode whereas the marginal of the variational reference places significant mass on both modes. Because the variational reference covers both modes, the length of the annealing path from the reference to the target is shorter and it is easier to obtain samples from the target distribution using parallel tempering.

In the setting of Bayesian posterior inference—a key application of PT, and the focus of this work—the target $\pi_1$ is the posterior distribution, and the reference distribution $\pi_0$ is typically set to the prior. From the perspective of PT communication efficiency, this is a poor choice in general; the prior is often quite different from the posterior, resulting in a high GCB. As an extreme (but common) example, we show in this work that when the posterior distribution concentrates in the large-data limit, the restart rate with a fixed reference tends to zero and PT becomes computationally infeasible (Proposition 3.1). On the other hand, the posterior often exhibits regularities—asymptotic normality in certain parameters, for example—that motivate the need for a choice of PT reference that can automatically adapt to the target to obtain computational gains.

In this work, we develop and analyze a novel PT algorithm that automatically adapts a variational reference distribution within a parametric family, $\mathcal{Q} = \{q_\phi : \phi \in \Phi\}$. This adaptive reference family addresses the shortcomings of using the prior as a PT reference: we show that in the large-data limit, the restart rate with an appropriate variational reference improves arbitrarily (Proposition 3.2). We find that even when one is not in the large data setting, our method can provide large empirical gains compared to fixed-reference PT in a wide range of realistic Bayesian inference scenarios. The method is based on two major methodological contributions. First, we adapt the parameter $\phi$ to minimize the forward (inclusive) KL divergence $\mathrm{KL}(\pi_1 \| q_\phi)$ instead of directly taking gradients with respect to the GCB itself. This approach is particularly advantageous when $\mathcal{Q}$ is an exponential family: Theorem 3.5 shows that the forward KL is a good surrogate of the GCB, and minimizing the forward KL amounts to matching moments, which involves no extra tuning effort from the user. We perform moment matching in a simple iterative fashion, in rounds of increasingly many PT draws; Theorem 3.4 identifies conditions that guarantee that the variational parameter estimate converges to the optimum. Second, we combine two references—one fixed, one variational—by "gluing" two PT algorithms together (each based on one of the references, see Fig. 1). We demonstrate that this "stabilized" method is necessary for obtaining a reliable PT algorithm: adaptation with just the variational reference alone can lead to "forgetting" the structure of the posterior distribution (e.g., multi-modality, as shown in Fig. 2). Although this requires more computational effort, we show that under idealized conditions the restart rate of our adaptive method is no lower than half the restart rate of standard, fixed reference PT (Theorem 3.6) after accounting for the doubled computation time. In practice, it is often much better. Finally, the paper presents an extensive empirical study of the performance of our method in a variety of real-world Bayesian models, including spatial models (sparse random field Poisson regression) and functional data analysis (Bayesian estimation of ODE parameters), among others. We find that our method can substantially increase the performance of PT.

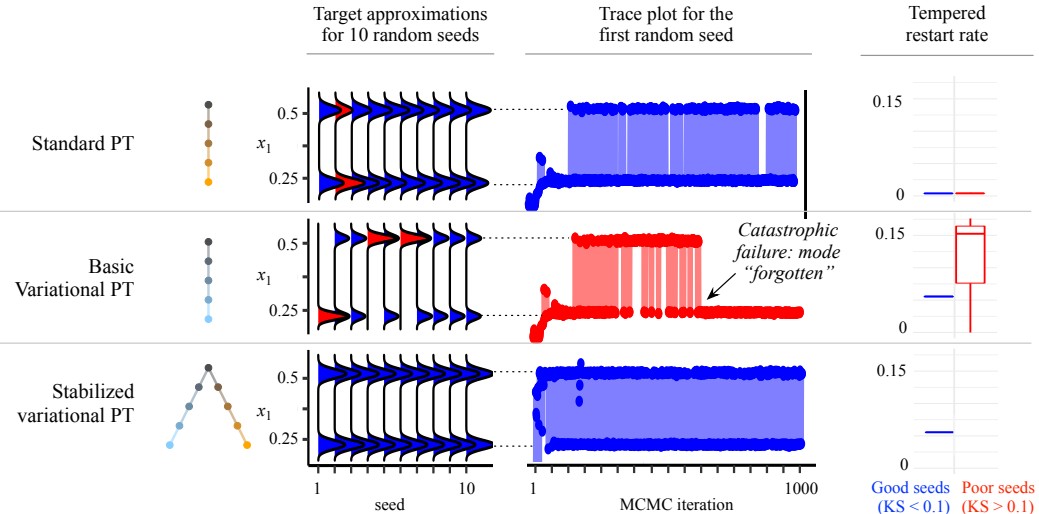

Figure 2: Comparison of three PT methods for Bayesian ODE parameter estimation with an mRNA transfection dataset [5, page 8]. **Top row:** standard PT succeeds in achieving a positive but small restart rate (a measure of PT efficiency; higher is better). **Middle row:** basic implementation of only one variational reference distribution leads to catastrophic failures in 3 out of 10 independent runs. The variational distribution sometimes "forgets" one of the modes (red denotes "bad" approximations with a Kolmogorov-Smirnov distance for the marginal posterior over the model parameter $x_1$ larger than 0.1) and becomes stuck in the other, which leads to an overestimation of the restart rate. **Bottom row:** using both a fixed and an adaptive reference addresses this issue and leads to markedly better performance in terms of restart rate compared to standard PT. All methods have a comparable cost per iteration; all use the same variational reference family and run the same total number of chains and iterations.

**Related work.** The idea of adapting the tractable end-point of a path of distributions has been explored most thoroughly in the context of estimation of normalization constants [24, 23, 39, 14]. However, the importance of having both a fixed and adaptive reference has not appeared in this literature. In the PT literature, the closest related work appears to be [31, section 5] and [7, section 4.1], which consider PT algorithms with non-prior references. However, no adaptive algorithm is proposed; the reference distribution is set to a fixed variational approximation in [31] and to a Gaussian approximation at an analytically tractable mode in [7]. Reference adaption has been considered in passing in the annealed importance sampling and sequential Monte Carlo literature, but without carefully considering the importance of stabilization discussed in our work, e.g. [15]. Another related line of work is the design of adaptive independence Metropolis-Hastings (IMH) proposals [9, 12, 32, 26]. In particular, the theoretical work studying convergence of adaptive IMH has recognized the usefulness of combining the adaptive reference with a fixed reference [3, Section 7]. However, compared to the PT context, this combination is more straightforward to achieve in IMH via alternation of samplers. Finally, variational inference has previously been used to aid Monte Carlo methods, e.g., to learn proposals in sequential Monte Carlo [13], annealed importance sampling [34], and MCMC [25], and to form an initial reference for MCMC [17, 33]. None of these works connect the KL objective directly to the performance of a PT algorithm as we do in this paper.

## 2   Background

This work builds on a recent state-of-the-art PT method, non-reversible parallel tempering (NRPT) [36]. The goal is to sample from a target distribution, $\pi_1$, on a state space $\mathcal{X}$. This is achieved by constructing a sequence of $N + 1$ distributions from $\pi_1$ to a reference distribution $\pi_0$ also on $\mathcal{X}$. Given an *annealing schedule* $\mathcal{B}_N = (\beta_n)_{n=0}^N$, where $0 = \beta_0 < \beta_1 < \cdots < \beta_N = 1$ with mesh size $\|\mathcal{B}_N\| = \max_n |\beta_n - \beta_{n-1}|$, the distribution for each chain $n = 0, \ldots, N$ is given by $\pi_{\beta_n}$, where for all $\beta \in [0, 1]$, $\pi_\beta(x) \propto \pi_0^{1-\beta}(x) \cdot \pi_1^\beta(x)$.

---

**Algorithm 1** Non-reversible parallel tempering (NRPT)

---

**Require:** Initial state $\mathbf{x}_0$, annealing schedule $\mathcal{B}_N$, # iterations $T$, annealing path $\pi_\beta$

  $r_n \leftarrow 0$ for $n = 0, 1, \dots, N-1$

  **for** $t = 1, 2, \dots, T$ **do**

    $\mathbf{x} \leftarrow \texttt{LocalExploration}(\mathbf{x}_{t-1})$                 ▷ Local exploration kernels (e.g., HMC)

    $S_t \leftarrow$ **if** $t$ is even $S_{\text{even}}$ **else** $S_{\text{odd}}$

    **for** $n = 0, 1, \dots, N-1$ **do**

      $\alpha_n \leftarrow \alpha_n(\mathbf{x})$                         ▷ Acceptance probability from (1)

      $r_n \leftarrow r_n + (1 - \alpha_n)/T$          ▷ Store chain communication rejection rate estimates

      $U_n \leftarrow \text{Unif}(0, 1)$

      **if** $n \in S_t$ and $U_n \leq \alpha_n$ **then** $x^{n+1}, x^n \leftarrow x^n, x^{n+1}$     ▷ Swap components $n$ and $n+1$ of $\mathbf{x}$

    **end for**

    $\mathbf{x}_t \leftarrow \mathbf{x}$

  **end for**

  **Return:** $(\mathbf{x}_t)_{t=1}^T$, $(r_n)_{n=0}^{N-1}$

---

In parallel tempering, we simulate a Markov chain $\mathbf{X}_t = (X_t^0, \dots, X_t^N)$ that targets the product distribution $\boldsymbol{\pi}$ on $\mathcal{X}^{N+1}$ given by $\boldsymbol{\pi}(\mathbf{x}) = \pi_{\beta_0}(x^0) \cdot \pi_{\beta_1}(x^1) \cdots \pi_{\beta_N}(x^N)$ as the unique invariant probability distribution of the Markov chain. Note that the target distribution $\pi_1$ is a marginal of $\boldsymbol{\pi}$, hence Monte Carlo averages based on $X_t^N$ converge to the correct posterior expectations. Each iteration of PT involves two steps— *local exploration* and *communication*—as shown in Algorithm 1. The local exploration step involves applying, for each chain $n$, a $\pi_{\beta_n}$-invariant Markov kernel (e.g. Hamiltonian Monte Carlo). The communication step involves Metropolized swaps of states between adjacent chains; one alternates between swaps of chains $n$ and $n+1$ for all $n$ in $S_{\text{even}}$ followed by $S_{\text{odd}}$, where $S_{\text{even}}$ and $S_{\text{odd}}$ are the even and odd subsets of $\{0, \dots, N-1\}$, respectively. The deterministic alternation between even and odd swaps enjoys remarkable theoretical and empirical properties and ensures that the performance of parallel tempering does not degrade as $N$ increases [36]. Each proposed swap of component $n$ and $n+1$ of $\mathbf{x} = (x^0, \dots, x^N)$ is accepted or rejected independently with probability

$$\alpha_n(\mathbf{x}) = 1 \wedge \frac{\pi_{\beta_n}(x^{n+1}) \cdot \pi_{\beta_{n+1}}(x^n)}{\pi_{\beta_n}(x^n) \cdot \pi_{\beta_{n+1}}(x^{n+1})}. \tag{1}$$

To characterize the performance of a parallel tempering method we study how often samples *restart*, i.e. travel from the reference $\pi_0$ to the target $\pi_1$ [36]. Studying restarts isolates the effect of communication from the problem-specific characteristics of local exploration. (An improved local exploration kernel will of course improve overall performance.) When it is possible to obtain i.i.d. samples from $\pi_0$, the number of restarts is empirically found to be related to the effective sample size in the target distribution chain [36]. Formally, the *restart rate* $\tau(\mathcal{B}_N)$ from $\pi_0$ to $\pi_1$ with schedule $\mathcal{B}_N$ is the fraction of PT iterations that result in a restart. The maximum value of $\tau$ is $1/2$, since a communication swap is proposed with $\pi_0$ at every other iteration of PT. If the local exploration is efficient (Assumption A.2) then $\tau(\mathcal{B}_N) = (2 + 2\sum_{n=0}^{N-1} \frac{r_n}{1-r_n})^{-1}$ [36, Corollary 1], where $r_n = 1 - \mathbb{E}[\alpha_n(\mathbf{X})]$ and $\mathbf{X} \sim \boldsymbol{\pi}$. Asymptotically as $N \to \infty$, the round trip rate converges to a constant known as the *asymptotic restart rate* $\tau$: $\lim_{\|\mathcal{B}_N\| \to 0} \tau(\mathcal{B}_N) = \tau = (2 + 2\Lambda(\pi_0, \pi_1))^{-1}$ [36, Theorem 3], where $\Lambda(\pi_0, \pi_1)$ is the *global communication barrier* (GCB) between $\pi_0$ and $\pi_1$,

$$\Lambda(\pi_0, \pi_1) = \frac{1}{2} \int_0^1 \mathbb{E}[|\ell(X_\beta) - \ell(X'_\beta)|] \, \mathrm{d}\beta, \qquad \ell(x) = \log \frac{\pi_1(x)}{\pi_0(x)}, \qquad X_\beta, X'_\beta \overset{\text{i.i.d.}}{\sim} \pi_\beta. \tag{2}$$

The GCB can be estimated using the rejection rates $r_n$ for all adjacent pairs of chains, $\Lambda(\pi_0, \pi_1) \approx \sum_{n=0}^{N-1} r_n$, where $r_n = 1 - \mathbb{E}[\alpha_n(\mathbf{X})]$ for $\mathbf{X} \sim \boldsymbol{\pi}$, with the approximation error decreasing to zero at a rate $O(N^{-2})$ as the number of chains $N$ increases [36, Section 5.2]. GCB values near zero imply that swaps are typically accepted and communication is efficient.

## 3 Parallel tempering with a variational reference

A key degree of freedom one has when using PT is the reference distribution, $\pi_0$. Although the standard approach is to set $\pi_0$ to the prior, Eq. (2) suggests that the GCB might be quite large—and

hence communication performance quite poor—when the prior and posterior differ significantly, which commonly occurs in practice. Indeed, Proposition 3.1 motivates the importance of choosing the reference carefully: in a typical Bayesian model, as one obtains more data, the restart rate for the prior reference tends to zero. This result relies on Assumption B.3 in Appendix B, which stipulates standard technical conditions sufficient for, e.g., asymptotic consistency of the MLE, a Bernstein-von Mises result for asymptotic normality of the posterior [28], along with PT-specific assumptions such as efficient local exploration. We emphasize this result is a motivation for our methodology; the proposed algorithms apply much more broadly and not only in the data limit.

**Proposition 3.1** (Large-data restart rate, fixed reference). *Consider data $Y_m = \{Y_i\}_{i=1}^m$ generated i.i.d. from a model with likelihood $L(y|x_0)$, $x_0 \in \mathcal{X} \subset \mathbb{R}^d$, satisfying the conditions in Assumption B.3. Denote $\pi_{1,m}$ to be the posterior conditioned on $Y_m$. Then, in the large-data limit, the asymptotic restart rate $\tau_m$ associated with the annealing path from $\pi_0$ to $\pi_{1,m}$ degrades arbitrarily, i.e., $\tau_m \to 0$ almost surely as $m \to \infty$.*

In this section, we demonstrate that allowing the reference to be *tunable* addresses this issue.

## 3.1 Annealing paths with a variational reference

Let $\mathcal{Q} = \{q_\phi : \phi \in \Phi\}$ be a parametric family of distributions on $\mathcal{X}$, and for each $\phi \in \Phi$, denote the annealing path from the reference distribution $q_\phi$ to the target $\pi_1$ by

$$\forall \beta \in [0,1], \quad \pi_{\phi,\beta}(x) \propto q_\phi(x)^{1-\beta} \cdot \pi_1(x)^\beta.$$

Note that for this modified annealing path, the target distribution $\pi_1$ remains the same although the reference may change. In the Bayesian framework, this means that the prior $\pi_0$ and posterior $\pi_1$ remain the same while the variational reference $q_\phi$ is tuned. To ensure the variational reference family $\mathcal{Q}$ is compatible with the asymptotic PT theory developed in [36], we will assume $\mathcal{Q}$ is a *PT-suitable family* for the target $\pi_1$, i.e., each $q_\phi \in \mathcal{Q}$ shares the same support at the target and satisfies some mild moment conditions (Assumption A.1 in Appendix A). We will also assume throughout that the fixed reference $\pi_0$ is itself *PT-suitable* for $\pi_1$. PT-suitability is sufficient to guarantee that the restart rate is inversely related to the GCB and that the schedule-tuning procedure from [36, Section 5.4] is justified.

## 3.2 Exponential variational reference family

The variational reference family $\mathcal{Q}$ should be flexible enough to match the target $\pi_1$ reasonably well, but also simple enough to enable i.i.d. sampling, pointwise evaluation, and tractable optimization. Proposition 3.2 suggests that in the large-data limit, the family of multivariate Gaussian distributions often suffices. In particular, unlike the fixed prior reference—whose restart rate decays to zero in the large-data limit—there exists a sequence of multivariate normal reference distributions so that the restart rate tends to its maximum value of $1/2$. Note again that this large-data setting is just one instance in which a tunable reference helps; our method in this work applies much more broadly, and does not require a Gaussian reference or rely on the asymptotic setup in Proposition 3.2. In particular, our method is advantageous in any setting where the GCB decreases compared to fixed-reference PT.

**Proposition 3.2** (Large-data restart rate, variational reference). *Consider the setting of Proposition 3.1, and suppose $\mathcal{Q} = \{\mathcal{N}(\mu,\Sigma) : \mu \in \mathbb{R}^d, \Sigma \in \mathbb{R}^{d \times d}, \Sigma = \Sigma^\top \succ 0\}$ is a PT-suitable family for all targets $\pi_{1,m}$ almost surely. Then for any fixed $N > 1$, there exists a random sequence $\mu_m \in \mathbb{R}^d$, $\Sigma_m \in \mathbb{R}^{d \times d}$ such that for any schedule $\mathcal{B}_N$, in the large-data limit, the restart rate $\tau_m(\mathcal{B}_N)$ associated with $\pi_{1,m}, \mathcal{N}(\mu_m, \Sigma_m)$ converges to the maximum possible value. I.e., for any schedule $\mathcal{B}_N$ we have $\tau_m(\mathcal{B}_N) \xrightarrow{p} 1/2$ as $m \to \infty$.*

Proposition 3.2 motivates the use of a tunable variational reference that can adapt to the target, as opposed to a fixed reference. In this work, we consider the general class of exponential reference families of full-rank where the distributions take the form $q_\phi(x) = h(x) \exp(\phi^\top \eta(x) - A(\phi))$, for base density $h$, natural parameter $\phi$, sufficient statistic $\eta$, and log partition function $A$. Aside from being flexible enough to match posteriors arbitrarily well in the large-data limit, a key advantage of an exponential reference family is that it is straightforward to fit: one can obtain the forward (inclusive) KL divergence minimizer $q_{\phi_{KL}}$ using a simple gradient-free moment matching procedure because $\mathbb{E}_{\phi_{KL}}[\eta] = \mathbb{E}_1[\eta]$, where $\mathbb{E}_\phi$ and $\mathbb{E}_1$ denote expectations with respect to $q_\phi$ and $\pi_1$, respectively [16].

Indeed, under slightly more stringent technical assumptions in the setting of Propositions 3.1 and 3.2—namely Assumption B.9—Proposition 3.3 shows that we may use this forward KL fit as the reference sequence $\mathcal{N}(\mu_m, \Sigma_m)$ for which the restart rate is asymptotically maximized. Assumption B.9 stipulates that the differences between the posterior mean and MLE, as well as between the inverse Fisher information and scaled posterior variance, are not too large.

**Proposition 3.3** (Large-data restart rate, moment matched reference). *Consider the setting of Proposition 3.2 and suppose that Assumption B.9 also holds. Then, the conclusion of Proposition 3.2 holds if $\mu_m, \Sigma_m$ are set to the mean and variance of $\pi_{1,m}$ conditioned on $\boldsymbol{Y}_m$, respectively.*

### 3.3 Tuning the variational reference

In practice, we fit the exponential family reference iteratively by running NRPT for multiple tuning rounds $r = 1, \ldots, R$; in each tuning round $r$ we run $T_r = 2^r$ iterations with variational parameter $\hat{\phi}_r$. Using the generated states $(\mathbf{X}_{t,r})_{t=1}^{T_r}$, we obtain the parameter $\hat{\phi}_{r+1}$ for round $r + 1$ by solving

$$\mathbb{E}_{\hat{\phi}_{r+1}}[\eta] = \frac{1}{T_r} \sum_{t=1}^{T_r} \eta(X_{t,r}^N). \tag{3}$$

Note that by relying on sufficient statistics, we are not required to keep in memory the MCMC trace or to loop over MCMC samples when performing variational parameter optimization. For example, when $\mathcal{Q}$ is a Gaussian family, Eq. (3) simplifies to setting the mean vector and covariance matrix to the empirical mean and covariance obtained from the target chain samples $X_{t,r}^N$. When a full (non-diagonal) covariance matrix is used, one should start tuning when $T_r \geq d$. We additionally use the samples from each round to tune the annealing schedule $\mathcal{B}_N$ using the procedure from [36].

Theorem 3.4 shows that if the absolute spectral gap $\mathrm{Gap}(\phi)$ [11] of the PT Markov chain with reference $q_\phi$ is bounded away from zero, and the number of iterations in each round tends to infinity at an appropriate rate, then $\hat{\phi}_r$ will converge almost surely to the forward KL minimizer $\phi_{\mathrm{KL}}$. Although Theorem 3.4 stipulates that $\eta$ is bounded, this is a technicality that is not required in practice.

**Theorem 3.4** (Convergence of variational reference tuning). *Suppose $\mathcal{Q} = \{q_\phi : \phi \in \Phi\}$ is a PT-suitable exponential family of full rank with sufficient statistic $\eta(x)$ bounded in $x$. Further assume that $\phi_{KL}$ exists and is unique. Suppose each round of tuning starts at stationarity and there is $\kappa > 0$ such that $\mathrm{Gap}(\phi) \geq \kappa > 0$ for all $\phi$ and $T_r = \Omega(2^r)$ as $r \to \infty$. Then, (1): $\hat{\phi}_r \to \phi_{KL}$ almost surely as $r \to \infty$; and (2): for all $0 < \epsilon < \frac{1}{2}$, almost surely there exists an $R(\epsilon)$ such that for all $r \geq R(\epsilon)$, $\|\mathbb{E}_{\hat{\phi}_r}[\eta] - \mathbb{E}_1[\eta]\| \leq T_r^{-\frac{1}{2} + \epsilon}$.*

### 3.4 Forward KL as a surrogate objective

We now provide a general theoretical justification for the minimization of the forward KL divergence as opposed to the global communication barrier, which is appealing as it enables a simple gradient-free moment matching procedure. First, note that when $\pi_1 \in \mathcal{Q}$, minimizing the KL divergence and the GCB is equivalent since they are both divergences [36]. Theorem 3.5 generalizes this to the more usual case where $\pi_1 \notin \mathcal{Q}$, demonstrating that the GCB at the forward KL minimum is bounded by quantities that depend on the flexibility of the variational family. In particular, provided that there exists a $\phi_0 \in \Phi$ such that the difference between log densities of the target $\pi_1$ and reference $q_{\phi_0}$ is bounded by a function $g$, then the GCB evaluated at the forward KL minimizer is bounded by a term involving expectations of $g$ under the target and distributions $q_\phi$ that are close to $\pi_1$.

**Theorem 3.5** (Forward KL proxy for the GCB). *Suppose that $\mathcal{Q} = \{q_\phi : \phi \in \Phi\}$ is a PT-suitable exponential family of full rank. Let $g$ be any function such that for some $\phi_0 \in \Phi$ and for all $x \in \mathcal{X}$, $|\log \pi_1(x) - \log q_{\phi_0}(x)| \leq g(x)$. Then, if $\phi_{KL} = \arg\min_\phi KL(\pi_1 \| q_\phi)$ exists and is unique, we have that $\Lambda(q_{\phi_{KL}}, \pi_1) \leq \sqrt{\frac{1}{2}\left(\mathbb{E}_1[g] + \sup_{\phi \in \Phi'} \mathbb{E}_\phi[g]\right)}$, where $\Phi' = \{\phi : KL(\pi_1 \| q_\phi) \leq KL(\pi_1 \| q_{\phi_0})\}$.*

We consider in Appendix D two simple examples to verify that the upper bound given by Theorem 3.5 is small enough for practical purposes.

**Algorithm 2** Variational PT

---

**Require:** initial state $\mathbf{x}_0$, # chains $\bar{N} = 2N$, # total tuning rounds $R$, target $\pi_1$, reference family $\mathcal{Q} = \{q_\phi : \phi \in \Phi\}$, initial reference parameter $\phi$, fixed reference $\pi_0$
$\quad \mathcal{B}_{\phi,N}, \mathcal{B}_N \leftarrow (0, 1/N, 2/N, \ldots, 1)$        $\triangleright$ Initialize annealing parameters uniformly with $N_\phi = N$
$\quad$ **for** $r = 1, 2, \ldots, R$ **do**
$\qquad T \leftarrow 2^r$        $\triangleright$ Double the number of iterations in the next tuning round
$\qquad \bar{\pi}_{\phi,\beta} \leftarrow \texttt{Concatenate}(\pi_{\phi,\beta}, \pi_\beta)$        $\triangleright$ Concatenate paths using (4)
$\qquad \bar{\mathcal{B}}_{\phi,\bar{N}} \leftarrow \texttt{Concatenate}(\mathcal{B}_{\phi,N}, \mathcal{B}_N)$        $\triangleright$ Concatenate schedules using (5)
$\qquad (\mathbf{x}_t)_{t=1}^T, (r_n)_{n=0}^{\bar{N}-1} \leftarrow \texttt{NRPT}(\mathbf{x}_0, \bar{\mathcal{B}}_{\bar{N}}, T, \bar{\pi}_{\phi,\beta})$        $\triangleright$ PT with two references
$\qquad \mathcal{B}_{\phi,N} \leftarrow \texttt{UpdateSchedule}((r_n)_{n=0}^{N-1}, \mathcal{B}_{\phi,N})$        $\triangleright$ Tune annealing parameters for $\pi_\phi$ [36]
$\qquad \mathcal{B}_N \leftarrow \texttt{UpdateSchedule}((r_n)_{n=\bar{N}-1}^N, \mathcal{B}_N)$        $\triangleright$ Tune annealing parameters for $\pi$ [36]
$\qquad \mathbf{x}_0 \leftarrow \mathbf{x}_T$        $\triangleright$ Initialization for next round
$\qquad \phi \leftarrow \texttt{UpdateReference}((\mathbf{x}_t)_{t=1}^T)$        $\triangleright$ Tune according to Eq. (3) or another procedure
$\quad$ **end for**
$\quad$ **Return:** $(\mathbf{x}_t)_{t=1}^T$

---

## 3.5 Stabilization with a fixed reference

In Section 3.3 we provided a result (Theorem 3.4) guaranteeing convergence of the adaptive scheme assuming the existence of an absolute spectral gap. In practice, the risk is that certain regions of $\Phi$ may significantly degrade the absolute spectral gap under the basic variational scheme discussed so far. For example, as shown in Fig. 2 (middle row), if the posterior is multimodal, the variational reference may quickly center on one mode; because subsequent rounds of tuning use samples that depend on the variational reference, these samples may largely come from that one mode, causing the variational reference to remain trapped there for many tuning rounds.

To address this issue, we introduce parallel tempering with two reference distributions, using both the original (fixed) reference and a variational reference, which we call "stabilized variational PT", illustrated in Fig. 1. We create an annealing path that starts at $q_\phi$, proceeds along an annealing path to $\pi_1$, and then moves on a new path from $\pi_1$ to $\pi_0$, connecting all three distributions. This modification adds significant robustness; as long as there are some restarts from the fixed reference, the target chain will escape the local optima and provide a more accurate estimate of $\pi_1$ used to tune $q_\phi$. In general, a well-tuned variational reference can provide a significant reduction in GCB compared to just a fixed path, but keeping the fixed reference ensures that the method will never do significantly worse than standard NRPT even if the variational reference tuning performs poorly (Theorem 3.6 below). Our variational PT algorithm with two references is presented in Algorithm 2, in which `UpdateSchedule` refers to [36, Algorithms 2, 3].

To formalize the notion of a piecewise path, let $\pi_{\phi,\beta}$ be the annealing path between $q_\phi$ and $\pi_1$, and let $\pi_\beta$ be the annealing path between the fixed reference $\pi_0$ and $\pi_1$. We define the concatenated (piecewise) path $\bar{\pi}_{\phi,\beta}$,

$$\bar{\pi}_{\phi,\beta} = \begin{cases} \pi_{\phi,2\beta} & 0 \leq \beta \leq \frac{1}{2}, \\ \pi_{2-2\beta} & \frac{1}{2} \leq \beta \leq 1. \end{cases} \tag{4}$$

This new annealing path can be used within the NRPT Algorithm 1 as any other path. To tune the annealing schedule within each leg of PT with two references, we define the schedules $\mathcal{B}_{\phi,N_\phi} = (\beta_{\phi,n})_{n=0}^{N_\phi}$ and $\mathcal{B}_N = (\beta_n)_{n=0}^N$ for the legs connecting $q_\phi$ and $\pi_0$ to $\pi_1$, respectively. Then, we define the concatenated schedule $\bar{\mathcal{B}}_{\phi,\bar{N}} = (\bar{\beta}_{\phi,n})_{n=0}^{\bar{N}}$ where $\bar{N} = N_\phi + N$ and

$$\bar{\beta}_{\phi,n} = \begin{cases} \frac{1}{2}\beta_{\phi,n} & 0 \leq n \leq N_\phi \\ 1 - \frac{1}{2}\beta_{\bar{N}-n} & N_\phi \leq n \leq \bar{N} \end{cases}. \tag{5}$$

This concatenated schedule $\bar{\mathcal{B}}_{\phi,\bar{N}}$ and path $\bar{\pi}_{\phi,\beta}$ are provided as input to the NRPT algorithm.

Finally, we provide an analysis of the worst-case performance of variational PT with two reference distributions. We show that the asymptotic restart rate of PT with two references is always greater than or equal to the restart rate with either one of the two references alone. Because PT with two references requires twice the amount of computation, this amounts to a worst case of half the performance of regular PT with a fixed reference. In practice we often find that including a variational reference substantially improves the PT restart rate.

Let $\bar{\tau}(\bar{\mathcal{B}}_{\phi,\bar{N}})$ be the restart rate for $\pi_1$ for the concatenated path, i.e. the rate at which samples from either reference $q_\phi$ or $\pi_0$ reach the target $\pi_1$. Theorem 3.6 shows that if the Markov chain *efficiently explores locally* (Assumption A.2), then the restart rate of multiple-reference PT is the sum of the restart rate for $\pi_1$ between $q_\phi$ and $\pi_0$ denoted $\tau_\phi(\mathcal{B}_{\phi,N_\phi})$ and $\tau(\mathcal{B}_N)$ respectively.

**Theorem 3.6** (Restart rate of NRPT with two reference distributions). *Let $q_\phi, \pi_0$ be PT-suitable references for the target $\pi_1$. Suppose the PT chains with references $q_\phi, \pi_0$ with schedules $\mathcal{B}_{\phi,N_\phi}, \mathcal{B}_N$ respectively efficiently explore locally (see Assumption A.2). Then $\bar{\tau}_\phi(\bar{\mathcal{B}}_{\phi,\bar{N}}) = \tau_\phi(\mathcal{B}_{\phi,N_\phi}) + \tau(\mathcal{B}_N)$. Moreover, if $\|\bar{\mathcal{B}}_{\phi,\bar{N}}\| \to 0$, then*

$$\lim_{\bar{N}\to\infty} \bar{\tau}_\phi(\bar{\mathcal{B}}_{\phi,\bar{N}}) = \frac{1}{2 + 2\Lambda(q_\phi, \pi_1)} + \frac{1}{2 + 2\Lambda(\pi_0, \pi_1)}.$$

## 4 Experiments

We consider various Bayesian inference problems: 11 based on real data, and 4 based on synthetic data (see Table 1 in Appendix F for the details of each). The range of problem settings considered include spatial statistics, Bayesian ODE parameter inference, phylogenetic inference, and several distinct Bayesian hierarchical models. In all examples the variational reference is a multivariate normal distribution with either a diagonal estimated covariance matrix (`VPT_diag`) or a full covariance matrix (`VPT_full`). The code for the experiments is made publicly available: Julia code is available at `https://github.com/UBC-Stat-ML/VariationalPT` and Blang code is at `https://github.com/UBC-Stat-ML/bl-vpt-nextflow`. A distributed implementation is also under development at `https://github.com/Julia-Tempering/Pigeons.jl`. Experimental details can be found in Appendix F.

### 4.1 Comparative efficiency of variational PT families and a PT baseline

We begin by comparing the communication efficiency of both `VPT_diag` and `VPT_full` to a state-of-the-art existing PT method, `NRPT` [36], which uses a single, fixed reference. Note that there is also a computational trade-off between the two variational families (but we remind the reader that both still yield convergence of the target chain to the posterior). In particular, `VPT_full` offers more flexibility—and hence a potentially lower GCB—at the cost of a higher computational cost per iteration and more variational parameters to fit, while `VPT_diag` has the same asymptotic computational cost per iteration as standard PT methods. We explore this tradeoff in Fig. 3.

We observe that both `VPT_diag` and `VPT_full` often substantially improve the restart rate compared to the `NRPT` baseline, and at worst achieve similar performance. The choice between `VPT_full` and `VPT_diag` depends on the problem: for example, in low-dimensional problems such as `Challenger` and `Simple-mix`, we observe that the full covariance in `VPT_full` is worth its additional cost per iteration. The situation is reversed in the `Transfection` problem. If one is pressed to select one PT variational family, we recommend `VPT_diag` as a safe default in light of Theorem 3.6 and of its computational cost per iteration asymptotically equivalent to `NRPT`.

We also note that the tuning procedure converges relatively quickly. We show the number of restarts for the first 2.5% of computation time as insets in Fig. 3, and find that the number of restarts for the three methods can be distinguished early on. We also show in Fig. 4(a) that the GCB estimates converge in a small number of rounds for two additional representative problems (`Lip Cancer` and `Vaccines`). Note also that the GCB is substantially lower when a variational reference is introduced.

### 4.2 Moment matching outperforms stochastic optimization

Next, we compare the proposed moment matching procedure described in Section 3 to several other stochastic optimization schemes to tune the variational reference. In contrast to moment matching—which is free of tuning hyper-parameters—we show in Appendix F.11.2 that stochastic optimization schemes require extensive hyper-parameter tuning, including step size schedule, choice of optimizer, and surrogate objective function. Moreover, we show in Fig. 5 (left) that it is difficult to specify a "default" hyper-parameter setting for stochastic optimization methods; a setting that works well on one problem (`Rocket`) generally will not work well on other problems (e.g., `Change Point`, `Titanic`). In contrast, moment matching performed well on all 14 problems we considered without requiring tuning. Consequently, our moment matching procedure is a better candidate for integration

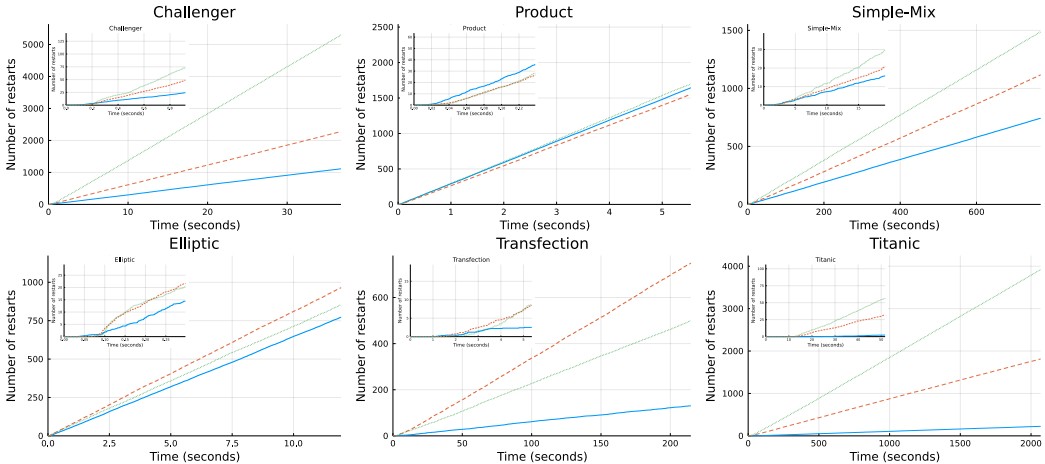

Figure 3: Number of restarts (higher is better) versus computation time in seconds for several models, with `VPT_full` in green, `VPT_diag` in red, and the `NRPT` baseline in blue. The two variational PT methods generally provide a comparable or better rate of restarts per second. Insets highlight the initial 2.5% of computation time, demonstrating that tuning of the variational references stabilizes quickly.

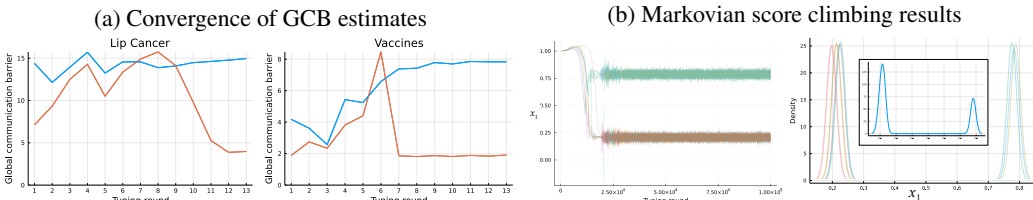

Figure 4: **(a)** GCB for `VPT_diag` (red) and `NRPT` (blue) versus tuning round. **(b, left)** Tuning variational parameters using `MSC` with 10 replications (colours). The mean of the variational distribution for the model parameter $x_1$ in the `Transfection` model is presented. The true marginal posterior distribution of $x_1$ is bimodal (Fig. 2). We see here that `MSC` chooses one of the two modes for the estimation of the mean parameter. **(b, right)** Variational Gaussian approximation of the same parameter in the `Transfection` model produced by 10 `MSC` runs with different seeds (colours). The aggregate of the different runs is shown as an inset, cf. Fig. 2.

in probabilistic programming languages (PPLs), in which users do not expect to be required to frequently change algorithmic tuning hyper-parameters. To illustrate this point, we have extended an existing open source PPL, Blang [6], to include our method tuned via moment matching (code available at `https://github.com/UBC-Stat-ML/bl-vpt`).

### 4.3 Comparison to an externally tuned reference

We also compare our moment-matching variational reference tuning procedure to a reference tuned by an existing procedure outside of the PT context. In particular, we tested Markovian score climbing (`MSC`) [29]—which also optimizes the forward KL—for tuning a Gaussian reference with a diagonal covariance. Details of `MSC` tuning, including sensitivity to stochastic optimization settings, can be found in Appendix F.11. The results for the `Transfection` model are presented in Fig. 4. In contrast to our stabilized moment matching approach (Fig. 2), tuning the reference using `MSC` in this example results in systematic catastrophic forgetting of one of the modes (Fig. 4 (b)) in all 10 replicates. We additionally refer readers to [21] for recent developments in score-based methods to minimize the forward KL divergence.

### 4.4 Comparison of variational PT topologies

Finally, we compare the three algorithms introduced in Fig. 1 (`Basic Variational PT`, `Stabilized Variational PT` and `NRPT`) on nine Bayesian inference problems (see Fig. 5, right).

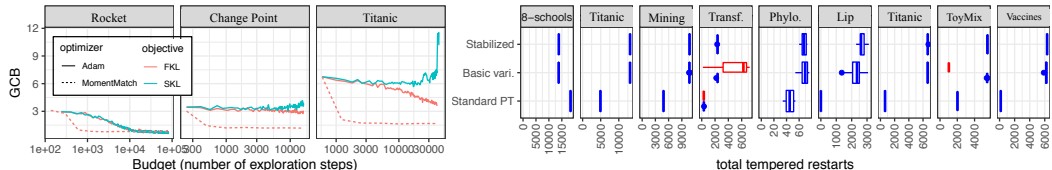

Figure 5: **Left:** comparison of stochastic optimization and moment matching. An optimizer parameter setting that works well for the `Rocket` problem (Adam+FKL/SKL, step size scale 0.1) does not generalize well to other problems (`Change-Point`, `Titanic`). In contrast, moment matching reliably finds a well-tuned reference in all 14 problems considered in this paper. **Right:** the same plot as in Fig. 2, but on a larger selection of models.

We set up the algorithms so that the runtime per PT iteration has the same asymptotic complexity: this is done by selecting a diagonal covariance matrix for the Gaussian variational family and using the same total number of chains for all methods.

The results confirm the initial findings of Fig. 2: only the stabilized method always avoids catastrophic forgetting of modes. Moreover, in all but one example considered, we find that `Stabilized Variational PT` exhibits improved performance in terms of the number of restarts compared to the `NRPT` baseline (Fig. 5). The exception is the `8-schools` problem, in which the posterior is not well approximated by a diagonal Gaussian. Even in this case, as suggested by Theorem 3.6, the performance does not degrade by more than a factor two. At the other end of the spectrum, for the `Vaccines` hierarchical model, the number of restarts increases >40-fold compared to the `NRPT` baseline, and for the spatial sparse conditional auto-regressive (CAR) model applied to the `Lip Cancer` problem, the performance jumps from zero restarts to >2300 restarts.

More results and details can be found in Appendix F.11.3, including alternative topological arrangements of variational PT algorithms, effective sample size per second results, as well as global communication barriers for the problems considered in this section.

## 5   Conclusion

This paper addressed sampling from a complex target distribution within the parallel tempering framework by constructing a generalized annealing path connecting the posterior to an adaptively tuned variational reference. Experiments in a wide range of realistic Bayesian inference scenarios demonstrate the large empirical gains achieved by our method. Potential future work includes extending the gradient-free tuning methodology to larger classes of variational families for the reference distribution. Further, heavy-tailed distributions can violate Assumption A.1 and it is not clear without further examination what the implications are on the convergence of the proposed algorithms. Therefore, another possible direction for future work is to develop methods suitable for heavy-tailed target distributions.

## Acknowledgements

NS acknowledges the support of a Vanier Canada Graduate Scholarship. ABC and TC acknowledge the support of an NSERC Discovery Grant. We also acknowledge use of the ARC Sockeye computing platform from the University of British Columbia.

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
