## A  PT-suitable reference family

**Assumption A.1** (PT-suitable reference)**.** *We say $\pi_0$ is PT-suitable for the target $\pi_1$ if:*

1. *(Full support):* $\operatorname{supp}(\pi_0) = \operatorname{supp}(\pi_1)$.

2. *(Regularity): The log-likelihood ratio between $\pi_1$ and $\pi_0$, $\ell(x) = \log \frac{\pi_1(x)}{\pi_0(x)}$ satisfies,*

$$\max\{\mathbb{E}_0[|\ell|^3], \, \mathbb{E}_1[|\ell|^3]\} < \infty.$$

   *where we denote $\mathbb{E}_0$, and $\mathbb{E}_1$ as the expectation with respect to $\pi_0$ and $\pi_1$ respectively.*

*We say that a family $\mathcal{Q} = \{q_\phi : \phi \in \Phi\}$ is PT-suitable for the target $\pi_1$ if for all $\phi$ the conditions above hold with $q_\phi$ in place of $\pi_0$.*

**Assumption A.2** (Efficient local exploration)**.** *Suppose $\pi_0$ is a PT-suitable reference for $\pi_1$, with log-likelihood $\ell(x) = \log \pi_1(x) - \log \pi_0(x)$ and schedule $\mathcal{B}_N$. Let $\mathbf{X}_t = (X_t^0, \dots, X_t^N)$ be the PT chain stationary with respect to*

$$\boldsymbol{\pi}(\mathbf{x}) = \prod_{n=0}^{N} \pi_{\beta_n}(x^n)$$

*and $K_{\beta_n}$ be the $\pi_{\beta_n}$-stationary Markov kernel for the local exploration step. We will say $\mathbf{X}_t$ efficiently explores locally if [36, Section 3.3],*

1. *Stationarity:* $\mathbf{X}_0 \sim \boldsymbol{\pi}$.

2. *Efficient local exploration (ELE) : For all $n$ and $t$, if $\bar{X}_t^n \sim K_{\beta_n}(X_t^n, \mathrm{d}\bar{x})$, then $\ell(X_t^n)$ is independent of $\ell(\bar{X}_t^n)$.*

## B  Large-data Asymptotics

### B.1  Conditional convergence in distribution

Suppose $(\mathcal{X}, d_{\mathcal{X}})$ is a metric space and let $X, X_1, X_2, \dots$ be random variables taking values in $\mathcal{X}$. Define a sequence of $\sigma$-algebras $(\mathcal{F}_m)_{m=1}^{\infty}$ such that $\mathcal{F}_m \subset \mathcal{F}_{m+1}$. As $m \to \infty$, we say $X_m | \mathcal{F}_m$ converges in distribution to $X$, denoted $X_m | \mathcal{F}_m \xrightarrow{d} X$, if for all bounded and continuous $f : \mathcal{X} \to \mathbb{R}$,

$$\mathbb{E}[f(X_m)|\mathcal{F}_m] \xrightarrow[m \to \infty]{a.s.} \mathbb{E}[f(X)].$$

Similarly, we define conditional convergence in probability, denoted $X_m | \mathcal{F}_m \xrightarrow{p} X$, if for all $\epsilon > 0$,

$$\mathbb{P}(d_{\mathcal{X}}(X_m, X) > \epsilon | \mathcal{F}_m) \xrightarrow[m \to \infty]{a.s.} 0.$$

**Lemma B.1** (Conditional portmanteau lemma)**.** *The following are equivalent:*

1. $X_m | \mathcal{F}_m \xrightarrow{d} X$ *as $m \to \infty$.*

2. $\mathbb{E}[f(X_m)|\mathcal{F}_m] \xrightarrow{a.s.} \mathbb{E}[f(X)]$ *as $m \to \infty$, for all bounded Lipschitz functions $f : \mathcal{X} \to \mathbb{R}$.*

3. $\mathbb{P}[X_m \in A | \mathcal{F}_m] \xrightarrow{a.s} \mathbb{P}[X \in A]$ *as $m \to \infty$ for all $A \subset \mathcal{X}$ such that $\mathbb{P}[X \in \partial A] = 0$.*

The proof of this Lemma is identical to the portmanteau lemma for weak convergence by replacing probabilities/expectations with conditional probabilities/expectations (for example, see [38, Section 2.1]).

**Lemma B.2.** *Suppose $X, X_1, X_2, \dots$ and $X', X_1, X_2, \dots$ are $\mathcal{X}$-valued random variables. Then, the following hold:*

1. *If $X_m | \mathcal{F}_m \xrightarrow{d} X$ as $m \to \infty$ then $X_m \xrightarrow{d} X$.*

2. If $X_m|\mathcal{F}_m \xrightarrow{p} X$ as $m \to \infty$ then $X_m \xrightarrow{p} X$.

3. If $X_m|\mathcal{F}_m \xrightarrow{p} X$ as $m \to \infty$ then $X_m|\mathcal{F}_m \xrightarrow{d} X$.

4. If $X_m|\mathcal{F}_m \xrightarrow{d} X$ as $m \to \infty$, and $X$ is a constant a.s., then $X_m|\mathcal{F}_m \xrightarrow{p} X$.

5. Fatou's lemma: If $X_m|\mathcal{F}_m \xrightarrow{d} X$ as $m \to \infty$, then for all $f : \mathcal{X} \to [0, \infty)$,
$$\liminf_{m \to \infty} \mathbb{E}[f(X_m)|\mathcal{F}_m] \geq \mathbb{E}[f(X)], \quad a.s.$$

6. Continuous mapping theorem: If $\mathcal{X}'$ is a metric space and $g : \mathcal{X} \to \mathcal{X}'$ is a continuous function, then
$$X_m|\mathcal{F}_m \xrightarrow[m \to \infty]{d} X \quad \implies \quad g(X_m)|\mathcal{F}_m \xrightarrow[m \to \infty]{d} g(X).$$

7. Slutsky's theorem: If $X_m|\mathcal{F}_m \xrightarrow{d} X$ and $d_{\mathcal{X}}(X_m', X_m)|\mathcal{F}_m \xrightarrow{p} 0$, then $X_m'|\mathcal{F}_m \xrightarrow{d} X$.

8. Suppose $\mathcal{X} = \mathbb{R}^d$, with $X_m|\mathcal{F}_m \xrightarrow{d} X$. Suppose $A, A_1, \cdots \in \mathbb{R}^{d \times d}$ such that $A, A_m \in \mathcal{F}_m$ and as $m \to \infty$ $A_m \xrightarrow{a.s.} A$, where $A$ is a constant. Then,
$$A_m X_m|\mathcal{F}_m \xrightarrow[m \to \infty]{d} AX.$$

*Proof of Lemma B.2.*      1. For any bounded and continuous $f$,
$$\lim_{m \to \infty} \mathbb{E}[f(X_m)] = \mathbb{E}\left[\lim_{m \to \infty} \mathbb{E}[f(X_m)|\mathcal{F}_m]\right] = \mathbb{E}[f(X)],$$
We can exchange the expectation and limit by the dominated convergence theorem.

2. For $\epsilon > 0$,
$$\mathbb{P}(d_{\mathcal{X}}(X_m, X) > \epsilon) = \mathbb{E}[\mathbb{P}(d_{\mathcal{X}}(X_m, X) > \epsilon|\mathcal{F}_m)]$$
Since $\mathbb{P}(d_{\mathcal{X}}(X_m, X) > \epsilon|\mathcal{F}_m) \xrightarrow{a.s.} 0$ as $m \to \infty$, the result follows from the dominated convergence theorem.

3. Let $f$ be a $\kappa$-Lipschitz function bounded by $M$. Let $\epsilon > 0$,
$$\begin{aligned}|\mathbb{E}[f(X_m) - f(X)|\mathcal{F}_m]| &\leq \mathbb{E}[|f(X_m) - f(X)|1(d_{\mathcal{X}}(X_m, X) \leq \epsilon)|\mathcal{F}_m] \\ &\quad + \mathbb{E}[|f(X_m) - f(X)|1(d_{\mathcal{X}}(X_m, X) > \epsilon)|\mathcal{F}_m] \\ &= \kappa\epsilon + 2M\mathbb{P}(d_{\mathcal{X}}(X_m, X) > \epsilon|\mathcal{F}_m).\end{aligned}$$
Since $X_m \xrightarrow{p} X$ as $m \to \infty$, we have $\mathbb{P}(d_{\mathcal{X}}(X_m, X) > \epsilon|\mathcal{F}_m) \xrightarrow{a.s.} 0$, therefore
$$\lim_{m \to \infty} |\mathbb{E}[f(X_m) - f(X)|\mathcal{F}_m]| \leq \kappa\epsilon, \quad a.s.$$
The result follows by taking $\epsilon \to 0$.

4. Since $X$ is a.s. constant there exits $x_0$ such that $\mathbb{P}(X = x_0) = 1$. Then for all $\epsilon > 0$, if $A_\epsilon = \{x : d_{\mathcal{X}}(x, x_0) > \epsilon\}$, we have $\mathbb{P}(X \in A_\epsilon) = 0$. Since $X_m|\mathcal{F} \xrightarrow{d} X$, we have
$$\mathbb{P}(d_{\mathcal{X}}(X_m, X) > \epsilon|\mathcal{F}_m) = \mathbb{P}(X_m \in A_\epsilon|\mathcal{F}_m) \xrightarrow[m \to \infty]{a.s.} \mathbb{P}(X \in A_\epsilon) = 0.$$

5. We adapt the proof of Fatou's lemma that holds for random variables that converge in distribution instead of almost surely adapted from [19, Lemma 5.11].

   For any $K > 0$, we have $x \to x \wedge K$ is a bounded and continuous function. Since $X_m|Y_m \xrightarrow{d} X$, this implies that, almost surely,
$$\liminf_{m \to \infty} \mathbb{E}[X_m|\mathcal{F}_m] \geq \lim_{m \to \infty} \mathbb{E}[X_m \wedge K|\mathcal{F}_m] = \mathbb{E}[X \wedge K].$$

Since this is true for any $K$, and $X \wedge K \xrightarrow{a.s.} X$ as $K \to \infty$, by the monotone convergence theorem,
$$\liminf_{m \to \infty} \mathbb{E}[X_m | \mathcal{F}_m] \geq \lim_{K \to \infty} \mathbb{E}[X \wedge K] = \mathbb{E}[X].$$

6. Fix any bounded and continuous $f : \mathcal{X}' \to \mathbb{R}$. Because $f \circ g : \mathcal{X} \to$ is a bounded and continuous function, and $X_m | \mathcal{F}_m \xrightarrow{d} X$, we have,
$$\mathbb{E}[f(g(X_m)) | \mathcal{F}_m] \xrightarrow[m \to \infty]{a.s.} \mathbb{E}[f(g(X))].$$

7. Let $f$ be a $\kappa$-Lipschitz function bounded by $M$. By triangle inequality,
$$|\mathbb{E}[f(X'_m) - f(X) | \mathcal{F}_m]| \leq \mathbb{E}[|f(X'_m) - f(X_m)| | \mathcal{F}_m]| + |\mathbb{E}[f(X_m) - f(X) | \mathcal{F}_m]|.$$

Since $X_m \xrightarrow{d} X$, we have $\mathbb{E}[f(X_m) - f(X) | \mathcal{F}_m] \xrightarrow{a.s.} 0$. Also for all $\epsilon > 0$, let $A_{\epsilon,m} = \{d_{\mathcal{X}}(X_m, X'_m) \leq \epsilon\}$. Note that
$$\begin{aligned}
\mathbb{E}[|f(X_m) - f(X'_m)| | \mathcal{F}_m] &= \kappa \mathbb{E}[|f(X_m) - f(X'_m)| 1(A_{\epsilon,m}) | \mathcal{F}_m] \\
&\quad + \mathbb{E}[|f(X_m) - f(X'_m)| 1(A^c_{\epsilon,m}) | \mathcal{F}_m] \\
&\leq \kappa \epsilon + 2M \mathbb{P}(A^c_{\epsilon,m} | \mathcal{F}_m)
\end{aligned}$$

Since $d_{\mathcal{X}}(X_m, X'_m) | \mathcal{F}_m \xrightarrow{p} 0$, we have $\mathbb{P}(A^c_{\epsilon,m} | \mathcal{F}_m) \xrightarrow{a.s} 0$. Also since $X_m | \mathcal{F}_m \xrightarrow{d} X$, we have $\mathbb{E}[f(X_m) - f(X) | \mathcal{F}_m] \xrightarrow{a.s.} 0$.

8. Fix $\epsilon > 0$. Note that
$$\begin{aligned}
P(\|A_m - A\| > \epsilon | \mathcal{F}_m) &= \mathbb{E}[1(\|A_m - A\| > \epsilon) | \mathcal{F}_m] \\
&= 1(\|A_m - A\| > \epsilon) \\
&\to 0 \quad a.s.
\end{aligned}$$

This implies $A_m | \mathcal{F}_m \xrightarrow{p} A$ as $m \to \infty$.

Note that $(X_m, A) | \mathcal{F}_m \xrightarrow{d} (X, A)$ by the continuous mapping theorem with $x \to (x, A)$. Next, note that $\|(X_m, A_m) - (X_m, A)\| = \|A_m - A\|$, where use the matrix element-wise Euclidean norm. We are given that $A_m | \mathcal{F}_m \xrightarrow{p} A$. We now show that $(X_m, A_m) | \mathcal{F}_m \xrightarrow{d} (X, A)$. To this end, note that
$$(X_m, A) | \mathcal{F}_m \xrightarrow{d} (X, A), \qquad (0, A_m) | \mathcal{F}_m \xrightarrow{p} (0, A).$$

Because $\|(X_m, A_m) - (X_m, A)\| = \|(0, A_m - A)\| = \|A_m - A\|$ and $A_m | \mathcal{F}_m \xrightarrow{p} A$, we have by Slutsky's theorem $(X_m, A_m) | \mathcal{F}_m \xrightarrow{d} (X, A)$. The result follows by an application of the continuous mapping theorem with the function $(x, A) \to Ax$.

$\square$

## B.2 Model assumptions

The following sets of assumptions are only used to prove the large-data limit results of Proposition 3.1, Proposition 3.2, and Proposition 3.3. We suppose the data is $\mathbf{Y}_m = \{Y_i\}_{i=1}^m$ drawn i.i.d. from distribution $L(y; x_0) dy$, where $L(y; x)$ defines a statistical model parametrized by $x \in \mathcal{X}$ where $\mathcal{X}$ is an open subset of $\mathbb{R}^d$. We denote the log-likelihood function $\ell$ and $\ell_m$ for the model and data respectively as,
$$\ell(y; x) = \log L(y; x), \quad \ell_m(x) = \sum_{i=1}^m \ell(Y_i; x)$$

We denote $x_{\mathrm{MLE},m}$ to be the maximum likelihood estimator
$$x_{\mathrm{MLE},m} \in \arg\max_x \ell_m(x).$$

We will use $\ell'$ and $\ell''$ to denote the gradient and Hessian of $\ell$ with respect to $x$, and use $I(x)$ to denote the Fisher information matrix,

$$I(x) = -\mathbb{E}[\ell''(Y;x)] = -\int \ell''(y;x)L(y;x)\mathrm{d}y,$$

and $I_m(x)$ for the observed information,

$$I_m(x) = -\ell''_m(x) = -\sum_{i=1}^{m} \ell''(Y_i;x).$$

We will always use a subscript $m$ to indicate that the quantity is dependent on the data.

Given a prior $\pi_0$ distribution over $\mathcal{X}$, we define the posterior, $\pi_{1,m}$ with density conditional on $\mathbf{Y}_m$,

$$\pi_{1,m}(x) \propto \pi_0(x) \prod_{i=1}^{m} L(Y_i;x) = \pi_0(x)\exp(\ell_m(x)).$$

We will use $\pi_{\beta,m}$ to denote the power posterior

$$\pi_{\beta,m} \propto \pi_0(x)\exp(\beta\ell_m(x)).$$

For the remainder of this section we will assume the following regularity conditions.

**Assumption B.3.**

1. *Euclidean state space: $\mathcal{X} \subset \mathbb{R}^d$ has an open set containing $x_0$.*

2. *Continuity of prior density: The prior density $\pi_0$ is continuous and positive in a neighbourhood of $x_0$.*

3. *Regularity of log-likelihood: There is $K > 0$, such that for $\|x - x_0\| \leq K$, $\ell(y;x)$ continuously 3 times differentiable, and there is a $M(y)$ such that,*

$$|\ell'''(y;x)| \leq M(y), \quad \int M(y)L(y;x_0)\mathrm{d}y < \infty.$$

4. *Score at the MLE: For all $m$, $x_{\mathrm{MLE},m}$ exists, is unique, and $\ell'_m(x_{\mathrm{MLE},m}) = 0$ almost surely.*

5. *Strong consistency of MLE: $x_{\mathrm{MLE},m} \xrightarrow{a.s.} x_0$ as $m \to \infty$.*

6. *Fisher information: $I(x)$ is positive definite and continuous on a neighbourhood of $x_0$.*

7. *PT-Suitable: For all $m$, both $\pi_0$ and $\mathcal{Q}$ are almost surely PT-suitable for the posterior $\pi_{1,m}$ (see Assumption A.1).*

8. *Efficient local exploration: For all $m$, the PT chain with target $\pi_{1,m}$ and references in $\{\pi_0\} \cup \mathcal{Q}$, efficiently explore locally almost surely (see Assumption A.2).*

9. *Bernstein-von Mises: For $0 < \beta \leq 1$ and $X_{\beta,m} \sim \pi_{\beta,m}$ and $\mathbf{Y}_m \overset{iid}{\sim} L(\cdot;x_0)$,*

$$m^{1/2}(X_{\beta,m} - x_{\mathrm{MLE},m})|\mathbf{Y}_m \xrightarrow{d} Z,$$

*where $Z = N(0,\beta^{-1}I(x_0)^{-1})$.*

Note that Assumption B.3.9 at $\beta = 1$ can be satisfied by introducing appropriate regularity conditions. See the paper "Asymptotic Normality, Concentration, and Coverage of Generalized Posteriors" by Miller (2021) for some possible conditions. For $0 < \beta < 1$, the result for the power posterior holds by noting that the tempered log-likelihood is $\beta \cdot \ell$ and by invoking Bernstein-von Mises results on the tempered log-likelihood under model misspecification (where the true data generating mechanism is based on the non-tempered likelihood). Such results for model misspecification are also available in the mentioned paper.

## B.3 Preliminary results

We start off with an expansion of the log-likelihood, $\ell$, about the MLE, $x_{\mathrm{MLE},m}$. Define

$$Q_m(x) = -\frac{1}{2} \cdot (x - x_{\mathrm{MLE},m})^\top I_m(x_{\mathrm{MLE},m})(x - x_{\mathrm{MLE},m}).$$

We bound the difference between the log-likelihood at the MLE and the second-order term in the expansion of the log-likelihood.

**Lemma B.4.** *Suppose Assumption B.3 holds. Then,*

1. $m^{-1} I_m(x_{\mathrm{MLE},m}) \xrightarrow{a.s.} I(x_0)$ *as* $m \to \infty$.

2. *For all* $\|x - x_0\| < K/2$, *we have a.s. there is an* $\bar{m}$ *large enough such that for* $m \geq \bar{m}$

$$\ell_m(x) = \ell_m(x_{\mathrm{MLE},m}) + Q_m(x) + \epsilon_m(x)$$

*for some* $\epsilon_m(x)$ *satisfying*

$$|\epsilon_m(x)| \leq \frac{Mm}{3} \cdot \|x - x_{\mathrm{MLE},m}\|^3,$$

*and* $M = \int M(y) L(y; x_0) \mathrm{d}y < \infty$.

3. *For any sequence of random variables,* $X_m$, *such that* $m^{1/2}(X_m - x_{\mathrm{MLE},m})|\boldsymbol{Y}_m \xrightarrow{d} X$ *for some random variable* $X$, *we have*

$$\epsilon_m(X_m)|\boldsymbol{Y}_m \xrightarrow{p} 0,$$
$$\pi_0(X_m)|\boldsymbol{Y}_m \xrightarrow{p} \pi_0(x_0).$$

*Proof of Lemma B.4.*      1. By the triangle inequality,

$$\|m^{-1} I_m(x_{\mathrm{MLE},m}) - I(x_0)\| \leq \|m^{-1} I_m(x_{\mathrm{MLE},m}) - m^{-1} I_m(x_0)\| \\ + \|m^{-1} I_m(x_0) - I(x_0)\|.$$

We will now show that each term converges to 0 a.s.

Since $x_{\mathrm{MLE},m} \xrightarrow{a.s.} x_0$, we have a.s., for $m$ large enough $\|x_0 - x_{\mathrm{MLE},m}\| < K$. Therefore by the mean-value theorem,

$$\|m^{-1} I_m(x_{\mathrm{MLE},m}) - m^{-1} I_m(x_0)\| = \frac{1}{m} \sum_{i=1}^m \|\ell''(Y_i; x_{\mathrm{MLE},m}) - \ell''(Y_i; x_0)\|$$

$$\leq \frac{1}{m} \sum_{i=1}^m M(Y_i) \|x_{\mathrm{MLE},m} - x_0\|.$$

Since $x_{\mathrm{MLE},m} \xrightarrow{a.s.} x_0$, and $\frac{1}{m} \sum_{i=1}^m M(Y_i) \xrightarrow{a.s.} \int M(Y) L(y; x_0) \mathrm{d}y < \infty$, we have

$$\|m^{-1} I_m(x_{\mathrm{MLE},m}) - m^{-1} I_m(x_0)\| \xrightarrow[m \to \infty]{a.s.} 0.$$

By the strong law of large numbers we have

$$m^{-1} I_m(x_0) = \frac{1}{m} \sum_{i=1}^m \ell''(Y_i; x_0) \xrightarrow[m \to \infty]{a.s.} \int \ell''(y; x_0) L(y; x_0) \mathrm{d}y = I(x_0).$$

This implies

$$\|m^{-1} I_m(x_0) - I(x_0)\| \xrightarrow[m \to \infty]{a.s.} 0.$$

2. We use a second-order expansion around $x_{\mathrm{MLE},m}$ and Assumption B.3.4 to get,

$$\ell_m(x) = \ell_m(x_{\mathrm{MLE},m}) + (x - x_{\mathrm{MLE},m})^\top \ell'_m(x_{\mathrm{MLE},m}) + Q_m(x) + \epsilon_m(x)$$
$$= \ell_m(x_{\mathrm{MLE},m}) + Q_m(x) + \epsilon_m(x),$$

for some $\epsilon_m(x)$ satisfying,

$$|\epsilon_m(x)| \leq \frac{M_m}{6}\|x - x_{\mathrm{MLE},m}\|^3.$$

where $M_m = \sup\{|\ell'''_m(\xi)| : \|\xi - x_{\mathrm{MLE},m}\| < \|x - x_{\mathrm{MLE},m}\|\}$. We are done if we can show that $M_m \leq 2Mm$.

Suppose $\|\xi - x_{\mathrm{MLE},m}\| < \|x - x_{\mathrm{MLE},m}\|$. Then, by triangle inequality,

$$\|\xi - x_0\| \leq \|\xi - x_{\mathrm{MLE},m}\| + \|x_{\mathrm{MLE},m} - x_0\|$$
$$\leq \|x - x_{\mathrm{MLE},m}\| + \|x_{\mathrm{MLE},m} - x_0\|$$
$$\leq \|x - x_0\| + \|x_0 - x_{\mathrm{MLE},m}\| + \|x_{\mathrm{MLE},m} - x_0\|$$
$$< \frac{K}{2} + 2\|x_0 - x_{\mathrm{MLE},m}\|$$

By the strong consistency of the MLE Assumption B.3.5, we have a.s. there is an $m_0$ large enough such that for $m \geq m_0$, $\|x_{\mathrm{MLE},m} - x_0\| < K/4$ and thus $\|\xi - x_0\| < K$. This means that Assumption B.3.3 implies,

$$|\ell'''_m(\xi)| \leq \sum_{i=1}^{M} |\ell'''(Y_i; \xi)| \leq \sum_{i=1}^{m} M(Y_i).$$

Since $Y_i \sim L(y; x_0)\mathrm{d}y$, the law of large numbers implies $m^{-1}\sum_{i=1}^{m} M(Y_i)$ converges a.s. to $M$ as $m \to \infty$. Almost surely, there is an $\bar{m} \geq m_0$ such that for $m \geq \bar{m}$,

$$\frac{1}{m}\sum_{i=1}^{m} M(Y_i) < 2M < \infty.$$

Therefore, $M_m \leq 2mM$, which completes the proof.

3. Note that $\epsilon_m(X_m)$ satisfies almost surely for $m$ large enough,

$$|\epsilon_m(X_m)| \leq \frac{M}{3m^{1/2}}\|m^{1/2}(X_m - x_{\mathrm{MLE},m})\|^3.$$

By the conditional continuous mapping theorem, Lemma B.2.6,

$$\|m^{1/2}(X_m - x_{\mathrm{MLE},m})\|^3 | \mathbf{Y}_m \xrightarrow{d} \|X\|^3,$$

and $\epsilon_m(X_m)|\mathbf{Y}_m \xrightarrow{d} 0$ and hence also $\epsilon_m(X_m)|\mathbf{Y}_m \xrightarrow{p} 0$ and $\epsilon_m(X_m) \xrightarrow{p} 0$ by Lemma B.2.4.

Also note that $m^{1/2}(X_m - x_{\mathrm{MLE},m})|\mathbf{Y}_m \xrightarrow{d} X$ implies $X_m - x_{\mathrm{MLE},m}|\mathbf{Y}_m \xrightarrow{p} 0$, and hence by the continuous mapping theorem, $\pi_0(X_m)|\mathbf{Y}_m \xrightarrow{p} \pi_0(x_0)$.

$\square$

Next, we claim that the second-order term in the expansion of the log-likelihood around the MLE evaluated at an appropriate random variable converges to a transformed chi-squared random variable. This result will be used repeatedly in the proofs of Proposition 3.1, Proposition 3.2, and Proposition 3.3.

**Lemma B.5.** *Suppose Assumption B.3 holds. If $X_{\beta,m}, X'_{\beta,m} \sim \pi_{\beta,m}$, are independent conditioned on $\mathbf{Y}_m$, then for all $0 < \beta \leq 1$ we have as $m \to \infty$:*

1. $\epsilon_m(X_{\beta,m})|\mathbf{Y}_m \xrightarrow{p} 0$.

2. $Q_m(X_{\beta,m})|\mathbf{Y}_m \xrightarrow{d} -Q/(2\beta)$, *where* $Q \sim \chi^2_d$.

3. $Q_m(X_{\beta,m}) - Q_m(X'_{\beta,m})|\boldsymbol{Y}_m \xrightarrow{d} \frac{1}{2\beta}(Q - Q')$, where $Q, Q' \overset{iid}{\sim} \chi_d^2$.

4. $\ell_m(X_{\beta,m}) - \ell_m(X'_{\beta,m})|\boldsymbol{Y}_m \xrightarrow{d} \frac{1}{2\beta}(Q - Q')$,, where $Q, Q' \overset{iid}{\sim} \chi_d^2$.

*Proof of Lemma B.5.*      1. This follows immediately Assumption B.3.9. and Lemma B.4.3.

     2. Note that we can decompose $Q_m$ as

$$Q_m(X_{\beta,m}) = -\frac{1}{2}[m^{1/2}(X_{\beta,m} - x_{\mathrm{MLE},m})]^\top \left[\frac{1}{m}I_m(x_{\mathrm{MLE},m})\right][m^{1/2}(X_{\beta,m} - x_{\mathrm{MLE},m})].$$

By Assumption B.3.9 we have,

$$m^{1/2}(X_{\beta,m} - x_{\mathrm{MLE},m})|\boldsymbol{Y}_m \xrightarrow[m\to\infty]{d} N(0, \beta^{-1}I^{-1}(x_0)),$$

and $m^{-1}I_m(x_{\mathrm{MLE},m}) \xrightarrow{a.s.} I(x_0)$ by Assumption B.3.6. By Lemma B.2.7 and B.2.6 we get,

$$Q_m(X_{\beta,m})|\boldsymbol{Y}_m \xrightarrow[m\to\infty]{d} -Q/(2\beta),$$

where $Q \sim \chi_d^2$.

     3. Note that for any $0 < \beta \leq 1$, by using the arguments as above,

$$Q_m(X_{\beta,m})|\boldsymbol{Y}_m \xrightarrow{d} -Q/(2\beta)$$
$$Q_m(X'_{\beta,m})|\boldsymbol{Y}_m \xrightarrow{d} -Q/(2\beta),$$

where $Q \sim \chi_d^2$. Each of the $X_{\beta,m}, X'_{\beta,m}$ are assumed to be conditionally independent given the data $\boldsymbol{Y}_m$, and therefore

$$Q_m(X_{\beta,m}) - Q_m(X'_{\beta,m})|\boldsymbol{Y}_m \xrightarrow[m\to\infty]{d} (Q - Q')/(2\beta),$$

where $Q, Q' \sim \chi_d^2$ are independent.

     4. Finally, we employ Lemma B.4.2 and triangle inequality to get,

$$\left|[\ell_m(X_{\beta,m}) - \ell_m(X'_{\beta,m})] - [Q_m(X_{\beta,m}) - Q_m(X'_{\beta,m})]\right| \leq |\epsilon_m(X_{\beta,m})| + |\epsilon_m(X'_{\beta,m})|$$

By part Lemma B.5.1, we have $\epsilon_m(X_{\beta,m})|\boldsymbol{Y}_m \xrightarrow{p} 0$ and $\epsilon_m(X'_{\beta,m})|\boldsymbol{Y}_m \xrightarrow{p} 0$ so

$$\left[\ell_m(X_{\beta,m}) - \ell_m(X'_{\beta,m})\right] - \left[Q_m(X_{\beta,m}) - Q_m(X'_{\beta,m})\right]|\boldsymbol{Y}_m \xrightarrow{p} 0.$$

By Lemma B.5.3 and the conditional Slutsky's theorem Lemma B.2.7,

$$\ell_m(X_{\beta,m}) - \ell_m(X'_{\beta,m})|\boldsymbol{Y}_m \xrightarrow[m\to\infty]{d} (Q - Q')/(2\beta).$$

$\square$

## B.4   Proof of Proposition 3.1

*Proof of Proposition 3.1.* The asymptotic restart rate, $\tau_m$, is related to the GCB by

$$\tau_m = \frac{1}{2 + 2\Lambda(\pi_0, \pi_{1,m})},$$

where $\Lambda(\pi_0, \pi_{1,m})$ is the GCB between the prior $\pi_0$ and the posterior $\pi_{1,m}$. This implies that

$$0 \leq \limsup_{m\to\infty} \tau_m \leq \frac{1}{2 + 2\liminf_{m\to\infty} \Lambda(\pi_0, \pi_{1,m})}.$$

Therefore, we are are done if we can show that $\liminf_{m\to\infty} \Lambda(\pi_0, \pi_{1,m}) = \infty$ almost surely.

Suppose $X_{\beta,m}, X'_{\beta,m} \sim \pi_{\beta,m}$ are independent conditioned on $\mathbf{Y}_m$. Then,

$$\Lambda(\pi_0, \pi_{1,m}) = \frac{1}{2} \int_0^1 \mathbb{E}[|\ell_m(X_{\beta,m}) - \ell(X'_{\beta,m})|\,|\mathbf{Y}_m]\mathrm{d}\beta.$$

Since the integrand is positive, for any $\delta > 0$,

$$\Lambda(\pi_0, \pi_{1,m}) \geq \frac{1}{2} \int_\delta^1 \mathbb{E}[|\ell_m(X_{\beta,m}) - \ell(X'_{\beta,m})|\,|\mathbf{Y}_m]\mathrm{d}\beta$$

By taking the limit infimum of both sides, and using Fatou's lemma,

$$\liminf_{m\to\infty} \Lambda(\pi_0, \pi_{1,m}) \geq \frac{1}{2} \int_\delta^1 \liminf_{m\to\infty} \mathbb{E}[|\ell_m(X_{\beta,m}) - \ell(X'_{\beta,m})|\,|\mathbf{Y}_m]\mathrm{d}\beta.$$

From Lemma B.5.4 and the conditional continuous mapping theorem Lemma B.2.6 applied to $x \to |x|$, we have

$$|\ell_m(X_{\beta,m}) - \ell(X'_{\beta,m})|\,|\mathbf{Y}_m \xrightarrow[m\to\infty]{d} \frac{|Q - Q'|}{2\beta}, \quad Q, Q' \sim \chi^2_d.$$

By Lemma B.2.5, almost surely we have

$$\liminf_{m\to\infty} \Lambda(\pi_0, \pi_{1,m}) \geq \frac{1}{2} \int_\delta^1 \frac{1}{2\beta}\mathbb{E}\left[|Q - Q'|\right]\,d\beta = -\frac{1}{4}\mathbb{E}\left[|Q - Q'|\right]\log(\delta).$$

Since this is true for all $\delta > 0$, and since the right hand side increases to infinity at $\delta \to 0$, we have almost surely,

$$\liminf_{m\to\infty} \Lambda(\pi_0, \pi_{1,m}) \geq \lim_{\delta\to 0} -\frac{1}{4}\mathbb{E}\left[|Q - Q'|\right]\log(\delta) = \infty.$$

$\square$

## B.5 Proof of Proposition 3.2

**Lemma B.6.** *Assume $\pi_0$ is a PT-suitable reference for $\pi_1$, with log-likelihood $\ell$.*

1. *If $L_\beta = \ell(X_\beta)$, for $X_\beta \sim \pi_\beta$, then $L_\beta$ is stochastically non-decreasing in $\beta$, i.e. for all $y \in \mathbb{R}$, $\mathbb{P}(L_\beta > y)$ is a non-decreasing function of $\beta$.*

2. *Let $r(\beta, \beta')$ be a mean rejection rate for a swap between $\pi_\beta, \pi_{\beta'}$,*

$$r(\beta, \beta') = 1 - \mathbb{E}\left[1 \wedge \frac{\pi_\beta(X_{\beta'})\pi_{\beta'}(X_\beta)}{\pi_\beta(X_\beta)\pi_{\beta'}(X_{\beta'})}\right], \quad (X_\beta, X_{\beta'}) \sim \pi_\beta \times \pi_{\beta'}.$$

*If $[a, b] \subset [a', b'] \subset [0, 1]$, then $r(a, b) \leq r(a', b')$.*

*Proof of Lemma B.6.*    1. Fix $y \in \mathbb{R}$, and $0 \leq \beta < \beta' \leq 1$. We want to show that

$$0 \leq \mathbb{P}[L_{\beta'} > y] - \mathbb{P}[L_\beta > y] = \mathbb{E}[f(L_\beta, L_{\beta'})],$$

where $f(l, l') = 1(l' > y) - 1(l > y)$. We have

$$\mathbb{E}[f(L_\beta, L_{\beta'})] = \int f(\ell(x), \ell(x'))\frac{1}{Z(\beta)}\exp(\beta\ell(x))\frac{1}{Z(\beta')}\exp(\beta'\ell(x'))\mathrm{d}x\,\mathrm{d}x'$$

$$= \frac{Z(\beta)}{Z(\beta')}\mathbb{E}[f(L, L')\exp(\delta L')],$$

where $L, L' \overset{d}{=} \ell(X_\beta)$ are independent and $\delta = \beta' - \beta > 0$. Now, notice that

$$f(l, l') = 1(l' > y) - 1(l > y)$$
$$= 1(l' > y)1(l \leq y) - 1(l > y)1(l' \leq y).$$

Therefore, we have

$$\begin{aligned}
\mathbb{E}[f(L, L') \exp(\delta L')] &= \mathbb{E}[1(L' > y)1(L \le y) \exp(\delta L')] \\
&\quad - \mathbb{E}[1(L > y)1(L' \le y) \exp(\delta L')] \\
&= \mathbb{E}[1(L' > y)1(L \le y)(\exp(\delta L') - \exp(\delta L))] \\
&\ge 0,
\end{aligned}$$

where the second to last line used the fact that $L, L'$ are i.i.d.

2. To simplify notation, suppose first $a = a'$. Denote the cumulative distribution function of $L_\beta$ by $F_\beta$. From (a), we have that $F_b \ge F_{b'}$. It follows that we can construct a random variable $L'$ which is equal in distribution to $L_{b'}$ and such that $L_b(\omega) \le L'(\omega)$ for all outcomes $\omega$ in the probability space. This is achieved by setting $L' = F_{L_{b'}}^{-1} \circ F_{L_b} \circ L_b$, where $F^{-1}$ denotes the generalized inverse cumulative function. To see why, note that $F_{L_b} \circ L_b$ is uniformly distributed, being a probability integral transform. Hence $F_{L_{b'}}^{-1}$ applied to that uniform yields a $L_{b'}$-distributed random variable. The inequality $L_b(\omega) \le L'(\omega)$ follows from $F_b \ge F_{b'}$.

Next, define $f(\delta) = 1 - 1 \wedge \exp(-\delta)$, which is an increasing function in $\delta$. Hence, $f((b-a)(L_b - L_a)) \le f((b-a)(L' - L_a))$ for all outcomes, and so

$$r(a, b) = \mathbb{E}[f((b-a)(L_b - L_a))] \le \mathbb{E}[f((b-a)(L' - L_a))] = r(a, b'),$$

where in the last equality we also used that $L'$ is independent of $L_a$, being a deterministic transformation of the random variable $L_b$. Finally, if $a < a'$, use $r(a, b) \le r(a, b') \le r(a', b')$ where the last inequality is obtained using a very similar argument as above.

$\square$

**Lemma B.7.** *Suppose $\mathcal{Q}$ is almost surely a PT-suitable reference family for all targets $\pi_{1,m}$. Also assume that for all $m$, the PT chain with target $\pi_{1,m}$ and references $\mathcal{Q}$, efficiently explore locally almost surely. Given a (random) sequence $q_m \in \mathcal{Q}$, if $\alpha_m$ is the average acceptance probability between $q_m$ and $\pi_{1,m}$, then if $\alpha_m \xrightarrow{p} 1$, then for any $\mathcal{B}_N$, we have $\tau_m(\mathcal{B}_N) \xrightarrow{p} \frac{1}{2}$ as $m \to \infty$.*

*Proof of Lemma B.7.* For any schedule $\mathcal{B}_n = (\beta_n)_{n=0}^N$, let $\{r_{n,m}\}_{n=0}^{N-1}$ be the average rejection rates between components $n$ and $n+1$. For any $n$, we have by Lemma B.6,

$$0 \le r_{n,m} \le 1 - \alpha_m.$$

Since $\alpha_m \xrightarrow{p} 1$, we have $r_{n,m} \xrightarrow{p} 0$ as $m \to \infty$ and

$$\tau_m(\mathcal{B}_N) = \frac{1}{2 + 2\sum_{n=0}^{N-1} \frac{r_{n,m}}{1 - r_{n,m}}} \xrightarrow[m \to \infty]{p} 1/2.$$

$\square$

**Lemma B.8.** *Suppose Assumption B.3 holds and $X_{0,m} \sim N(x_{\mathrm{MLE},m}, I_m(x_{\mathrm{MLE},m})^{-1})$. Then, as $m \to \infty$,*

$$m^{1/2}(X_{0,m} - x_{\mathrm{MLE},m})|\mathbf{Y}_m \xrightarrow{d} \mathcal{N}(0, I^{-1}(x_0)).$$

*Proof of Lemma B.8.* Since $X_{0,m} \sim N(x_{\mathrm{MLE},m}, I_m(x_{\mathrm{MLE},m})^{-1})$,

$$m^{1/2} \cdot \frac{I_m^{1/2}(x_{\mathrm{MLE},m})}{m^{1/2}} \cdot I^{-1/2}(x_0)(X_{0,m} - x_{\mathrm{MLE},m})|\mathbf{Y}_m \sim \mathcal{N}(0, I^{-1}(x_0)).$$

By Lemma B.4.1 and Assumption B.3.5, we have

$$\frac{I_m^{1/2}(x_{\mathrm{MLE},m})}{m^{1/2}} \cdot I^{-1/2}(x_0) \xrightarrow[m \to \infty]{a.s.} \mathbb{I}_d.$$

By Lemma B.2.7 it follows,

$$m^{1/2}(X_{0,m} - x_{\mathrm{MLE},m})|\mathbf{Y}_m \xrightarrow{d} \mathcal{N}(0, I^{-1}(x_0)).$$

$\square$

*Proof of Proposition 3.2.* Let $q_m = N(x_{\text{MLE},m}, I_m^{-1}(x_{\text{MLE},m})) \in \mathcal{Q}$. The acceptance probability $\alpha_m$ is

$$\alpha_m = \mathbb{E}\left[1 \wedge A_m(X_{0,m}, X_{1,m}) | \mathbf{Y}_m\right],$$

where $A_m(X_{0,m}, X_{1,m})$ is the acceptance ratio,

$$A_m(X_{0,m}, X_{1,m}) = \frac{q_m(X_{1,m}) \cdot \pi_{1,m}(X_{0,m})}{q_m(X_{0,m}) \cdot \pi_{1,m}(X_{1,m})}$$
$$= \frac{q_m(X_{1,m})}{q_m(X_{0,m})} \cdot \frac{\pi_0(X_{0,m})}{\pi_0(X_{1,m})} \cdot \frac{\exp(\ell_m(X_{0,m}))}{\exp(\ell_m(X_{1,m}))}.$$

Note that up to an additive constant $\log q_m(x) = Q_m(x)$, and so

$$\log q_m(X_{1,m}) - \log q_m(X_{0,m}) = Q_m(X_{1,m}) - Q_m(X_{0,m}).$$

Therefore the log-acceptance ratio satisfies

$$\log A_m(X_{0,m}, X_{1,m}) = \log \pi_0(X_{0,m}) - \log \pi_0(X_{1,m}) + \epsilon_m(X_{0,m}) - \epsilon_m(X_{1,m}),$$

where $\epsilon_m$ was defined in Lemma B.4.

Since we have the asymptotic normality of $X_{0,m}$ and $X_{1,m}$ conditioned on $\mathbf{Y}_m$ by Lemma B.5.9, and Lemma B.8, we can invoke Lemma B.4.3,

$$\epsilon_m(X_{0,m}), \epsilon_m(X_{1,m}) | \mathbf{Y}_m \xrightarrow[m \to \infty]{p} 0, \tag{6}$$
$$\log \pi_0(X_{0,m}), \log \pi_0(X_{1,m}) | \mathbf{Y}_m \xrightarrow[m \to \infty]{p} \log \pi_0(x_0).$$

Combining (6) we have the acceptance ratio satisfies,

$$A_m(X_{0,m}, X_{1,m}) | \mathbf{Y}_m \xrightarrow{p} 1.$$

To conclude, note that $1 \wedge A_m(X_{0,m}, X_{1,m}) \leq 1$, so by the dominated convergence theorem,

$$\alpha_m = \mathbb{E}[1 \wedge A_m(X_{0,m}, X_{1,m}) | \mathbf{Y}_m] \xrightarrow[m \to \infty]{a.s.} 1,$$

and hence $\alpha_m \xrightarrow{a.s.} 1$. The result follows from Lemma B.7. $\square$

### B.6 Proof of Proposition 3.3

Suppose $\mu_m$ and $\Sigma_m$ are posterior mean and variance conditional to $\mathbf{Y}_m$,

$$\mu_m = \mathbb{E}[X_{1,m} | \mathbf{Y}_m], \quad \Sigma_m = \text{Var}[X_{1,m} | \mathbf{Y}_m],$$

where $X_{1,m} \sim \pi_{1,m}$. We introduce a final set of assumptions that are required for the proof of Proposition 3.3.

**Assumption B.9.** *1. Posterior mean and MLE: As $m \to \infty$, $m^{1/2}(\mu_m - x_{\text{MLE},m}) \xrightarrow{a.s.} 0$.*

*2. Posterior variance and Fisher information: For all $m$, $\Sigma_m$ is almost surely positive definite and $m\Sigma_m \xrightarrow{a.s.} I^{-1}(x_0)$ as $m \to \infty$.*

For such results in the univariate case with convergence in probability, see [35] and [18].

Given $q'_m = N(\mu_m, \Sigma_m)$, define,

$$Q'_m(x) = -\frac{1}{2}(x - \mu_m)^\top \Sigma_m^{-1}(x - \mu_m).$$

**Lemma B.10.** *Suppose Assumption B.3 and Assumption B.9 hold. If $X'_{0,m} \sim q'_m$ and $X_{1,m} \sim \pi_{1,m}$, then as $m \to \infty$,*

*1. $m^{1/2}(X'_{0,m} - x_{\text{MLE},m}) | \mathbf{Y}_m \xrightarrow{d} \mathcal{N}(0, I^{-1}(x_0))$,*

2. $Q'_m(X_{1,m}) - Q_m(X_{1,m}) \xrightarrow{P} 0$,

3. $Q'_m(X'_{0,m}) - Q_m(X'_{0,m}) \xrightarrow{P} 0$.

*Proof of Lemma B.10.*      1. Note that since $X'_{0,m} \sim \mathcal{N}(\mu_m, \Sigma_m)$,

$$\frac{\Sigma_m^{-1/2}}{m^{1/2}} \cdot I^{-1/2}(x_0) \cdot m^{1/2}(X_{0,m} - \mu_m) | \mathbf{Y}_m \sim \mathcal{N}(0, I^{-1}(x_0)).$$

By Assumption B.9,

$$\frac{\Sigma_m^{-1/2}}{m^{1/2}} \cdot I^{-1/2}(x_0) \xrightarrow[m\to\infty]{a.s.} \mathbb{I}_d.$$

Therefore using Slutsky's theorem, Lemma B.2.7, it follows that

$$m^{1/2}(X'_{0,m} - \mu_m) | \mathbf{Y}_m \xrightarrow[m\to\infty]{d} \mathcal{N}(0, I^{-1}(x_0)).$$

Finally, the result follows from Slutsky's theorem using Assumption B.9.1.

2. Using the definition of $Q(x)$ and $Q'(x)$ we obtain the following decomposition,

$$Q'_m(x) - Q_m(x) = \epsilon_m^0 + \epsilon_m^1(x) + \epsilon_m^2(x)$$

where,

$$\epsilon_m^0 = -(\mu_m - x_{\mathrm{MLE},m})^\top \Sigma_m^{-1}(\mu_m - x_{\mathrm{MLE},m})$$
$$\epsilon_m^1(x) = -2(x - x_{\mathrm{MLE},m})^\top \Sigma_m^{-1}(x_{\mathrm{MLE},m} - \mu_m)$$
$$\epsilon_m^2(x) = -[m^{1/2}(x - x_{\mathrm{MLE},m})]^\top \cdot [m^{-1}(\Sigma_m^{-1} - I_m(x_{\mathrm{MLE},m}))] \cdot [m^{1/2}(x - x_{\mathrm{MLE},m})]$$

Now, using Assumption B.9, it follows that $\epsilon_m^0 \xrightarrow{P} 0$ as $m \to \infty$,

$$\epsilon_m^0 = -[m^{1/2}(\mu_m - x_{\mathrm{MLE},m})]^\top [m\Sigma_m]^{-1}[m^{1/2}(\mu_m - x_{\mathrm{MLE},m})]$$
$$= o_p(1) \cdot O_p(1) \cdot o_p(1)$$
$$= o_p(1).$$

Therefore we are done if we can show (1) $\epsilon_m^1(X_{1,m}) \xrightarrow{P} 0$ and (2) $\epsilon_m^2(X_{1,m}) \xrightarrow{P} 0$ as $m \to \infty$.

(1) follows from Assumption B.9 and Lemma B.5.9 since as $m \to \infty$,

$$\epsilon_m^1(X_{1,m}) = -2[m^{1/2}(X_{1,m} - x_{\mathrm{MLE},m})]^\top [m\Sigma_m]^{-1}[m^{1/2}(x_{\mathrm{MLE},m} - \mu_m)]$$
$$= O_p(1) \cdot O_p(1) \cdot o_p(1)$$
$$= o_p(1).$$

Finally for (2) notice that by Assumption B.9, and $m^{-1}(\Sigma_m^{-1} - I_m(x_{\mathrm{MLE},m})) \xrightarrow{P} 0$. Using Lemma B.5.9 as $m \to \infty$,

$$\epsilon_m^2(X_{1,m})$$
$$= -[m^{1/2}(X_{1,m} - x_{\mathrm{MLE},m})]^\top [m^{-1}(\Sigma_m^{-1} - I_m(x_{\mathrm{MLE},m}))][m^{1/2}(X_{1,m} - x_{\mathrm{MLE},m})]$$
$$= O_p(1) \cdot o_p(1) \cdot O_p(1)$$
$$= o_p(1)$$

3. Similar to the proof of Lemma B.10.2, we are done if we can show (3) $\epsilon_m^1(X'_{0,m}) \xrightarrow{P} 0$ and (4) $\epsilon_m^2(X'_{0,m}) \xrightarrow{P} 0$ as $m \to \infty$.

To show (3) we use Lemma B.10.1 and Assumption B.9,

$$\epsilon_m^1(X'_{0,m}) = -2[m^{1/2}(X'_{0,m} - x_{\mathrm{MLE},m})]^\top [m\Sigma_m]^{-1}[m^{1/2}(x_{\mathrm{MLE},m} - \mu_m)]$$
$$= O_p(1) \cdot O_p(1) \cdot o_p(1)$$
$$= o_p(1).$$

Finally, to obtain (4) follows from Lemma B.10.1 and $m^{-1}(\Sigma_m^{-1} - I_m(x_{\mathrm{MLE},m})) \xrightarrow{p} 0$,

$$
\begin{aligned}
\epsilon_m^2(X'_{0,m}) &= [m^{1/2}(X'_{0,m} - x_{\mathrm{MLE},m})]^\top [m^{-1}(\Sigma_m^{-1} - I_m)][m^{1/2}(X'_{0,m} - x_{\mathrm{MLE},m})] \\
&= O_p(1) \cdot o_p(1) \cdot O_p(1) \\
&= o_p(1).
\end{aligned}
$$

$\square$

*Proof of Proposition 3.3.* Suppose $X_{1,m} \sim \pi_{1,m}$ and $X'_{0,m} \sim q'_m = N(\mu_m, \Sigma_m) \in \mathcal{Q}$ are independent conditioned on $\mathbf{Y}_m$. Let $\alpha'_m = \mathbb{E}[1 \wedge A'_m(X'_{0,m}, X_{1,m})]$ be the average acceptance probability, where $A'_m(X'_{0,m}, X_{1,m})$ is the acceptance ratio,

$$
\begin{aligned}
A'_m(X'_{0,m}, X_{1,m}) &= \frac{q'_m(X_{1,m}) \cdot \pi_{1,m}(X'_{0,m})}{q'_m(X'_{0,m}) \cdot \pi_{1,m}(X_{1,m})} \\
&= \frac{q'_m(X_{1,m})}{q'_m(X'_{0,m})} \cdot \frac{\pi_0(X'_{0,m})}{\pi_0(X_{1,m})} \cdot \frac{\exp(\ell_m(X'_{0,m}))}{\exp(\ell_m(X_{1,m}))}.
\end{aligned}
$$

Note that up to an additive constant $\log q'_m(x) = Q'_m(x)$, and by Lemma B.10, we have as $m \to \infty$,

$$
\begin{aligned}
\log q'_m(X_{1,m}) - \log q'_m(X'_{0,m}) &= Q'_m(X_{1,m}) - Q'_m(X_{0,m}), \\
&= Q_m(X_{1,m}) - Q_m(X_{0,m}) + o_p(1).
\end{aligned}
$$

Therefore we have the log-acceptance ratio satisfies,

$$
\log A'_m(X'_{0,m}, X_{1,m}) = \log \pi_0(X'_{0,m}) - \log \pi_0(X_{1,m}) + \epsilon_m(X'_{0,m}) - \epsilon_m(X_{1,m}) + o_p(1).
$$

where $\epsilon_m$ was defined in Lemma B.4. By Lemma B.5.9, and Lemma B.10.1 we have $m^{1/2}(X_{1,m} - x_{\mathrm{MLE},m})$ and $m^{1/2}(X'_{0,m} - x_{\mathrm{MLE},m})$ are asymptotically normal conditioned on $\mathbf{Y}_m$. Therefore by Lemma B.4.3,

$$
\epsilon_m(X'_{0,m}), \epsilon_m(X_{1,m}) \xrightarrow[m\to\infty]{p} 0
$$

$$
\log \pi_0(X'_{0,m}), \log \pi_0(X_{1,m}) \xrightarrow[m\to\infty]{p} \log \pi_0(x_0),
$$

and the acceptance ratio $A'_m(X'_{0,m}, X_{1,m}) \xrightarrow{p} 1$.

To conclude, note that $1 \wedge A'_m(X'_{0,m}, X_{1,m}) \le 1$, so by dominated convergence theorem,

$$
\lim_{m\to\infty} \mathbb{E}[\alpha_m] = \lim_{m\to\infty} \mathbb{E}[1 \wedge A_m(X_{0,m}, X_{1,m})] = 1,
$$

and hence $\alpha'_m \xrightarrow{p} 1$. $\square$

# C  Proof of Theorem 3.4

*Proof of Theorem 3.4.* Since $\eta$ is bounded there is a $K > 0$ such that $\eta(x) = (\eta_1(x), \ldots, \eta_d(x))$ where $\eta_i : \mathcal{X} \to [-K, K]$. Fix $0 < \epsilon < \frac{1}{2}$. Suppose $(\mathbf{X}_{t,r})_{t=1}^{T_r}$ are the draws from the chain parameterized by $\hat{\phi}_r$ at round $r$. Define for each $i = 1, \ldots, d$,

$$A_{r,i} = \left\{ \left| \frac{1}{T_r} \sum_{t=1}^{T_r} \eta_i(X_{t,r}^N) - \mathbb{E}_1[\eta_i] \right| > \delta_r \right\},$$

where $\delta_r$ is a sequence to be determined. By assumption, we have $|\eta_i(x)| \le K$ for some $K > 0$. We can therefore apply Theorem 1 of [11] (Hoeffding's inequality for Markov chains) to get that

$$P\left(A_{r,i}|\hat{\phi}_r\right) \le 2 \exp\left( -\frac{\mathrm{Gap}(\hat{\phi}_r)}{2 - \mathrm{Gap}(\hat{\phi}_r)} \cdot \frac{\delta_r^2 T_r}{2K^2} \right)$$

$$\le 2 \exp\left( -\frac{\kappa}{2 - \kappa} \cdot \frac{\delta_r^2 T_r}{2K^2} \right).$$

The last inequality used the assumption that $\mathrm{Gap}(\phi)$ is bounded below by $\kappa$ on $\Phi$. By taking expectations over $\hat{\phi}_r$ and by setting $\delta_r = d^{-1} T_r^{-1/2+\epsilon}$, we obtain

$$P(A_{r,i}) \le 2 \exp\left( -\frac{\kappa}{2 - \kappa} \cdot \frac{T_r^\epsilon}{2d^2 K^2} \right).$$

Since $T_r = \Omega(2^r)$, we have by the ratio test,

$$\sum_{r=1}^{\infty} P(A_{r,i}) < \infty.$$

From the Borel-Cantelli lemma it follows that for each $i$, $P(A_{r,i} \text{ i.o.}) = 0$, and thus a.s. there exists $R_i(\epsilon)$ and such that for all $r \ge R_i(\epsilon)$,

$$\left\| \frac{1}{T_r} \sum_{t=1}^{T_r} \eta_i(X_{t,r}^N) - \mathbb{E}_1[\eta_i] \right\| \le \frac{1}{d} T_r^{-\frac{1}{2}+\epsilon}.$$

Since $\hat{\phi}_{r+1}$ is chosen to satisfy,

$$\mathbb{E}_{\hat{\phi}_{r+1}}[\eta] = \frac{1}{T_r} \sum_{t=1}^{T_r} \eta(X_{t,r}^N),$$

by the triangle inequality, we have for all $r > \max\{R_1(\epsilon), \ldots, R_d(\epsilon)\} = R(\epsilon)$,

$$\|\mathbb{E}_{\hat{\phi}_{r+1}}[\eta] - \mathbb{E}_1[\eta]\| \le T_r^{-\frac{1}{2}+\epsilon}$$

In particular, since $\mathbb{E}_1[\eta] = \mathbb{E}_{\phi_{\mathrm{KL}}}[\eta]$, we have a.s.

$$\lim_{r \to \infty} \mathbb{E}_{\hat{\phi}_r}[\eta] = \mathbb{E}_{\phi_{\mathrm{KL}}}[\eta].$$

Since $\mathcal{Q}$ is an exponential family of full-rank, convergence in the mean of the sufficient statistic is equivalent to the convergence of the the natural parameters. That is,

$$\hat{\phi}_r \xrightarrow[r \to \infty]{a.s.} \phi_{\mathrm{KL}}.$$

$\square$

## D  Upper bounds on the GCB

### D.1  Proof of Theorem 3.5

Suppose $X_{\phi,\beta}, X'_{\phi,\beta} \sim \pi_{\phi,\beta}$ are indepednent. By Jensen's inequality and Result 4 in [8],

$$
\begin{aligned}
\Lambda(q_\phi, \pi_1) &= \frac{1}{2} \int_0^1 \mathbb{E}[|\ell_\phi(X_{\phi,\beta}) - \ell_\phi(X_{\phi,\beta})|] \, d\beta \\
&\le \frac{1}{2} \left( \int_0^1 \mathbb{E}[(\ell_\phi(X_{\phi,\beta}) - \ell_\phi(X'_{\phi,\beta}))^2] \, d\beta \right)^{1/2} \\
&= \sqrt{\frac{1}{2} \mathrm{SKL}(q_\phi, \pi_1)}.
\end{aligned}
\tag{7}
$$

Also, because $\phi_{\mathrm{KL}}$ minimizes the forward KL divergence, it follows that

$$
\mathbb{E}_{\phi_{\mathrm{KL}}}[\eta] = \mathbb{E}_{\pi_1}[\eta].
$$

In particular, by taking a dot product with $\phi$ for any $\phi \in \Phi$ it holds that

$$
\mathbb{E}_{\phi_{\mathrm{KL}}}[\log q_\phi] - \mathbb{E}_{\pi_1}[\log q_\phi] = \mathbb{E}_{\phi_{\mathrm{KL}}}[\log h] - \mathbb{E}_{\pi_1}[\log h].
\tag{8}
$$

From Eq. (8),

$$
\begin{aligned}
|\mathrm{SKL}(q_{\phi_{\mathrm{KL}}}, \pi_1)| &= \left| \mathbb{E}_{\phi_{\mathrm{KL}}} \left[ \log \frac{\pi_1}{q_{\phi_{\mathrm{KL}}}} \right] - \mathbb{E}_{\pi_1} \left[ \log \frac{\pi_1}{q_{\phi_{\mathrm{KL}}}} \right] \right| \\
&= \left| \mathbb{E}_{\phi_{\mathrm{KL}}} [\log \pi_1] - \mathbb{E}_{\pi_1} [\log \pi_1] + \mathbb{E}_{\pi_1} [\log h] - \mathbb{E}_{\phi_{\mathrm{KL}}} [\log h] \right| \\
&= \left| \mathbb{E}_{\phi_{\mathrm{KL}}} [\log \pi_1 - \log q_{\phi_0}] - \mathbb{E}_{\pi_1} [\log \pi_1 - \log q_{\phi_0}] \right| \\
&\le \mathbb{E}_{\phi_{\mathrm{KL}}} [|\log \pi_1 - \log q_{\phi_0}|] + \mathbb{E}_{\pi_1} [|\log \pi_1 - \log q_{\phi_0}|] \\
&\le \mathbb{E}_{\phi_{\mathrm{KL}}} [g] + \mathbb{E}_{\pi_1} [g] \\
&\le M_1 + M_2.
\end{aligned}
\tag{9}
$$

Therefore, combining Eq. (7) and 9,

$$
\Lambda(q_\phi, \pi_1) \le \sqrt{\frac{1}{2}(M_1 + M_2)}.
$$

### D.2  Example with multivariate normal distributions

We consider here two simple examples to verify that the upper bound given by Theorem 3.5 is small enough for practical purposes.

To put into perspective the numerical values obtained in the examples below, note that the GCBs measured by [36] in 17 problems were in the range 0.4–88.

**Example D.1.** *Suppose that $\pi_1 \sim N(0, \Sigma_1)$ and $q_\phi \sim N(0, \Sigma_0(\phi))$ where*

$$
\Sigma_1 = \begin{bmatrix} 1 & \rho \\ \rho & 1 \end{bmatrix} \qquad \text{and} \qquad \Sigma_0(\phi) = \begin{bmatrix} \phi_1 & 0 \\ 0 & \phi_2 \end{bmatrix}
$$

*for some $0 \le \rho < 1$. By applying Theorem 3.5 we have*

$$
\Lambda(q_{\phi_{KL}}, \pi_1) \le \sqrt{-\frac{1}{2} \log(1 - \rho^2) + \frac{\rho}{1 - \rho}}.
$$

*Substituting $\rho = 0.9, 0.95, 0.99$ provides GCB upper bounds of approximately $3.14, 4.49, 10.05$, respectively, while we obtained values of $\Lambda(q_{\phi_{KL}}, \pi_1)$ of $\approx 0.8$, $1.0$ and $1.5$ using our stabilized moment matching algorithm.*  ◁

*Proof.* Based on moment-matching, $\phi_{\mathrm{KL}} = (1, 1)'$ and therefore $\Sigma_0(\phi_{\mathrm{KL}}) = \mathbb{I}_2$ is the $2 \times 2$ identity matrix.

We have that

$$
\begin{aligned}
\left|\log \pi_1(x) - \log q_{\phi_{\mathrm{KL}}}(x)\right| &= \left| -\frac{1}{2}\log(1-\rho^2) - \frac{1}{2}x^\top \Sigma_1^{-1} x + \frac{1}{2}x^\top \mathbb{I}_2 x \right| \\
&= \left| -\frac{1}{2}\log(1-\rho^2) - \frac{1}{2}x^\top \left(\Sigma_1^{-1} - \mathbb{I}_2\right) x \right| \\
&\leq -\frac{1}{2}\log(1-\rho^2) + \frac{1}{2}\|x\|^2 \max\{|\lambda_{\max}|, |\lambda_{\min}|\} \\
&= -\frac{1}{2}\log(1-\rho^2) + \frac{1}{2}\|x\|^2 \max\left\{\frac{\rho}{1-\rho}, \frac{\rho}{1+\rho}\right\} \\
&= -\frac{1}{2}\log(1-\rho^2) + \frac{1}{2}\|x\|^2 \frac{\rho}{1-\rho} \\
&=: g(x)
\end{aligned}
$$

by the min-max eigenvalue theorem. It can then be verified that

$$
M_1 = M_2 = -\frac{1}{2}\log(1-\rho^2) + \frac{\rho}{1-\rho}.
$$

$\square$

**Example D.2.** *Suppose that the target is a mixture of normal distributions so that*

$$
\pi_1 \sim 0.5 \cdot N(-\mu, 1) + 0.5 \cdot N(\mu, 1) \qquad \text{and} \qquad q_\phi \sim N(0, \phi),
$$

*for some $\mu$. We estimate the expectations in the upper bound of the GCB using Monte Carlo draws from $q_{\phi_{\mathrm{KL}}}$ and $\pi_1$. For $\mu = 5, 10, 100$ we find that $\Lambda(q_{\phi_{\mathrm{KL}}}, \pi_1)$ is upper bounded by approximately 1.7, 3.2, and 32, respectively (up to Monte Carlo estimation error based on 1,000,000 Monte Carlo simulation draws for $\mu = 5, 10$, and 100,000,000 Monte Carlo simulations for $\mu = 100$), while we obtained values of $\Lambda(q_{\phi_{\mathrm{KL}}}, \pi_1)$ of $\approx 2.3$, 2.8 and 4.2 using our stabilized moment matching algorithm.* ◁

*Proof.* Based on moment-matching, we obtain that $\phi_{\mathrm{KL}} = \mu^2 + 1$. Denote the pdf of the standard normal density by $\phi(\cdot)$. Then,

$$
\begin{aligned}
\left|\log \pi_1(x) - \log q_{\mathrm{KL}}(x)\right| &= \left| \log\left(\frac{1}{2}\phi(x+\mu) + \frac{1}{2}\phi(x-\mu)\right) - \frac{1}{\sqrt{\mu^2+1}} \cdot \phi\left(\frac{x}{\sqrt{\mu^2+1}}\right) \right| \\
&=: g(x).
\end{aligned}
$$

$\square$

# E Proof of Theorem 3.6

The proof of this theorem is almost identical to the proof of Theorem 1 in [36]. The main difference is that we study two delayed renewal processes simultaneously instead of one.

First, define the index process for the $j$-th machine for $j = 0, 1, \ldots, \bar{N}$ as $(n_t(j), \epsilon_t(j))$ where $n_t(j) \in \{0, 1, \ldots, \bar{N}\}$, $\epsilon_t(j) \in \{-1, 1\}$, and $\bar{N} = N_\phi + N$. Here, $n_t(j)$ denotes the annealing parameter index for the $j$-th machine at iteration $t$ and $\epsilon_t(j) = 1$ if after iteration $t$ the annealing parameter on machine $j$ will be proposed to increase to index $n_t(j) + 1$. Otherwise, $\epsilon_t^j = -1$. In particular, machine $j$ is storing annealing parameter $\bar{\beta}_{n_t(j)}$, and therefore $n_t(j) = 0, N_\phi, \bar{N}$ means the machine $j$ is at the annealing parameter corresponding to $q_\phi, \pi_1, \pi_0$, respectively.

Informally, a restart occurs when a sample from *either* one of the two references reaches the target distribution chain. Because the target distribution is placed between the two references, we see that we can count the number of restarts by defining two delayed renewal processes and summing the number of restarts for each renewal process. We ensure that we are not double-counting any restarts by introducing two processes instead of one.

Machine $j$ undergoes a restart from $q_\phi$ when $n_t(j)$ goes from 0 to $N_\phi$. Similarly, we will say machine $j$ undergoes a restart from $\pi_0$ when $n_t(j)$ goes from $\bar{N}$ to $N_\phi$. We define $\mathcal{T}_{\phi,t}(j)$ and $\mathcal{T}_t(j)$ to be the total number of restarts on machine $j$ from $q_\phi$ and $\pi_0$ respectively by time iteration $t$. We will denote the total number of restarts by time $t$ from $q_\phi$ and $\pi_0$ as $\mathcal{T}_{\phi,t}$ and $\mathcal{T}_t$ respectively, so that

$$\mathcal{T}_{\phi,t} = \sum_{j=0}^{\bar{N}} \mathcal{T}_{\phi,t}(j), \quad \mathcal{T}_t = \sum_{j=0}^{\bar{N}} \mathcal{T}_t(j)$$

Formally, define

$$T_{\phi,0}^-(j) = \inf\{t : (n_t(j), \epsilon_t(j)) = (0, -1)\}$$

and then recursively define for $k \geq 1$

$$T_{\phi,k}^+(j) = \inf\{t > T_{\phi,k-1}^-(j) : (n_t(j), \epsilon_t(j)) = (N_\phi, +1)\}$$
$$T_{\phi,k}^-(j) = \inf\{n > T_{\phi,k}^+(j) : (n_t(j), \epsilon_t(j)) = (0, -1)\}.$$

We note that $T_{\phi,k}^+(j)$ corresponds to the time of the $k$-th restart from $q_\phi$ on machine $j$ and

$$\mathcal{T}_{\phi,t}(j) = \max\{k : T_{\phi,k}^+(j) \leq t\}.$$

We have $\mathcal{T}_{\phi,t}(j)$ is a delayed renewal process counting the number of times a sample travels from chain 0 (targeting $q_\phi$) to chain $N_1$ (targeting $\pi_1$) with inter-arrival times $T_{\phi,k}(j) = T_{\phi,k}^+(j) - T_{\phi,k-1}^+(j)$

Similarly, define

$$T_0^+(j) = \inf\{t : (n_t(j), \epsilon_t(j)) = (\bar{N}, +1)\}$$

and then recursively define for $k \geq 1$

$$T_k^-(j) = \inf\{t > T_{k-1}^+(j) : (n_t(j), \epsilon_t(j)) = (N_\phi, -1)\}$$
$$T_k^+(j) = \inf\{t > T_k^-(j) : (n_t(j), \epsilon_t(j)) = (\bar{N}, +1)\}.$$

We note that $T_k^-(j)$ corresponds to the time of the $k$-th restart from $\pi_0$ on machine $j$ and

$$\mathcal{T}_t(j) = \max\{k : T_k^-(j) \leq t\}.$$

We have $\mathcal{T}_t(j)$ is a delayed renewal process counting the number of times a sample travels from chain $\bar{N}$ (targeting $\pi_0$) to chain $N_1$ (targeting $\pi_1$) with inter-arrival times $T_k(j) = T_k^-(j) - T_{k-1}^-(j)$.

Although it is possible for a sample to travel from $q_\phi$ to $\pi_1$ and then to $\pi_0$ before returning to $q_\phi$, note that we are not double-counting or missing any restarts by including two renewal processes. More importantly, by introducing two renewal processes (instead of one), we ensure that the times between successive restarts from a given reference on each machine are independent and identically distributed.

In particular, under Assumption A.2, the inter-arrival times $\{T_{\phi,k}(j)\}_k$, and $\{T_k(j)\}_k$ for $\mathcal{T}_{\phi,t}(j)$ and $\mathcal{T}_t(j)$ respectively are i.i.d. for each machine $j$ with distributions $T_\phi$ and $T$ respectively. The round trip rate is thus,

$$
\begin{aligned}
\bar{\tau}_\phi(\bar{\mathcal{B}}_{\phi,\bar{N}}) &= \lim_{t\to\infty} \frac{1}{t}\mathbb{E}[\mathcal{T}_{\phi,t} + \mathcal{T}_t] \\
&= \sum_{j=0}^{\bar{N}} \lim_{t\to\infty} \frac{1}{t}\mathbb{E}[\mathcal{T}_{\phi,t}(j)] + \sum_{j=0}^{\bar{N}} \lim_{t\to\infty} \frac{1}{t}\mathbb{E}[\mathcal{T}_t(j)] \\
&= \frac{\bar{N}+1}{\mathbb{E}[T_\phi]} + \frac{\bar{N}+1}{\mathbb{E}[T]},
\end{aligned}
\tag{10}
$$

where the last equality follows from the renewal theorem. We find an expression for $\mathbb{E}[T_\phi]$ and argue the form for $\mathbb{E}[T]$ by symmetry. As in [36], we omit the $j$ index for the machine number and define $T^{\pi_1}$ and $T^{q_\phi}$ as the first time on a machine where the index reaches $n_t = N_\phi$ and $n_t = 0$ respectively:

$$
\begin{aligned}
T^{\pi_1} &= \min\{t : (n_t, \epsilon_t) = (N_\phi, 1)\} \\
T^{q_\phi} &= \min\{t : (n_t, \epsilon_t) = (0, -1)\}.
\end{aligned}
$$

We define $A_{n,\epsilon}^{\pi_1}$ and $A_{n,\epsilon}^{q_\phi}$ to be the expected time for a machine with index process initialized at $(n_0, \epsilon_0) = (n, \epsilon)$, to reach $\pi_1$ and $q_\phi$ respectively,

$$
\begin{aligned}
A_{n,\epsilon}^{\pi_1} &= \mathbb{E}[T^{\pi_1}|n_0 = n, \epsilon_0 = \epsilon] \\
A_{n,\epsilon}^{q_\phi} &= \mathbb{E}[T^{q_\phi}|n_0 = n, \epsilon_0 = \epsilon].
\end{aligned}
$$

We can decompose the expected inter-arrival time $\mathbb{E}[T_\phi]$ as at the expected time for a machine to travel from $q_\phi$ to $\pi_1$ plus the expected time to travel from $\pi_1$ to $q_\phi$,

$$
\mathbb{E}[T_\phi] = A_{0,-1}^{\pi_1} + A_{N_\phi,+1}^{q_\phi}.
\tag{11}
$$

Also, for notation convenience, we redefine $r_n$, as the probability of a swap is accepted and respectively rejected between machines with annealing parameter $\bar{\beta}_{n-1}$ and $\bar{\beta}_n$.

**Lemma E.1.**

$$
A_{0,-1}^{\pi_1} = N_\phi + 1 + 2 \cdot \sum_{n=1}^{N_\phi} n \cdot \frac{r_n}{1 - r_n}.
\tag{12}
$$

*Proof of Lemma E.1.* Note that by definition, $A_{0,-1}^{\pi_1} = \mathbb{E}[T^{\pi_1}|(n_0, \epsilon_0) = (0, -)]$. Because it is impossible for the index process to reach chains $N_\phi + 1, N_\phi + 2, \ldots, \bar{N}$ before time $T^\tau$, these chains do not enter the calculations for $A_{0,-1}^{\pi_1}$. Therefore (12) is the same expression for "$a_\uparrow^{0,-}$" as in [36] but with $N$ replaced by $N_\phi$. □

**Lemma E.2.**

$$
A_{N_\phi,+1}^{q_\phi} = 2(\bar{N}+1) - (N_\phi + 1) + 2 \cdot \sum_{n=1}^{N_\phi} (\bar{N}+1 - n) \cdot \frac{r_n}{1 - r_n}.
$$

*Proof of Lemma E.2.* It follows from the proof of Theorem 1 in [36] that for $p \in \{\pi_1, q_\phi\}$, and $1 \le n \le \bar{N}$, $A_{n,\epsilon}^p$ satisfies the following recursive relation,

$$
A_{n,+1}^p - A_{n-1,+1}^p = r_n(A_{n,+1}^p - A_{n-1,-1}^p) - 1
\tag{13}
$$
$$
A_{n,-1}^p - A_{n-1,-1}^p = r_{n-1}(A_{n,+1}^p - A_{n-1,-1}^p) + 1.
\tag{14}
$$

If we define $C_n^p$ and $D_n^p$,

$$
C_n^p = A_{n,+1}^p + A_{n-1,-1}^p
\tag{15}
$$
$$
D_n^p = A_{n,+1}^p - A_{n-1,-1}^p.
\tag{16}
$$

By adding and subtracting (13) and (14), we get the joint recursion in $C_n^p$ and $D_n^p$, for $n = 1, \ldots, \bar{N}$:

$$C_{n+1}^p - C_n^p = r_{n+1}D_{n+1}^p + r_n D_n^p, \tag{17}$$

$$(1 - r_n)D_n^p = (1 - r_{n+1})D_{n+1}^p + 2. \tag{18}$$

If the machine's index process is initialized at $(n_0, \epsilon_0) = (0, -1)$, then $A_{0,-1}^{q_\phi} = 0$. We can then substitute in $n = 1$ into (15) and (16) to get,

$$C_1^{q_\phi} = D_1^{q_\phi}. \tag{19}$$

Similarly if the machine's index process is initialized at $(n_0, \epsilon_0) = (\bar{N}, +1)$, then $A_{\bar{N},+1}^{q_\phi} = 1 + A_{\bar{N},-1}^{q_\phi}$. We can substitute this into (14) for $n = \bar{N}$ to get,

$$(1 - r_{\bar{N}})D_{\bar{N}}^{q_\phi} = 2.$$

Using recursion (18), we find for $n = 1, \ldots, \bar{N}$,

$$(1 - r_n)D_n^{q_\phi} = 2(\bar{N} + 1 - n).$$

By adding (15) and (16), that for $1 \leq n \leq \bar{N}$,

$$2A_{n,+1}^p = C_n^p + D_n^p. \tag{20}$$

We can decompose $C_n^{q_\phi}$ as a telescoping sum, and use (17) within initial condition (19) to get the following expression for $2A_{n,+1}^{q_\phi}$ in terms of $D_n^{q_\phi}$

$$
\begin{aligned}
2A_{N_\phi,+1}^{q_\phi} &= C_n^{q_\phi} + D_n^{q_\phi} \\
&= C_1^{q_\phi} + D_n^{q_\phi} + \sum_{n=1}^{N_\phi - 1} (C_{n+1}^{q_\phi} - C_n^{q_\phi}) \\
&= D_1^{q_\phi} + D_{N_\phi}^{q_\phi} + \sum_{n=1}^{N_\phi - 1} (r_{n+1}D_{n+1}^{q_\phi} + r_{N_\phi}D_{N_\phi}^{q_\phi}) \\
&= (1 - r_1)D_1^{q_\phi} + (1 - r_{N_\phi})D_{N_\phi}^{q_\phi} + 2\sum_{n=1}^{N_\phi} r_n D_n^{q_\phi} \\
&= 2\bar{N} + 2(\bar{N} + 1 - N_\phi) + 2\sum_{n=1}^{N_\phi} 2(\bar{N} + 1 - n)\frac{r_n}{1 - r_n}.
\end{aligned}
$$

We arrived at the last line by using (20). Therefore, by dividing by 2 we arrive at our result. $\qquad \square$

By combining Lemma E.1 and Lemma E.2, and using Eq. (11),

$$\mathbb{E}[T_\phi] = A_{0,-1}^{\pi_1} + A_{N_\phi,+1}^{q_\phi}.$$

$$= 2(\bar{N} + 1) + 2(\bar{N} + 1)\sum_{n=1}^{N_\phi} \frac{r_n}{1 - r_n}. \tag{21}$$

By symmetry, we can repeat the same calculation to compute $\mathbb{E}[T]$, the expected restart time for $\pi_0$ to get,

$$\mathbb{E}[T] = 2(\bar{N} + 1) + 2(\bar{N} + 1)\sum_{n=N_\phi+1}^{\bar{N}} \frac{r_n}{1 - r_n}. \tag{22}$$

Therefore, Eq. (21) and Eq. (22) in Eq. (10) imply that

$$
\begin{aligned}
\bar{\tau}_\phi(\bar{\mathcal{B}}_{\phi,\bar{N}}) &= \frac{\bar{N} + 1}{\mathbb{E}[T_\phi]} + \frac{\bar{N} + 1}{\mathbb{E}[T]} \\
&= \frac{1}{2 + 2\sum_{n=1}^{N_\phi} \frac{r_n}{1 - r_n}} + \frac{1}{2 + 2\sum_{n=N_\phi+1}^{\bar{N}} \frac{r_n}{1 - r_n}} \\
&= \tau_\phi(\mathcal{B}_{\phi,N_\phi}) + \tau(\mathcal{B}_N).
\end{aligned}
$$

Finally, it follows from Theorem 3 in [36], that as $\|\mathcal{B}_N\|, \|\mathcal{B}_{\phi,N_\Phi}\| \to 0$ we have

$$\lim_{N_\phi \to \infty} \tau_\phi(\mathcal{B}_{\phi,N_\phi}) = \frac{1}{2 + 2\Lambda(q_\phi, \pi_1)}$$

$$\lim_{N \to \infty} \tau(\mathcal{B}_N) = \frac{1}{2 + 2\Lambda(\pi_0, \pi_1)}.$$

Therefore, as $\|\bar{B}_{\phi,\bar{N}}\| \to 0$ chain-asymptotic restart rate satisfies,

$$\lim_{\bar{N} \to \infty} \bar{\tau}_\phi(\bar{\mathcal{B}}_{\phi,\bar{N}}) = \frac{1}{2 + 2\Lambda(q_\phi, \pi_1)} + \frac{1}{2 + 2\Lambda(\pi_0, \pi_1)}.$$

| | Inference problem | $n$ | $d$ | $\hat{\Lambda}(\pi_0, \pi_1)$ |
|---|---|---|---|---|
| Synthetic | Product | 100,000 | 2 | 3.7 |
| | Simple-Mix (collapsed) | 300 | 5 | 4.3 |
| | Elliptic | 100,000 | 2 | 4.4 |
| | Toy-Mix | NA | 1 | 9.3 |
| Real data | Transfection | 52 | 5 | 6.4 |
| | Titanic | 887 | 9 | 7.7 |
| | Rockets (collapsed) | 5,667 | 2 | 3.4 |
| | Challenger | 23 | 2 | 4.2 |
| | Change-point | 109 | 7 | 3.3 |
| | Vaccines | 77,828 | 12 | 7.8 |
| | Lip Cancer | 536 | 60 | 15 |
| | Pollution | 22,548 | 275 | 56.9 |
| | 8 schools | 8 | 10 | 0.8 |
| | Phylogenetic inference | 249 | 10,395 | 7.0 |
| | Spring failure (improper prior) | 10 | 1 | Not defined |

Table 1: Summary of the models considered in this paper. The sample size $n$, number of model parameters $d$, and fixed reference GCB $\hat{\Lambda}(\pi_0, \pi_1)$ (defined for models with a proper prior distribution). The label "collapsed" is used for the models that are based on [36] but with some latent variables analytically marginalized.

## F   Details of experiments

### F.1   ODE parameters (mRNA transfection data)

The first data set that we consider ("Transfection") is a time series data set based on mRNA transfection data and an ordinary differential equation (ODE) model [5]. The data are observations $O_t$ at times $t = 1, 2, \ldots, 52$ modelled as $O_t | k_{m_0}, \delta, \beta, t_0, \sigma \sim N(\mu, \sigma^2)$ where $k_{m_0}, \delta, \beta, t_0, \sigma$ are parameters and

$$\mu = \frac{k_{m_0}}{\delta - \beta} \left(1 - \exp(-(\delta - \beta) \cdot (t - t_0))\right) \cdot \exp(-\beta \cdot (t - t_0)).$$

The priors placed on the parameters are all log-uniform (i.e., distributed according to $10^U$ where $U$ denotes a random variable with a uniform distribution on $[a, b]$). Specifically, $k_{m_0}, \delta, \beta \sim$ LogUniform$(-5, 5)$, $t_0 \sim$ LogUniform$(-2, 1)$, and $\sigma \sim$ LogUniform$(-2, 2)$, with all parameters a priori independent. We use 50 PT chains, unless stated otherwise. Data is included in the supplement under `bl-vpt/data/m_rna_transfection/processed.csv`.

### F.2   GLM (Titanic data)

Next, we consider a binary generalized linear model ("Titanic") with binary response data, $Y_i$, indicators of survival for the $i$-th Titanic passenger, along with several covariates, $\mathbf{x}_i$. We assume that $Y_i | \beta_0, \beta_1, \mathbf{x}_i \sim$ Bernoulli$(1/(1 + \exp(-(\beta_0 + \beta_1^\top \mathbf{x}_i))))$. Note that this example is similar to the one used by [36] although our prior differs in that we assume that $\beta_0, \beta_{1,1}, \beta_{1,2}, \ldots, \beta_{1,7} \sim$ Cauchy$(\sigma)$ with scale parameter $\sigma$. Further, $\sigma \sim$ Exp$(1)$. All parameters are assumed to be independent in the prior. We use 30 PT chains, unless stated otherwise. Data is included in the supplement under `bl-vpt/data/titanic`.

### F.3   Unidentifiable product

The unidentifiable product model ("Product") is an artificial data set of size $n = 100,000$. The number of failures of an experiment, $n_f$, given the number of trials $n_t$ and parameters $x$ and $y$ is modelled as $n_f | n_t, x, y \sim$ Bin$(n_t, x \cdot y)$. We place priors $X, Y \sim U(0, 1)$ and set $n_t = 100,000$. We observe $n_f = 50,000$ failures. For this model, $d = 2$. Due to identifiability, the posterior concentrates on a thin curve in the square $[0, 1] \times [0, 1]$. We use 15 PT chains, unless stated otherwise.

## F.4  Mixture model

We consider a multi-modal posterior that arises from a normal mixture model ("Simple-Mix"). We model data $X_1, X_2, \ldots, X_m | \mu_1, \mu_2, \sigma_1, \sigma_2, \pi \sim \pi \cdot N(\mu_1, \sigma_1^2) + (1 - \pi) \cdot N(\mu_2, \sigma_2^2)$ with priors $\mu_1, \mu_2 \sim N(150, 100^2)$, $\sigma_1, \sigma_2 \sim U(0, 100)$, and $\pi \sim U(0, 1)$. Due to the label-switching problem, the resulting posterior is multi-modal. In all the experiments we marginalize over the mixture model indicator variables in contrast to [36]. We use 10 PT chains, unless stated otherwise. Data is included in the supplement under `VariationalPT/data/simple-mix.csv`.

## F.5  Bayesian hierarchical model

We also consider a Bayesian hierarchical model based on rocket launch failure data ("Rockets"). The number of rocket launch failures, $f_r$, along with the total number of rocket launches, $l_r$, for $R = 367$ types of rockets is obtained. Given the probability of rocket launch failure for rocket $r$, we model $f_r | \pi_r, l_r \sim \text{Bin}(l_r, \pi_r)$. Given parameters $m$ and $s$, we model $\pi_r | m, s \sim \text{Beta}(ms, (1 - m)s)$ and use the hyper-prior $m \sim U(0, 1)$ and $s \sim \text{Exp}(0.1)$ (rate parameter). In this example we perform inference over each of the $\pi_r$ and $m, s$ so that $d = 369$ and $n = 5,667$. In all the experiments we marginalize over the random effects in contrast to [36]. We use 10 PT chains, unless stated otherwise. Data is included in the supplement under `bl-vpt/data/failure_counts.csv`.

## F.6  Weakly identifiable elliptic curve

A weakly identifiable model is also considered ("Elliptic"). This data set is an artificial data set with the number of failures of an experiment, $n_f$, given the number of trials $n_t$ and parameters $x, y$ modelled as $n_f | n_t, x, y \sim \text{Bin}(n_t, p(x, y))$ where

$$p'(x, y) = y^2 - x^3 + 2x - 0.5$$

$$p(x, y) = \begin{cases} 0 & p'(x, y) < 0 \\ 1 & p'(x, y) > 1 \\ p'(x, y) & \text{otherwise.} \end{cases}$$

We observe $n_t = 100,000$ trials and $n_f = 50,000$ failures. We use the prior $X, Y \sim U(-3, 3)$. We use 30 PT chains, unless stated otherwise.

## F.7  GLM (Challenger data)

Another GLM but applied to Challenger shuttle O-ring data ("Challenger"). The responses consist of binary indicator variables for incidents, $Y_i$, for 23 shuttle launches at temperatures $x_i$. We model the responses as $Y_i | x_i, \beta_0, \beta_1 \sim \text{Bern}(1/(1 + \exp(-(\beta_0 + \beta_1 x_i))))$ and use the prior $\beta_0, \beta_1 \sim N(0, 100)$. We use 15 PT chains, unless stated otherwise. Data is included in the supplement under `bl-vpt/data/challenger`.

## F.8  Additional Blang models

We have implemented the models in Sections F.1–F.7 both procedurally in Julia and declaratively in the Blang probabilistic programming language (PPL) [6] to ensure agreement of the results. We then implemented additional models in Blang, briefly described in this section, to illustrate that the proposed method can be seamlessly incorporated into a PPL. To see more details on these additional models (as well as those described in Sections F.1–F.7), see their PPL source code in the sub-directory `bl-vpt/src/main/java/ptbm/models/` of the supplement.

**Lip Cancer and Pollution:** a sparse Conditional Auto Regressive (CAR) Poisson regression spatial model, constructed as in `https://mc-stan.org/users/documentation/case-studies/mbjoseph-CARStan.html` (full model available in `bl-vpt/src/main/java/ptbm/models/SparseCAR.bl`). For the Lip Cancer problem, we use the dataset documented in the R package `CARBayesdata`, see `bl-vpt/data/scotland_lip_cancer` in the supplement for more information on preprocessing. The Pollution problem is based on the same model but with the bigger dataset documented in `bl-vpt/data/pollution_health`, also obtained from the `CARBayesdata` package on CRAN, which is under a GPL-2 / GPL-3 license. This

provides some examples of how the global communication barrier can be decreased in a realistic data analysis scenario: in the first dataset, from 15 to 6, and in the second, from 57 to 7. For the Pollution dataset, we found the GCB to decrease substantially after considering a larger number of tuning rounds with 4 million PT scans. The GCB with the variational reference during each of the tuning rounds for this model is presented in Fig. 6.

**Vaccines:** the data (`bl-vpt/data/vaccines/data.csv`) consists in the following Phase III COVID-19 vaccines clinical trials: Pfizer-BioNTech, Moderna-NIH, and two AZ-Oxford trials ('South Africa B.1.351' and 'Combined'). The number of cases in each arm of each trial is modelled using a binomial. In each control arm, the probability parameter is set to a trial-specific incidence parameter. In each 'treatment' arm, the probability parameter is set to the same incidence parameter multiplied by one minus a trial-specific efficacy parameter. While the incidence and efficacy parameters are trial specific, the parameters of these distributions are tied across the four trials (full model available in `bl-vpt/src/main/java/ptbm/models/Vaccines.bl`).

**Change-point:** a time series model of discrete counts. We assume one change point uniformly distributed on the time series. The observations at each time point is negative binomial distributed with one set of parameters for the observations before the change point and one set for after the change point (full model available in `bl-vpt/src/main/java/ptbm/models/Mining.bl`). This model is applied to the dataset of annual numbers of accidents due to disasters in British coal mines for years from 1850 to 1962, considered in Carlin et al. (1992) (`bl-vpt/data/mining-disasters.csv`).

**8 schools:** the classical problem from Rubin (1981), see `bl-vpt/src/main/java/ptbm/models/EightSchools.bl` and `bl-vpt/data/eight-schools.csv`.

**Phylogenetics:** The same setup as the *phylogenetic species tree problem* from [36] (`bl-vpt/src/main/java/ptbm/models/PhylogeneticTree.bl`). The data consists in aligned mtDNA sequences (`bl-vpt/data/FES_8.g.fasta`). We only fit variational distributions to the real-valued evolutionary parameters—there is work on constructing variational families for phylogenetic trees [4, 20], but we leave the combination of these families with our inference method for future work.

**Spring failure:** The density estimation problem for spring failure data described in Davison (2003), example 4.2, with a Cauchy likelihood (`bl-vpt/src/main/java/ptbm/models/ImproperCauchy.bl`). We use the Lebesgue improper prior for the Cauchy likelihood's location parameter. This illustrates another use case for our method: when standard PT is applied to a model with an improper prior, one has to select $\beta_0 > 0$ such that $\beta > \beta_0$ guarantees that $\pi_\beta$ is a (normalizable) probability distribution, moreover it is in general not possible to get i.i.d. samples from $\pi_{\beta_0}$. In contrast, our method works with $\beta_0 = 0$ and allows i.i.d. sampling from the variational chain. In the stabilized variant, for the non-adaptive "leg" one can use a fixed variational parameter instead of $\pi_0$. Results for this model are shown in Fig. 7.

**ToyMix:** the density $f(x) = 0.5\phi(x; -r, 0.01) + 0.5\phi(x; r, 0.01)$ where $\phi(\cdot; \mu, \sigma^2)$ is the normal density (`bl-vpt/src/main/java/ptbm/models/ToyMix.bl`). We use $r = 10$.

### F.9 Markovian score climbing

For the `Transfection` model, we implemented the Markovian score climbing (MSC) algorithm of [29]. In the MSC algorithm, there are three main tuning parameters: the number of tuning rounds $K$, the number of samples used in each tuning round $S$, and the step size (or step size sequence) $\epsilon$. The inputs to the algorithm also include the starting values of the variational parameters $\lambda_0$, and the starting sample observation $z_0$. For the mean-field normal variational references, we use the marginal means and log standard deviations as the variational parameters so that all parameters can take on values in $\mathbb{R}$. We initialize each of the log standard deviation parameters to $\log(100.0)$. For the mean parameters, we initialize them to $(1.0, 1.0, 1.0, 0.32, 1.0)$, which are also the initial values provided to the MCMC samplers in the PT algorithms. We use $K = 100,000$ tuning rounds with $S = 100$ and a step size of $\epsilon = 0.0005$. The initial sample provided is the vector of initial mean parameters. The MSC experiments are run 10 times for each selection of simulation settings.

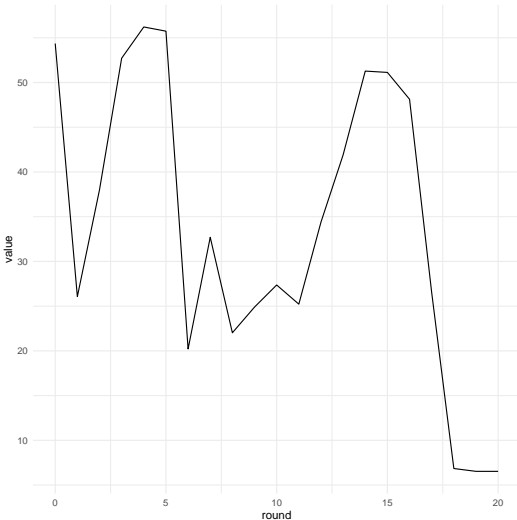

Figure 6: Global communication barrier for the Pollution model in the variational reference chain during each of the tuning rounds. The GCB for the fixed reference chain is estimated to be 57.

Initially, we tried larger values of the step size parameters, starting at $\epsilon = 0.1$. However, we observed that the estimated variational parameters jumped around a large range of values. We therefore decreased the value of $\epsilon$ until there were no convergence issues (i.e., no exceptionally large gradients and the variational parameter estimates seemed to converge based on plots of the parameter estimates against the tuning round number). We also ran the experiment with a smaller step size of $\epsilon = 0.00025$. However, we noticed that the variational parameter estimates displayed a similar behaviour: the parameter estimate would center on only one of the two modes.

### F.10    Additional Details

We use the same number of chains in each leg of variational PT with two reference distributions. For ESS calculations in Julia, we use all samples from the target distribution, including those obtained during annealing schedule and variational reference tuning rounds. For the implementation of variational PT with two references, we employ two separate PT chains: one with a fixed reference and one with a variational reference. During each tuning round, samples from the target distribution in each chain are pooled together to estimate the variational parameters for the reference distribution. In Blang, we implement the topology in which the target distributions between the instances of PT are connected. More information about these different topologies can be found in Appendix F.11.3. In the Julia implementation of the variational PT algorithm, the covariance matrices for the Gaussian reference are estimated using the functions `std()` and `cov()`. The estimates of the variance therefore differ by a factor of $n/(n-1)$ compared to the MLE estimates where $n$ is the sample size for parameter estimation.

All experiments use the same exploration kernel, namely a slice sampling algorithm with "doubling and shrinking" [30, Sections 4.1–4.2]. We performed preliminary experiments with HMC. However, we found that having to tune one HMC sampler for each annealed chain was onerous. We leave the problem of adapting several annealed exploration kernels to future work. Slice sampling in contrast does not have sensitive tuning parameters that require tuning. For the initial state in each chain for the simulations of Section 4.1, the same state is used for all PT initializations for the different seeds. The initial state was chosen to lie in a region with positive density with respect to the reference and target distributions.

Unless mentioned otherwise, all PT algorithms use Deterministic Even Odd (DEO) swaps and the NRPT adaptive schedule algorithm. See [36] for details. Julia simulations were run on an Intel i9-10900K processor with 32 GB of RAM. We also acknowledge use of the ARC Sockeye computing platform (Gold and Silver Xeon 2.1 GHz processors) from the University of British Columbia.

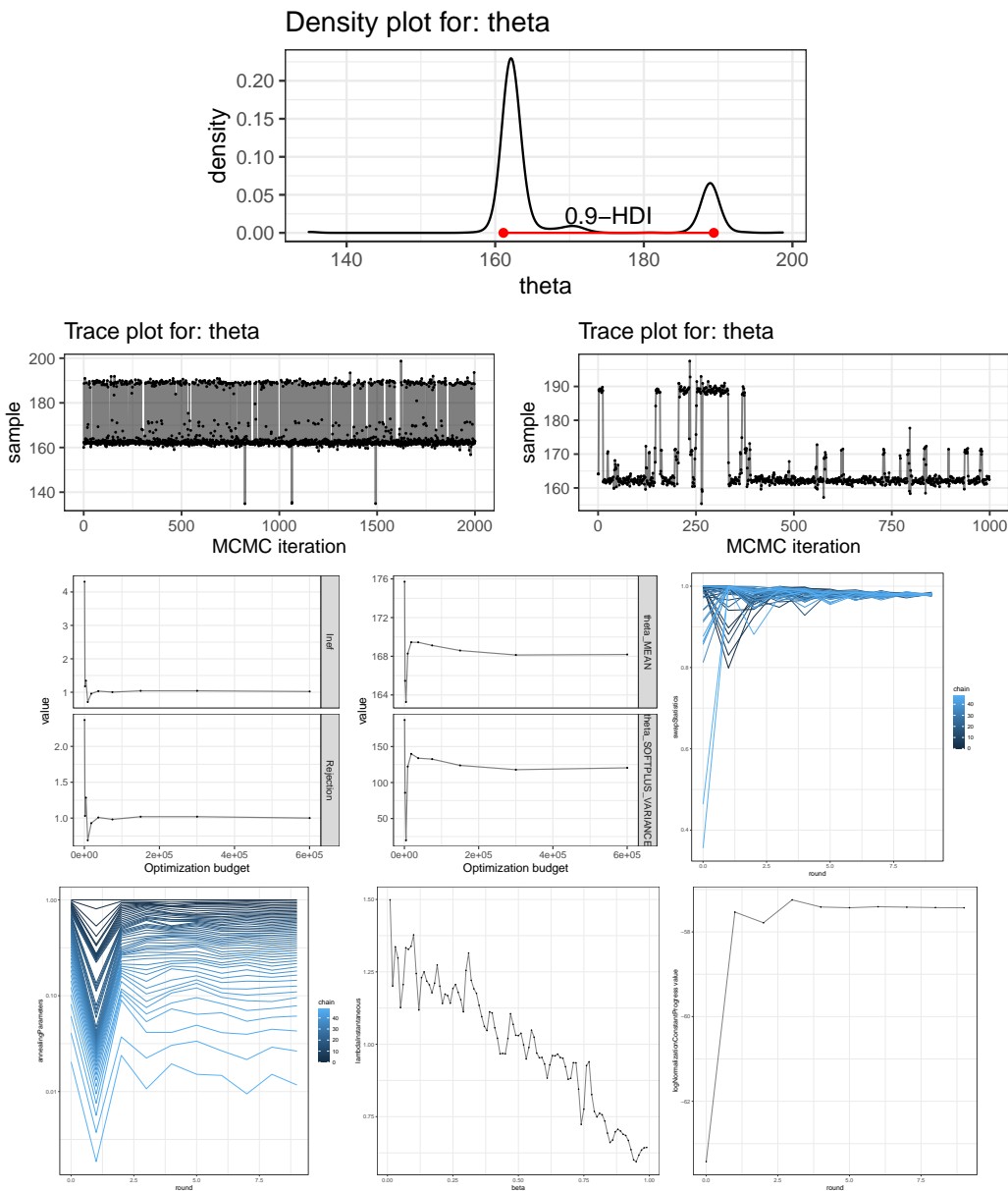

Figure 7: Stabilized variational PT applied to a density estimation problem with a Cauchy likelihood and an improper prior (Section F.8). **First row:** post-burn-in approximation of the posterior distribution over the location parameter $\theta$. Multi-modality is clearly visible. **Second row, left:** trace plot of the same parameter obtained from stabilized variational PT. The stabilized variational PT algorithm is able to frequently switch modes. **Second row, right:** same trace plot but obtained from a single chain MCMC algorithm with the same exploration kernel. **Third row, left:** estimates of two objective functions as a function of the adaptation rounds ("Inef" refers to the sum of the rejection odds, "Rejection," to the sum of rejection rates). **Third row, center:** value of the variational parameters as a function of the adaptation round. The facet "theta_MEAN" refers to the mean of the variational normal distribution approximating the posterior on $\theta$, and "theta_SOFTPLUS_VARIANCE", to the (reparameterized) variance parameter of the same variational distribution. **Third row, right:** average swap acceptance probabilities for the $N = 50$ chains as a function of the adaptation round. **Bottom row, left:** each chain's $\beta_i$ as a function of the adaptation round. **Bottom row, center:** estimated local communication barrier. Even with the relatively large number of chains used in this example, it appears intrinsically less smooth than for the other examples considered in this paper. This may be due to the heavier likelihood tails used in this example. **Bottom row, right:** estimated normalization constant based on the stepping stone method as a function of the adaptation round.

### F.11 Additional experimental results

#### F.11.1 Additional plots

Additional plots accompanying the results of Section 4.1 are presented in Fig. 8 and Fig. 9. Additional plots for the Cauchy example can be found in Fig. 7. For these figures, the local communication barrier (LCB) between $\pi_0$ and $\pi_1$ at $\beta \in [0, 1]$ is

$$\lambda(\pi_0, \pi_1, \beta) = \frac{1}{2}\mathbb{E}[|\ell(X_\beta) - \ell(X'_\beta)|], \qquad \ell(x) = \log \frac{\pi_1(x)}{\pi_0(x)}, \qquad X_\beta, X'_\beta \overset{\text{i.i.d.}}{\sim} \pi_\beta.$$

The LCB can be viewed as an instantaneous communication barrier for each point along the path between $\pi_0$ and $\pi_1$. The integral of the LCB yields the GCB under appropriate moment conditions.

#### F.11.2 Additional results comparing stochastic gradient optimization and moment matching

This section provides additional details for the results described in Section 4.2.

For the stochastic gradient experiments, we considered optimizing several surrogate functions in addition to the global communication barrier (GCB). The first one is the SKL objective, which was used in [37] as a GCB surrogate for optimizing the path between a fixed reference and the target. An estimator of the gradient of the SKL is derived in [37]. We also considered a straightforward variant of this estimator for the forward KL objective. Even when considering a surrogate objective function, we measure performance using the quantity we are ultimately interested in, the GCB, estimated using the sum of the communication rejection rates. Methods optimizing the GCB are labelled 'Rejection' to emphasize that they are technically optimizing the sum of rejections.

We compared the following optimization algorithms: the moment matching method described in Section 3, basic stochastic gradient descent (SGD), where we update variational parameters using the formula $\phi^{(i+1)} \leftarrow \phi^{(i)} - \alpha(i+1)^{-0.6}g^{(i)}$, following [27, Section 4], where $g^{(i)}$ is the stochastic gradient estimate, and Adam [22]. We found both SGD and Adam to be sensitive to the scale $\alpha$ of the updates (where $\alpha$ is defined above for SGD, and a similar parameter appears in Adam, also denoted $\alpha$ in [22]), which we varied in the grid $1/100, 1/10, 1, 10$ (this grid was iteratively increased starting at 1 until the optima laid inside the grid).

We re-parameterized the variance parameters to lie in the real line using a differentiable transformation, which we set to the soft-plus transformation, $\log 1p(\exp(x))$. All methods use the same exploration kernel (Section F.10), which is the expensive inner loop of all optimization methods considered, so we use the number of exploration kernel applications as a model for computational time (abscissa for all plots in this subsection). To set the number of chains, we used the heuristic in [36, Section 5.3], $N^* \approx 2\Lambda$. We estimated $\Lambda \approx 3.5$ and used 8 chains for these experiments. For the stochastic optimization methods, we based the estimator of the gradient on 20 samples each interspersed with one scan, where a scan consists in one application of the local exploration kernel to each variable in each chains, and one set of odd or even swaps.

Simultaneous optimization of the annealing schedule and the variational parameters is straightforward with our moment matching scheme: this is done using Algorithms 2 and 3 in [36] which, like our scheme, proceed in rounds and hence can be performed in combination with round-based variational moment matching. Therefore, for all experiments involving moment matching, we learn an optimal schedule from scratch (i.e., initialized at an equally spaced schedule). For the stochastic optimization methods, simultaneous optimization of annealing schedules is more involved. The GCB is not a suitable objective, as when the number of chains gets larger, the GCB is asymptotically invariant to the schedule (provided the mesh size goes to zero, see [36, Corollary 2]). It would be possible to use the scheme of [27], however this would add more tuning parameters. Instead, for these experiments, we provide all stochastic optimization methods with an optimally tuned schedule and exclude the cost of creating this optimal schedule from the computational budget calculation. As we shall see even with this head start, stochastic optimization does not perform as well as moment matching.

Our implementations of SGD and Adam are augmented with a form of error recovery: when a parameter update leads to a variational parameter where the GCB estimate is not finite (e.g. due to underflow or overflow), we roll back to the previous parameter value. Our implementation stores only one previous valid parameter setting to avoid keeping a history in memory. As a result, some of the stochastic optimization methods move to regions where with high probabilities the proposed next

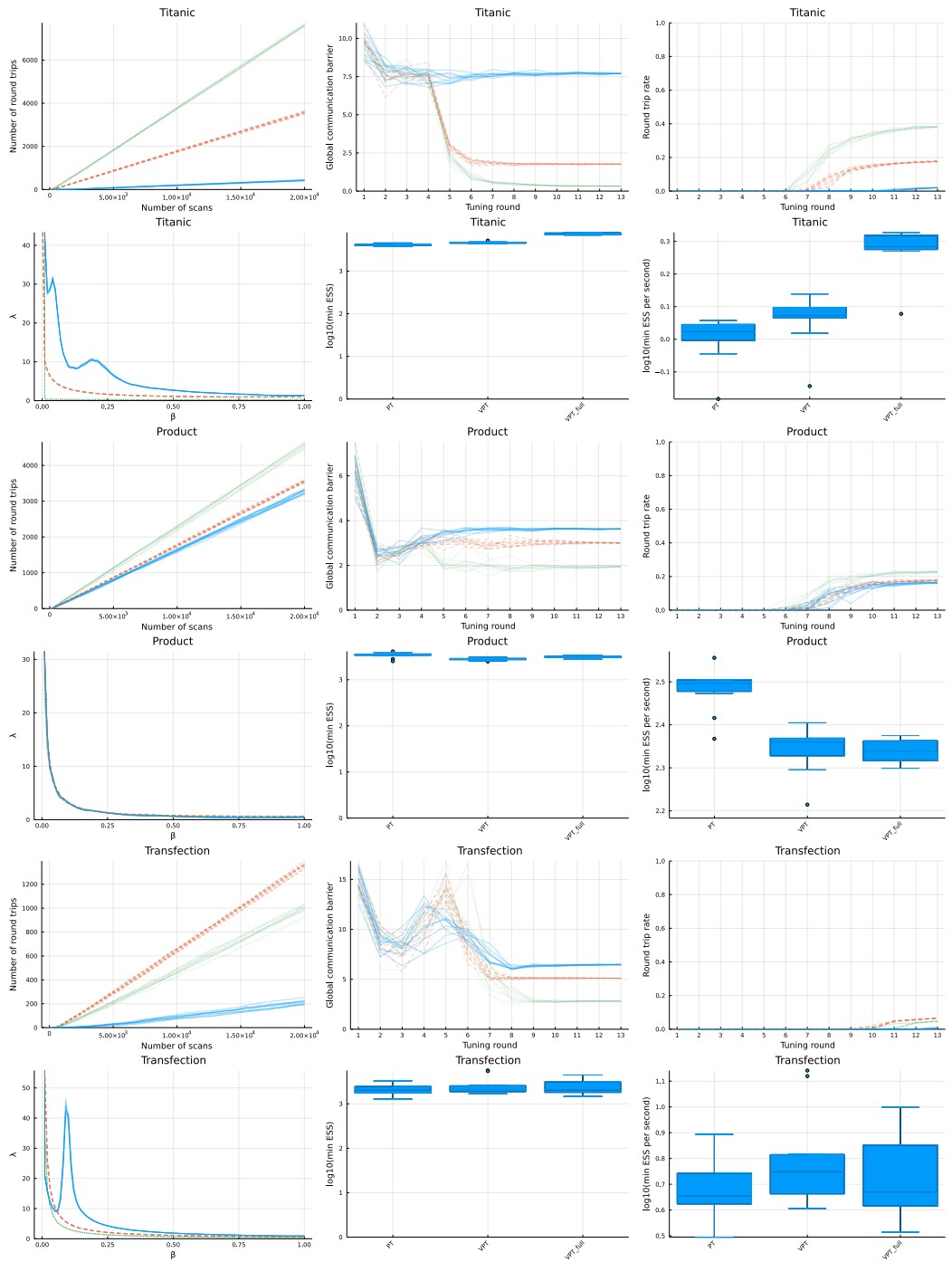

Figure 8: Number of round trips, GCB, round trip rate, local communication barrier (LCB), ESS, and ESS per second. Green/red: Full-covariance/mean-field variational PT. Blue: NRPT.

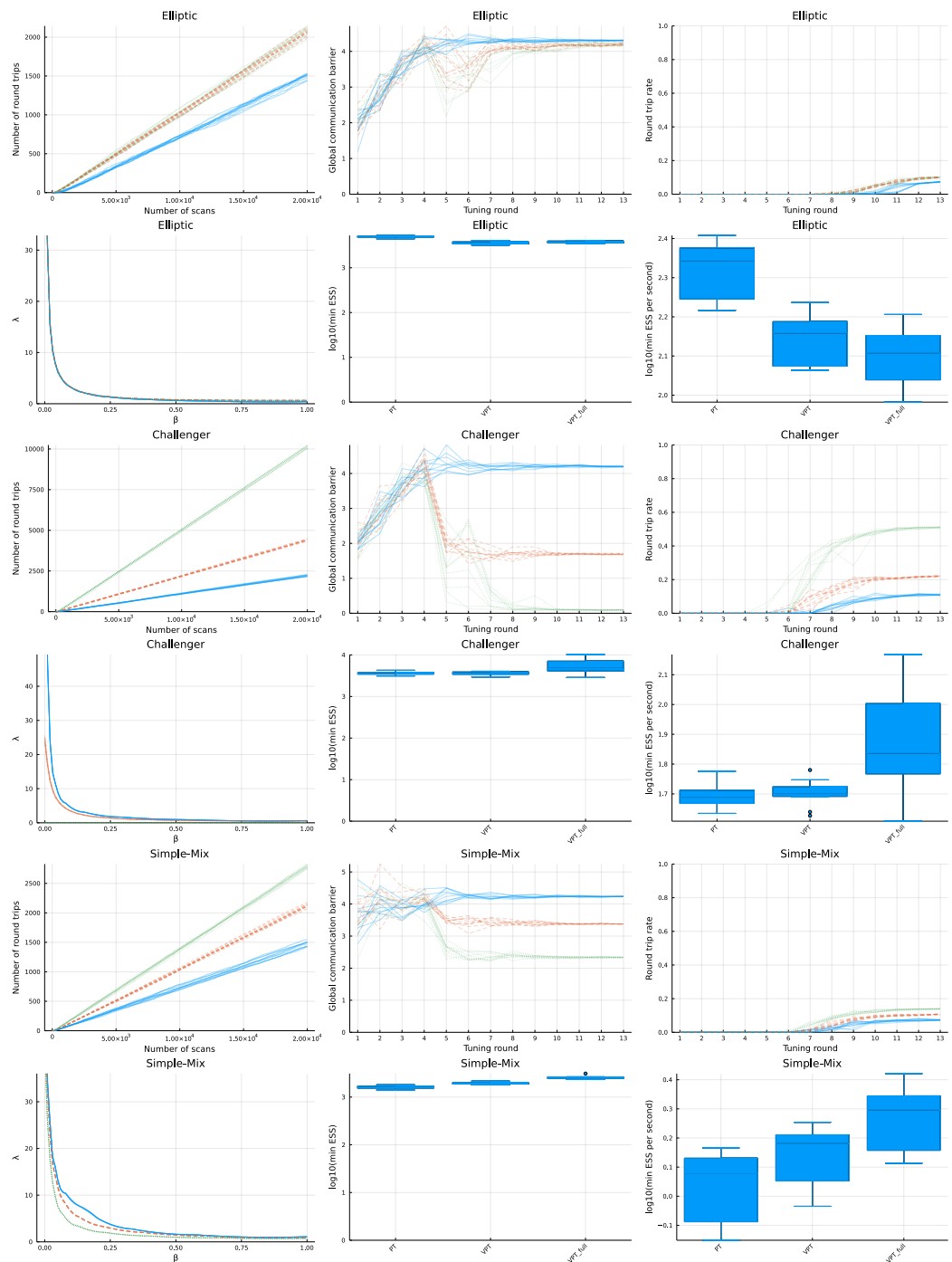

Figure 9: Number of round trips, GCB, round trip rate, LCB, ESS, and ESS per second. Green/red: Full-covariance/mean-field variational PT. Blue: NRPT.

position leads to a rollback—this occurs especially when the step size scale is too large, as shown in Fig. 11. More precisely, this occurs with Adam+FKL for scales larger than 1, with Adam+GCB for all scales considered, with SGD+GCB for scales larger than 0.1, and for all SKL methods for scales larger than 1. We show in Fig. 12 the mean GCB objective, averaged over the finite values (individual traces shown in Fig. 13).

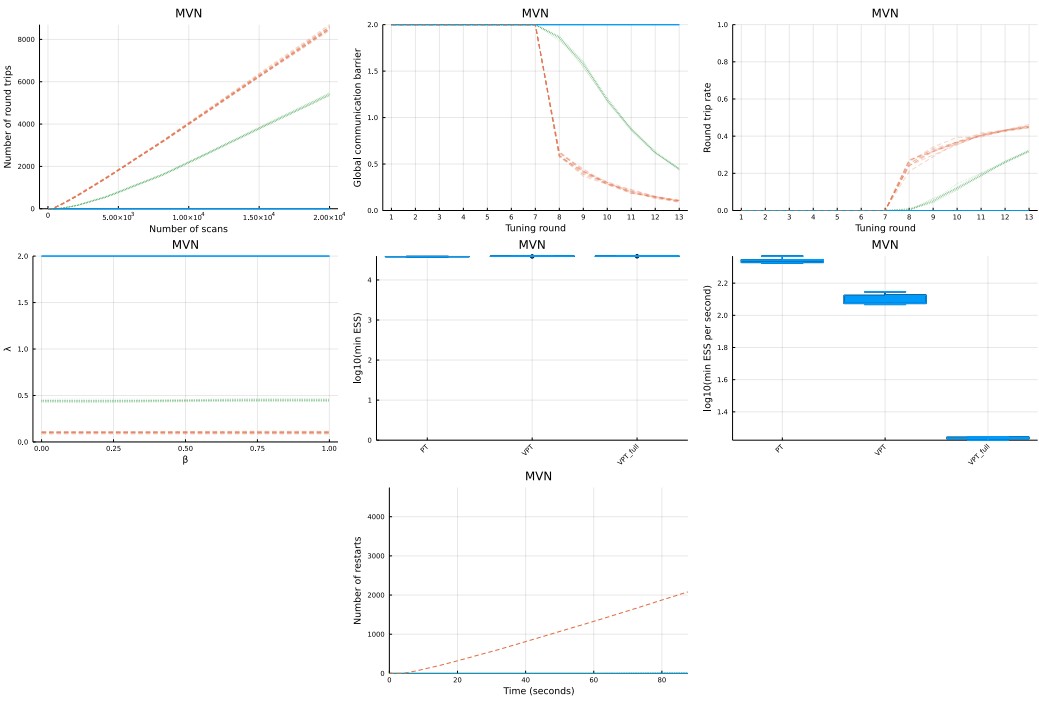

Figure 10: Results for the 100-dimensional multivariate normal target distribution.

Among the configurations that did not lead to non-finite values, the following methods eventually achieved the same optimal GCB loss within the total optimization budget: Adam+FKL+0.01, Adam+FKL+0.1, SGD+SKL+1, Adam+SKL+0.01, Adam+SKL+0.1, SGD+SKL+1. Among those, the fastest to reach that value were Adam+SKL+0.1 and Adam+FKL+0.1, with roughly equivalent performances.

We applied these best performing settings, Adam+SKL+0.1 and Adam+FKL+0.1, to two other problems to see how well they generalize. The two additional problems are: a Bayesian GLM model applied to the titanic dataset described in Section F.2 (with the only difference that we use normal priors here instead of Cauchy priors), and the change point problem described in Section F.8. We use the same experimental conditions as those described above for the Bayesian hierarchical model, with the difference that for the logistic regression problem we used a larger number of chains (20) to take into account the larger GCB ($\approx$8). The results are shown in Section 4.2 and demonstrate that stochastic optimization tuning does not generalize well from one problem to another, while moment matching performs well without requiring tuning.

To replicate the results in this subsection, run, from the root of the repository `bl-vpt-nextflow` at https://github.com/UBC-Stat-ML/bl-vpt-nextflow the command `./nextflow run optimization.nf` and the results will be produced in the directory `bl-vpt-nextflow/deliverables/optimization`.

### F.11.3 Additional results comparing different PT topologies across several models

In this section, we perform an exhaustive empirical comparison of different PT algorithms incorporating a variational distribution. We also include a baseline consisting of PT with a fixed reference. Based on the results of the previous section, we perform optimization using moment matching.

In all experiments in this subsection, we use a normal variational family with diagonal covariance. This ensures that the running time is dominated by the local exploration kernels. We also ensure that the total number of chains is the same for all methods considered. It follows that one scan has comparable running time for all methods considered in this subsection.

To describe the PT algorithms we use the following notation: we use **T** to denote the target distribution; **F**, to denote a fixed reference distribution; **V**, to denote a variational distribution. We use a star

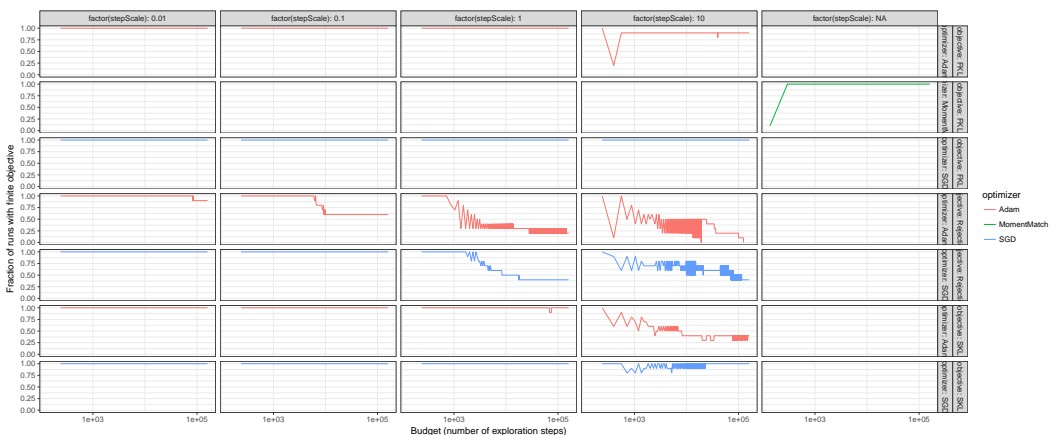

Figure 11: Fraction of the 10 replicates (each provided with a different random seed) that have a finite objective function at each optimization iteration.

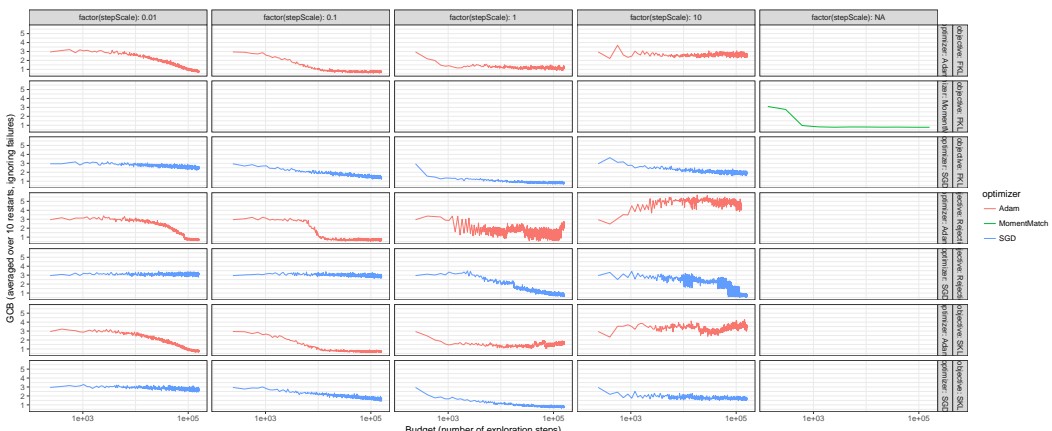

Figure 12: Aggregated performance of all optimization methods and surrogate functions considered in this work.

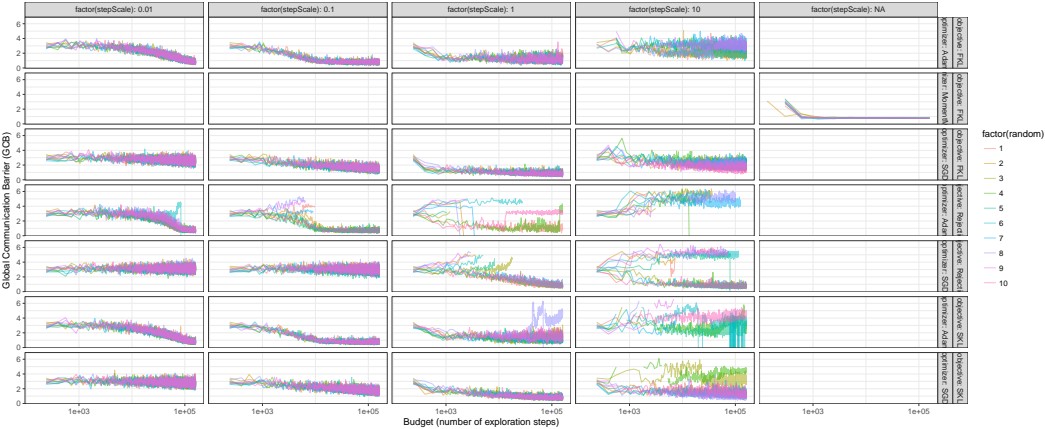

Figure 13: Performance traces of individual optimizers provided with different random seeds (colours).

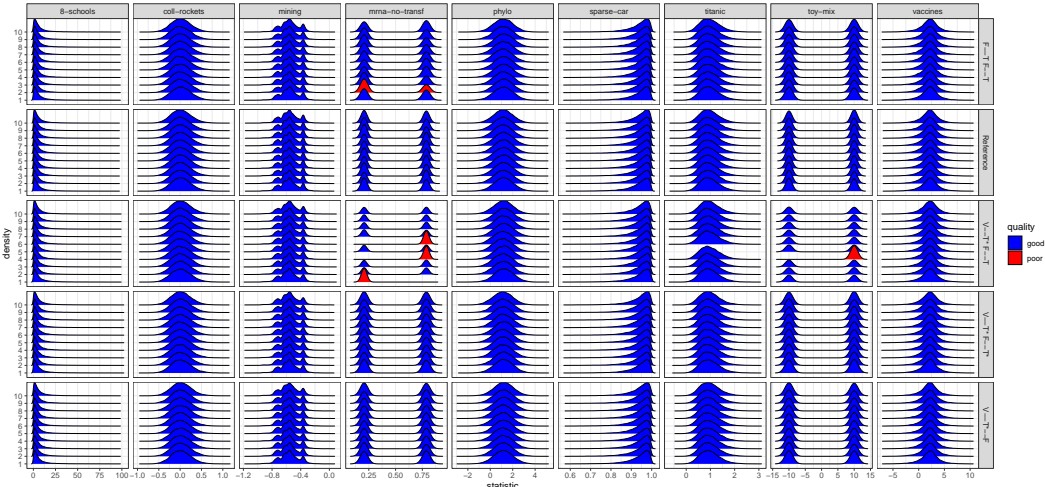

Figure 14: For each model (column), we selected a key statistic or parameter. We show here the approximations of the marginal posterior distributions of these key statistics, for the different algorithms (rows) and 10 random seeds. Those in red have Kolmogorov-Smirnov (KS) distance greater than 0.1 compared to the **"Reference."**

superscript to indicate which chain(s) are used to collect samples for moment matching when updating the variational distribution. Using this notation, the algorithms considered are:

**"F—T   F—T":**  two independent, non-interacting copies of standard PT. We use two copies in order to have a comparable running time per MCMC scan.

**"V—T$^\star$   F—T":**  a basic variational approximation (**V—T$^\star$**), with one independent, non-interacting standard PT, added to have comparable running time per MCMC scan.

**"V—T$^\star$   F—T$^\star$":**  a first type of stabilized variational approximation PT algorithm. As in the algorithm above, it consists of two independent PT algorithms, one using a variational approximation and the other with a fixed reference. In contrast to the one above, the samples from the two PT algorithms are pooled at each round when performing moment-matching.

**"V—T$^\star$—F":**  the same algorithm as above, but where the states of the two target distributions are swapped at every second scan in order to maintain non-reversibility of the index process [36, Section 3.4].

**"Reference":**  the same algorithm as **"F—T   F—T"**, but with ten times the number of MCMC scans as the other methods.

In the main paper, for simplicity, **"F—T   F—T"** is described as "standard PT"; **"V—T$^\star$   F—T"**, as "basic variational PT"; and **"V—T$^\star$—F"**, as "stabilized variational PT." As shown in this section **"V—T$^\star$—F"** and **"V—T$^\star$   F—T$^\star$"** have similar performance in terms of restart rate, however the former is more natural and easier to analyze (Theorem 3.6).

We consider the following models in this section, with the number of chains indicated in parentheses (number of chains selected to be approximately $2\Lambda$ as recommended in [36, Section 5.3]): the eight schools model ($N = 10$), a change point detection model applied on a dataset of mining incidents ($N = 10$), Bayesian estimation of ODE parameters for an mRNA transfection dataset ($N = 30$), Bayesian logistic regression applied to the titanic dataset ($N = 10$), two distinct Bayesian hierarchical models, one for prediction of rocket reliability ($N = 10$) and another tailored to estimating efficacy of multiple COVID-19 vaccines ($N = 20$), one sparse Conditional Auto-Regressive (CAR) spatial model applied to a lip cancer dataset ($N = 20$), one synthetic multi-modal example ($N = 20$), and one phylogenetic inference problem ($N = 20$). Refer to Sections F.1–F.8 for more information.

For each model we first identified one key statistic: either the one of scientific interest or one such that its marginal posterior distribution is multi-modal. We ran each combination of algorithm and model 10 times with different random seeds. The approximations of the marginal posterior distributions are shown in Fig. 14. Visual inspection shows that there are runs that clearly miss the multi-modal

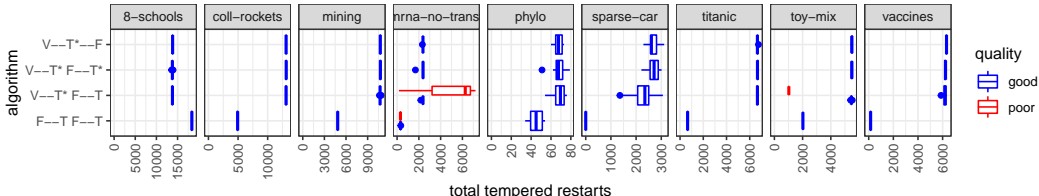

Figure 15: Tempered restarts for different PT algorithms (rows) and models (facets). Each algorithm is executed 10 times with different random seeds. The results are also segregated based on the KS criterion described in Fig. 14.

structure of the problem. To quantify this, we computed the Kolmogorov-Smirnov (KS) distance between the marginal posterior of this statistic obtained from the **"Reference"** run and that obtained from each algorithm. We mark in red the result from any random seed leading to a KS distance greater than 1/10.

Next, we looked at tempered restart count statistics for all combinations of models and algorithms, also splitting the results based on the above mentioned KS threshold. It is important to make this KS-based split, as runs that missed one of the mode may under-estimate the difficulty of the problem and hence over-estimate the number of tempered restarts. The results are shown in Fig. 15 and Table 2.

| model | algorithm | # KS<0.1 | Q1 | Q2 | Q3 |
|---|---|---|---|---|---|
| 8-schools | F–T F–T | 10 | 18310 | 18382 | 18414 |
| 8-schools | V–T* F–T | 10 | 13728 | 13788 | 13809 |
| 8-schools | V–T* F–T* | 10 | 13718 | 13770 | 13797 |
| 8-schools | V–T*–F | 10 | 13797 | 13829 | 13852 |
| coll-rockets | F–T F–T | 10 | 4876 | 4931 | 4953 |
| coll-rockets | V–T* F–T | 10 | 13048 | 13105 | 13160 |
| coll-rockets | V–T* F–T* | 10 | 13074 | 13128 | 13161 |
| coll-rockets | V–T*–F | 10 | 13081 | 13108 | 13151 |
| mining | F–T F–T | 10 | 4824 | 4848 | 4863 |
| mining | V–T* F–T | 10 | 10704 | 10728 | 10745 |
| mining | V–T* F–T* | 10 | 10700 | 10766 | 10823 |
| mining | V–T*–F | 10 | 10716 | 10750 | 10798 |
| mrna-no-transf | F–T F–T | 9 | 311 | 316 | 320 |
| mrna-no-transf | V–T* F–T | 7 | 2366 | 2381 | 2384 |
| mrna-no-transf | V–T* F–T* | 10 | 2361 | 2378 | 2414 |
| mrna-no-transf | V–T*–F | 10 | 2366 | 2376 | 2385 |
| phylo | F–T F–T | 10 | 40 | 45 | 51 |
| phylo | V–T* F–T | 10 | 65 | 70 | 73 |
| phylo | V–T* F–T* | 10 | 65 | 68 | 72 |
| phylo | V–T*–F | 10 | 65 | 68 | 71 |
| sparse-car | F–T F–T | 10 | 0 | 0 | 0 |
| sparse-car | V–T* F–T | 10 | 2063 | 2348 | 2501 |
| sparse-car | V–T* F–T* | 10 | 2556 | 2714 | 2878 |
| sparse-car | V–T*–F | 10 | 2579 | 2607 | 2810 |
| titanic | F–T F–T | 10 | 642 | 650 | 661 |
| titanic | V–T* F–T | 9 | 6615 | 6662 | 6691 |
| titanic | V–T* F–T* | 10 | 6649 | 6666 | 6715 |
| titanic | V–T*–F | 10 | 6644 | 6661 | 6687 |
| toy-mix | F–T F–T | 10 | 2010 | 2020 | 2038 |
| toy-mix | V–T* F–T | 9 | 5443 | 5458 | 5469 |
| toy-mix | V–T* F–T* | 10 | 5446 | 5491 | 5519 |
| toy-mix | V–T*–F | 10 | 5474 | 5484 | 5529 |
| vaccines | F–T F–T | 10 | 147 | 154 | 166 |
| vaccines | V–T* F–T | 10 | 6118 | 6208 | 6264 |
| vaccines | V–T* F–T* | 10 | 6168 | 6236 | 6262 |
| vaccines | V–T*–F | 10 | 6288 | 6316 | 6336 |

Table 2: Quantiles (0.25, 0.5, 0.75) of the number of restarts for the subset of the runs with KS < 0.1.

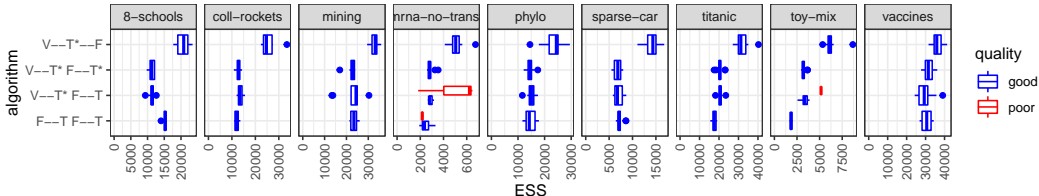

Figure 16: Effective Sample Size (ESS) for each model's key statistic, for different PT algorithms (rows) and models (facets). Each algorithm is executed 10 times with different random seeds. The results are also segregated based on the KS criterion described in Fig. 14.

In most of the models considered, we observed a large increase in the number of tempered restarts when going from standard PT to variational PT. We observed an increase of a factor ∼2.5 in the rocket model, ∼2.2 for the mining problem, ∼1.6 for the phylogenetic problem, ∼10.2 for the titanic problem, ∼40.3 for the vaccine problem (above speed-up estimates computed on the median column of Table 2). The gains are particularly impressive for the sparse CAR model applied to the lip cancer dataset, in which standard PT achieves 0 restarts while the median number of restarts for the variational methods are all higher than 2300 restarts.

For the mRNA and toy mixture problems, only the stabilized algorithms (**"V—T$^\star$—F"** and **"V—T$^\star$ F—T$^\star$"**) succeeded in avoiding catastrophic failures, and compared to standard PT led to an increase in median restarts of a factor ∼7.5 for the mRNA problem and of a factor ∼2.7 for the toy mixture problem. In the mRNA example, 3 out of 10 applications of the basic variational method, **"V—T$^\star$ F—T"**, led to a catastrophic failure, and 1 out of 10 in the toy mixture example. Note that the number of restarts is overestimated in the failed runs, highlighting the importance of using a stabilized algorithm.

The 8 schools problem provides an example where a variational approach based on a diagonal covariance matrix underperforms standard PT. In this example, the number of round trips decreased from a median of 18382 to a median of 13788, 13770, and 13829 for the three variational algorithms considered. But note that the drop in performance is less than 50%, which agrees with the result in Theorem 3.6.

Fig. 16 and Table 3 show performance in terms of the effective sample size (ESS) of each model's key statistic. The ESS is computed using the `mcmcse` R package which implements a batch mean estimator. For the non-interacting variational variants, **"V—T$^\star$ F—T"** and **"V—T$^\star$ F—T$^\star$"**, we observe modest ESS gains compared to standard PT in the rocket, sparse CAR, mRNA and titanic problems, while the gains are larger for the toy mixture problem (∼1.8 increase). For the interacting variational variant, **"V—T$^\star$—F"**, the gains are substantial in all problems considered. However, in contrast to the other algorithms where adding the ESS of the two copies is justified by independence, the ESS estimator may not be reliable in the case of **"V—T$^\star$—F"** given the interactions. The restart rate does not have this limitation so we recommend gauging the performance of the methods primarily based on their restart rate.

We also show the local and global communication barriers, $\lambda$ and $\Lambda$, for these models and algorithms in Fig. 17 and Fig. 18. Interestingly, in some of the cases, e.g. the transfection problem, the high gains in terms of tempered restarts obtained by going from standard PT to variational, are not as large when measured by $\Lambda$. This could be due to the variational $\lambda$ function being generally smoother and hence the optimal schedule easier to approximate in a finite number of rounds.

To replicate the results in this subsection, run, from the root of the repository `bl-vpt-nextflow` at https://github.com/UBC-Stat-ML/bl-vpt-nextflow the command `./nextflow run pt_topologies.nf` and the results will be produced in the directory `bl-vpt-nextflow/deliverables/pt_topologies`.

| model | algorithm | # KS<0.1 | Q1 | Q2 | Q3 |
|---|---|---|---|---|---|
| 8-schools | F–T F–T | 10 | 15011 | 15316 | 15506 |
| 8-schools | V–T* F–T | 10 | 11088 | 11290 | 11698 |
| 8-schools | V–T* F–T* | 10 | 10767 | 11222 | 12074 |
| 8-schools | V–T*–F | 10 | 19047 | 20811 | 22046 |
| coll-rockets | F–T F–T | 10 | 11323 | 12013 | 12700 |
| coll-rockets | V–T* F–T | 10 | 12799 | 13358 | 14301 |
| coll-rockets | V–T* F–T* | 10 | 12315 | 12829 | 13350 |
| coll-rockets | V–T*–F | 10 | 23473 | 24712 | 27128 |
| mining | F–T F–T | 10 | 22004 | 23410 | 24588 |
| mining | V–T* F–T | 10 | 22136 | 24382 | 24928 |
| mining | V–T* F–T* | 10 | 22111 | 22799 | 23663 |
| mining | V–T*–F | 10 | 31719 | 33186 | 33899 |
| mrna-no-transf | F–T F–T | 9 | 2180 | 2360 | 2709 |
| mrna-no-transf | V–T* F–T | 7 | 2755 | 2783 | 3013 |
| mrna-no-transf | V–T* F–T* | 10 | 2688 | 2816 | 2881 |
| mrna-no-transf | V–T*–F | 10 | 4787 | 5053 | 5350 |
| phylo | F–T F–T | 10 | 13061 | 14249 | 16406 |
| phylo | V–T* F–T | 10 | 14256 | 15036 | 15927 |
| phylo | V–T* F–T* | 10 | 13798 | 14476 | 15071 |
| phylo | V–T*–F | 10 | 21723 | 24421 | 25046 |
| sparse-car | F–T F–T | 10 | 6945 | 7071 | 7372 |
| sparse-car | V–T* F–T | 10 | 6234 | 6777 | 7817 |
| sparse-car | V–T* F–T* | 10 | 6122 | 6853 | 7460 |
| sparse-car | V–T*–F | 10 | 13244 | 14351 | 15011 |
| titanic | F–T F–T | 10 | 16942 | 17567 | 18469 |
| titanic | V–T* F–T | 9 | 20082 | 20601 | 21078 |
| titanic | V–T* F–T* | 10 | 19982 | 20406 | 20871 |
| titanic | V–T*–F | 10 | 29952 | 31216 | 33741 |
| toy-mix | F–T F–T | 10 | 1766 | 1843 | 1936 |
| toy-mix | V–T* F–T | 9 | 3186 | 3347 | 3631 |
| toy-mix | V–T* F–T* | 10 | 3093 | 3195 | 3307 |
| toy-mix | V–T*–F | 10 | 5942 | 6129 | 6296 |
| vaccines | F–T F–T | 10 | 28148 | 30611 | 32884 |
| vaccines | V–T* F–T | 10 | 26536 | 29297 | 31160 |
| vaccines | V–T* F–T* | 10 | 29905 | 31443 | 33514 |
| vaccines | V–T*–F | 10 | 34517 | 35911 | 38177 |

Table 3: Quantiles (0.25, 0.5, 0.75) of the ESS for the subset of the runs with KS < 0.1.

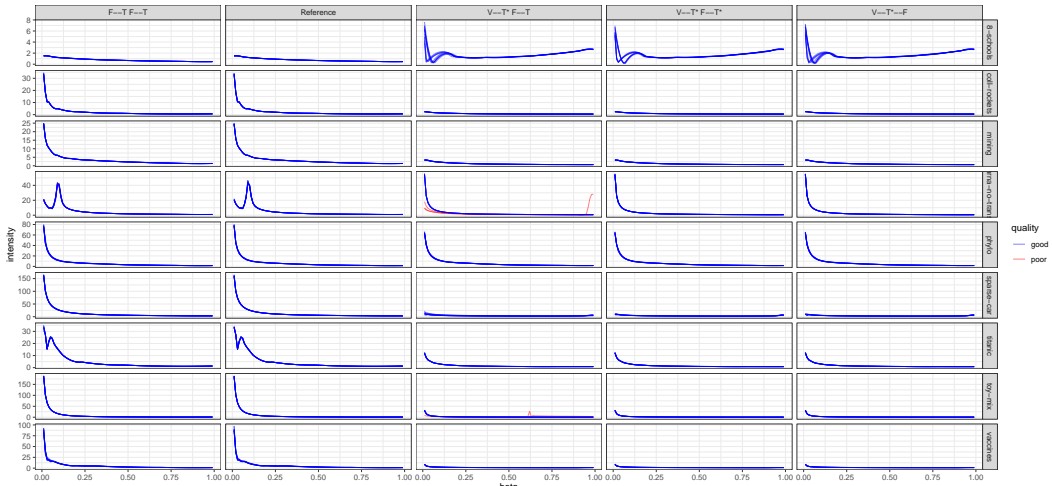

Figure 17: The local communication barrier for nine models (row). For the variational methods, the plot shows the local communication barrier between the variational distribution and the target; for standard PT, the plot shows the local communication barrier between the fixed reference and the target. We show the estimated functions for the five algorithms (columns) and ten random seeds. Red curves indicate "catastrophic failures" as described in Fig. 2.

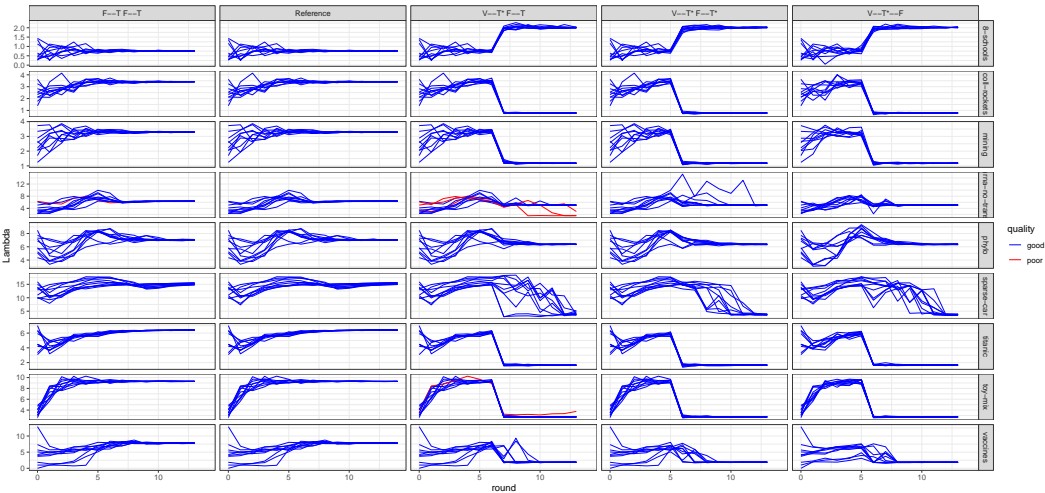

Figure 18: The global communication barrier estimates between the reference and target distributions for nine models (row), each shown as a function of the adaptation round. For the variational methods, the plot shows the global communication barrier between the variational distribution and the target; for standard PT, the global communication barrier is computed between the fixed reference and the target. We show the estimated communication barriers for the five algorithms (columns) and ten random seeds. Red curves indicate "catastrophic failures" as described in Fig. 2.