# OpenReview forum: "Parallel Tempering With a Variational Reference"
_NeurIPS.cc/2022/Conference — NeurIPS 2022 Accept_

### Official Review · Reviewer_hxQQ · 2022-07-08

**Rating:** 5
**Confidence:** 2
**Soundness:** 3 good
**Presentation:** 4 excellent
**Contribution:** 2 fair

**Summary:**

This paper proposes to learn the prior distribution adaptively for parallel tempering. In particular, the prior distribution is tuned to optimize a proxy objective (forward KL divergence to the posterior) with a simple gradient-free moment matching procedure. In theory, the variational prior reference proves to outperform fixed reference, but in practice it may get stuck in a single mode, which the authors resolve by mixing the adaptive and fixed reference distributions. Empirically, the proposed method achieves a big gain over existing methods on Bayesian inference tasks.

**Questions:**

See some of my comments in the weaknesses section.

Other questions:
- The idea of tunning prior is quite natural and similar ideas have been discussed in Bayesian deep learning. So it is quite surprising to me that it has never been used before in parallel tempering. I'm curious about the authors' thoughts on that.
- The authors mentioned that adding the fixed reference could help avoid the local optima and get a better estimate of the variational reference. However, the adaptive reference is still single-mode regardless. I wonder whether this would be an issue for harder problems.
- Parallel tempering seems to be a promising sampling method, but I haven't seen many applications in the context of deep learning. By contrast, many other algorithms (including HMC and AIS) have been used widely in deep learning. I'm curious about the reasons and I wonder if the authors have tried on Bayesian deep learning problems.


**Limitations:**

The authors discussed the limitations in the paper and I don't see any negative societal impact of this work.

**Strengths And Weaknesses:**

Strengths:
- The paper is very well written and easy to follow.
- The introduced algorithm is intuitive and theoretically sound. In the large data limit, the moment-matched reference could achieve the best possible restart rate of 1/2.
- The authors fixed the collapsed reference by adding fixed reference back in practice, which seems to work well empirically. To be fair, I'm not familiar with the datasets the authors used in the paper, so I don't know how convincing the empirical results are.


Weaknesses:
- Lack of discussions about the assumptions in theoretical analyses. For Propositions 3.1-3.3, the conclusions only hold under some assumptions mentioned in the Appendix. Adding some discussions or giving some intuitive explanations about the settings would be helpful for readers to understand the implications of all these propositions.
- All the experiments are done on traditional inference problems with relatively toy models. In this case, I would expect sampling to be "easy". For models like deep neural networks, the posterior could be very complicated and I don't think the combination of a fixed and an adaptive reference would be enough.

---

> ### Author Response · Authors · 2022-08-01
> **Response to Reviewer hxQQ**
>
> Thank you for reviewing our manuscript, and for your encouraging feedback! Below we provide a summary of the comments from the review, as well as a response to each.
>
> > “This paper proposes to learn the prior distribution adaptively for parallel tempering.”
>
> Other authors sometimes use the terms "prior" and "reference" interchangeably, something we avoid in our work; we would like to remind the reviewer that the goal of our paper is to tune the reference distribution, not the prior. Indeed the prior is part of the Bayesian statistical model, which we assume is fixed and given in our work. Most past work sets the reference and prior equal to each other (hence being used interchangeably); our work instead tunes the reference distribution carefully to maximize the restart rate. We show that the tuned reference often provides a much higher restart rate than if one sets the reference to the prior. We will endeavour to make sure this distinction is clear in the camera ready.
>
>
> > “Lack of discussions about the assumptions in theoretical analyses. For Propositions 3.1-3.3, the conclusions only hold under some assumptions mentioned in the Appendix. Adding some discussions or giving some intuitive explanations about the settings would be helpful for readers to understand the implications of all these propositions.”
>
> We completely agree that more discussion of our assumptions could be included in the main text. For the camera-ready, we will be sure to include this. For this response, here is the intuition for Propositions 3.1 and 3.2: we stipulate standard technical conditions sufficient for, e.g., asymptotic consistency of the MLE, a Bernstein-von Mises result for asymptotic normality of the posterior, along with PT-specific assumptions such as efficient local exploration. For Proposition 3.3, we additionally assume that the differences between the posterior mean and MLE, as well as between the inverse Fisher information and scaled posterior variance, are not too large asymptotically.
>
>
> > “All the experiments are done on traditional inference problems with relatively toy models. In this case, I would expect sampling to be "easy".”
>
> We note that of the 15 data sets considered in the paper, 11 are based on real world data. Table 1 in the appendix summarizes all models considered in the paper.
>
> In terms of scalability, we have indeed considered large models, e.g., d = 10,395 parameters in the Phylogenetic model. This is certainly not a toy problem in the context of accurate characterization of posterior distributions due to multi-modality and/or weak identifiability. In this challenging phylogenetic Bayesian inference problem, our variational PT algorithm increased the median effective sample size by 70% compared to standard PT. This example is highlighted in Table 3.
>
> We have also considered large datasets, e.g., n = 77,828 in the Vaccines problem and n = 22,548 in the Phylogenetic problem (in addition to two n=100,000 synthetic datasets). Once again, in the context of accurate characterization of Bayesian posterior distributions, these are not small problems.
>
> Finally, as a related addition, we reran the Pollution example (n = 22,548 and d = 275) with 4 million parallel tempering MCMC scans and found that the global communication barrier decreases from 57 (fixed reference, see Table 1 in the appendix) to 7 (variational reference), which corresponds to a seven-fold increase in the asymptotic restart rate. This example is highlighted in Figure 18 of the updated appendix.
>
> We believe that the extensive simulations on such larger models in the appendix illustrate the utility of our method for difficult Bayesian inference problems.
>
>
> > “The idea of tunning prior is quite natural and similar ideas have been discussed in Bayesian deep learning. So it is quite surprising to me that it has never been used before in parallel tempering.”
>
> We would like to emphasize that we are not tuning the prior in this paper. The prior and likelihood (i.e., the Bayesian statistical model) are fixed and given in our work. We are tuning the reference distribution, which is one end of the “path of distributions” in parallel tempering (the posterior is the other end). In past work on parallel tempering, the reference is often set equal to the prior, mostly for convenience. The main contribution in this work is a method to tune the reference more intelligently. We show that by tuning the reference, we can obtain substantial gains in performance over just setting the reference to the prior.
>
> As the reviewer points out, there is work in the deep learning literature that tunes the prior itself, e.g., “Model Selection for Bayesian Autoencoders” by Tran et al. (2021). This involves changing the statistical model, which we do not do in our work.
>
> We will be sure to clarify this distinction in the camera-ready version.
>
> (Response continued in the next comment.)

---

> > ### Author Response · Authors · 2022-08-01
> > **Response to Reviewer hxQQ (Continued)**
> >
> > (The first part of the response is contained in the previous comment.)
> >
> > > “The authors mentioned that adding the fixed reference could help avoid the local optima and get a better estimate of the variational reference. However, the adaptive reference is still single-mode regardless. I wonder whether this would be an issue for harder problems.”
> >
> > We emphasize that even a unimodal reference is able to recover modes of the target distribution through tempering, as illustrated in Figure 2 and Figure 12.
> >
> > > “Parallel tempering seems to be a promising sampling method, but I haven't seen many applications in the context of deep learning. By contrast, many other algorithms (including HMC and AIS) have been used widely in deep learning. I'm curious about the reasons and I wonder if the authors have tried on Bayesian deep learning problems.”
> >
> > Thanks for raising this point! The reason is that only recently has parallel tempering become practical, due to advances such as improvements in parallel architectures combined with work on non-reversible methods. We believe that this methodology is now a fantastic candidate for application to problems in Bayesian deep learning going forward.
> >
> > We also note that one can use any MCMC algorithm for “local exploration”—including HMC—as  discussed in the background of our paper (Section 2). If HMC is applicable to a given deep learning problem, then we expect our method may improve its performance further. We leave empirical investigation of this to future work.

---

### Official Review · Reviewer_BxuQ · 2022-07-10

**Rating:** 4
**Confidence:** 5
**Soundness:** 3 good
**Presentation:** 2 fair
**Contribution:** 2 fair

**Summary:**

The authors proposed an improved version of the parallel tempering algorithm to solve the non-scalability issue with respect to the data size. In particular, the authors show that in the large-data limit, the restart rate degrades arbitrarily to 0, which strongly affects the communications between the chains associated with the target distribution and the prior distribution. To tackle that issue, the authors proposed to adopt variational inference based on the exponential family. Theories and experiments show much better restart rates.

**Questions:**

Before the exploration issue is elegantly solved, I still have concerns about the usefulness of this method to improve parallel tempering in big data.

**Ethics Review Area:**

["I don’t know"]

**Strengths And Weaknesses:**

Pros:

I like the authors' insight on the weakness of parallel tempering with respect to the data size. Given a fixed schedule of parallel tempering, the communication efficiency does raise a concern in large-data limits. A major reason I am suspecting is that as the number of data points increases, the major mode becomes more dominant, which also inspires the authors to use a tunable prior based on variational inference.

Cons:

1. I think the proposed method is not the right solution to tackle that issue. As is known that parallel tempering not only cares about communication efficiency (or restart rates) but also focuses on the exploration-exploitation trade-off. The current method seems to solve the issue of communication inefficiency, but the impact of exploration is not clear. If we don't know **how much exploration is sacrificed**, why not just adopt a prior that is close enough to the target distribution? In that way, we can maintain a large enough restart rate via the most vanilla method.

2. The combination with a fixed reference further increases my concerns about this method in exploration, which has to resort to a different prior for exploration.

3. Regarding the theories, I feel this paper is more suitable for a journal review.

* I am familiar with Syed's JRSS-B'21 paper but the proof details of this work are not carefully checked.

---

> ### Author Response · Authors · 2022-08-01
> **Response to Reviewer BxuQ**
>
> Thank you for your efforts in reviewing our manuscript! Below we provide a summary of the comments from the review, as well as a response to each.
>
> > “...parallel tempering not only cares about communication efficiency (or restart rates) but also focuses on the exploration-exploitation trade-off. The current method seems to solve the issue of communication inefficiency, but the impact of exploration is not clear.”
>
> We are not entirely certain what this comment means. In parallel tempering, there are only two things that can affect performance of the overall MCMC method: the communication efficiency / restart rate, and the efficiency of the MCMC kernel in each chain (we refer to this kernel as “local exploration”, which is common in the parallel tempering literature, see e.g. Syed et al. 2021). In this paper, we aim to improve communication efficiency. Our method can be used with any local MCMC kernel. We show through our extensive experiments that our method improves communication efficiency substantially.
>
> We would also like to note that our method also improves standard MCMC metrics (that do not rely on using parallel tempering). For example, Table 3 in the appendix shows that our method can improve the effective sample size (ESS) in the target chain substantially. This can also be seen in Figure 14 in the appendix, as well as in past work (e.g., Figure 10 of the JRSS-B paper by Syed et al. 2021). We will clarify this point in the camera-ready version.
>
>
> > “If we don't know how much exploration is sacrificed, why not just adopt a prior that is close enough to the target distribution?”
>
> It is not possible to simply choose a prior that is close to the posterior distribution. In practice, we only have access to the posterior density up to a normalization constant; this is why methods like MCMC or variational inference are necessary.
>
> > “The combination with a fixed reference further increases my concerns about this method in exploration, which has to resort to a different prior for exploration.”
>
> We find that for a vast majority of problems, using only a variational reference alone can perform very well. See Figure 13 for examples. The reason for this improvement in performance is because we optimize the forward KL divergence, which covers the mass of the target distribution.
>
> However, we do find that in some cases, the variational reference by itself can become trapped in local optima. For example, this can happen when the posterior is multimodal and the variational reference becomes trapped in one mode. See Figure 2 and Figure 13 for examples of where this occurs.
>
> To guarantee to a practitioner that our method will never do much worse than the basic choice of setting the reference equal to the prior, we advocate using both references. Indeed, we show in Theorem 3.6 that our method will do at most two times worse (due to additional computation time for using two references) than just using the standard choice of setting the reference equal to the prior. Using two references makes our method more robust in practice. When the variational reference does well, as it usually does, our method has substantial gains. In rare cases where it performs poorly, we fall back to the prior reference.

---

> > ### Comment · Reviewer_BxuQ · 2022-08-01
> > **both exploration and communication matters in PT (further including variational inference is quite ugly)**
> >
> > Q: not possible to simply choose a prior that is close to the posterior distribution
> >
> > It is doable. For example, if we propose to simulate the posterior based on Langevin dynamics with temperature $T_1=1$, then setting e.g. $T_2=T_1 + \text{1e-10}$ for the high-temperature process, then the high-temperature process will asymptotically lead to a  prior close to the posterior (two densities have a large overlap), and communication is much improved, but the exploration is not solved at all!!

---

> > > ### Author Response · Authors · 2022-08-05
> > > **Response to Reviewer BxuQ**
> > >
> > > Thanks for the clarification!
> > >
> > > Please note that in this work, we require i.i.d. draws from the reference. See line 25 where we require a "reference distribution … for which i.i.d. sampling is tractable", and line 138 where we state that "the variational reference family Q should be...simple enough to enable i.i.d sampling...". The ability to take i.i.d. draws guarantees that we can sample the reference effectively, which is part of a standard set of sufficient conditions for PT rapid mixing. See, for example, "Conditions for Rapid Mixing of Parallel and Simulated Tempering on Multimodal Distributions" by Woodard et al. (2009). Your suggestion of a slightly tempered posterior as the reference is possible to implement but provides no guarantee on the reference samples.
> > >
> > > We also reiterate that regardless of the above, our method improves standard MCMC metrics that are agnostic to PT. We emphasize that Table 3 in our appendix shows that our method can improve the effective sample size (ESS) in the target chain substantially. Figure 14 in our paper as well as Figure 10 of the JRSS-B paper by Syed et al. (2021) further reinforce this point.

---

> > > > ### Comment · Reviewer_BxuQ · 2022-08-06
> > > > **The methodology is too ad-hot by including an additional VI**
> > > >
> > > > I maintain my evaluation because I don't think including variational inference is the right path to tackle the challenging issue. The method only seems to works well given that variational inference has found the largest mode (and stays there) to model, which itself is not true. In most cases, your VI may stick at a bad local mode and that makes the algorithm super unstable.

---

> > > > > ### Author Response · Authors · 2022-08-07
> > > > > **Response to Reviewer BxuQ**
> > > > >
> > > > > Thank you for the continuing discussion!
> > > > >
> > > > > > The method only seems to works well given that variational inference has found the largest mode (and stays there) to model, which itself is not true. In most cases, your VI may stick at a bad local mode and that makes the algorithm super unstable.
> > > > >
> > > > > Our paper includes theoretical results to the contrary of both remarks. These results are further supported by empirical evidence in the experiments.
> > > > >
> > > > > In Theorem 3.4 of our work (line 181) we prove that the variational reference converges asymptotically to the forward KL minimizer. Please note that in contrast to the reverse KL minimizer, the forward KL minimizer typically covers all of the modes of the distribution.
> > > > >
> > > > > The method is additionally supported by theory even in the finite-sample setting. Theorem 3.6 (line 240) guarantees that, in the worst-case scenario, our method has a restart rate at least half of that of standard PT.
> > > > >
> > > > > Finally, we do not observe any instability in our experiments. Figure 12 in the appendix shows that our method (V--T*--F) reliably results in an accurate approximation to the posterior, which is a substantial improvement beyond standard PT.

---

> > > > > > ### Comment · Reviewer_BxuQ · 2022-08-07
> > > > > > **more concerns about stability**
> > > > > >
> > > > > > I still have some concerns and may need the authors to clarify: suppose a particle is stuck in a very deep local mode before it explores the whole domain, the reference distribution is optimized mainly based on samples from this local mode, and hence the reference distribution (obtained from moment matching or similar) at this moment actually limits the exploration instead of facilitating exploration.
> > > > > >
> > > > > > Although it seems that in the longtime limit when the reference distribution is fully optimized, it is not an issue. However, it has to spend some exponentially long time to overcome this kind of metastability issue to learn a reasonable well distribution. As such, I don't think this method is promising enough for this challenging task.

---

> > > > > > > ### Author Response · Authors · 2022-08-08
> > > > > > > **Response to Reviewer BxuQ**
> > > > > > >
> > > > > > > Thank you for the feedback!
> > > > > > >
> > > > > > > > Suppose a particle is stuck in a very deep local mode before it explores the whole domain, the reference distribution is optimized mainly based on samples from this local mode, and hence the reference distribution (obtained from moment matching or similar) at this moment actually limits the exploration instead of facilitating exploration.
> > > > > > >
> > > > > > > Please note that the variational reference (when used in its stabilized flavour) does not limit exploration compared to standard PT. If standard PT with a fixed reference is able to detect all modes of the distribution, then so will our method. This is because the variational reference is tuned using samples from the target chain where exploration is assisted by **both** the fixed and variational reference. The fixed reference ensures that there is a source of restarts arriving at the target that are unaffected from the optimization of the variational reference. As long as there are restarts occurring from the fixed reference, it is reasonable to expect that the target chain will be mixing (e.g., see Woodard et al. (2009)). These restarts arrive at a linear rate and thus if the variational reference gets trapped in a deep mode, it will have a chance to escape in a **linear** (and not exponential) number of iterations.
> > > > > > >
> > > > > > >
> > > > > > > However, your intuition about failures is correct! For example, consider a case with severely separated modes: an equally-weighted mixture of N(1, 1) and N(x, 1) distributions where x is very large. If the prior is N(0,1), it is true that the variational reference might remain in the N(1,1) mode for a long time. However, this example is quite pathological. Standard PT, or any other MCMC method, will fail on this problem too. There is no way to reliably discover modes that are arbitrarily far away.
> > > > > > >
> > > > > > > In summary, the choice between prior PT methods and ours is clear. For **any** problem, our method is guaranteed to at least perform similarly to standard PT; and on most real problems, our method provides a substantial boost in performance. There are, of course, pathological examples where our method will take a long time to find the optimal variational reference, but contemporary inference algorithms are not expected to perform well on these.

---

### Official Review · Reviewer_LMG3 · 2022-07-10

**Rating:** 7
**Confidence:** 3
**Soundness:** 3 good
**Presentation:** 3 good
**Contribution:** 3 good

**Summary:**

The work presents a new Parallel Tempering scheme that adapts a variational reference distribution within a parametric family. They adopt a parameter to minimize the forward KL divergence between the parametric family and the target distribution. They combine a fixed and an adaptive reference that leads to better restart rate performance than the baseline.

**Questions:**

My main question is regarding the scalability of the algorithm in deeper and bigger models. It would be interesting to see the method in those applications since the choice of Exponential distribution would make the scheme a good candidate for SG-MCMC like SGLD. But if this is not the scope of this work this is totally understandable and my decision is not conditioned on that.

Although I come from the experimental part of Statistical Machine Learning community I like this work and I think it should be accepted in NeurIPS. This work fits the venue and could provide fruitful discussions between Parallel Tempering researchers.


Some comments regarding the presentation:
-I would put the figures closer to the references cause for the reader is very difficult to go back and forth in the text and try to find Figure and the corresponding text.
-You need a conclusions paragraph.
-Write more clear the advantages of the method in the introduction.
-line 21: gold standard methodology for distribution approximation -> I would avoid using phrases as gold standard
-should have legend on the figures and use the description to describe the plots and what the reader should focus on.


**Limitations:**

This is a theoretical work so negative societal impact is not discussed and the limitations are briefly but not clearly discussed in the main text (Subsection 3.5).

**Strengths And Weaknesses:**

Strengths
-Interesting and witty idea combining a fixed and adaptive reference in the scheme.
-Extensive theoretical analysis of the proposed scheme. The authors provide theoretical guarantees for the performance and convergence of the method.
-Good presentation of the work.

Weaknesses
-A lot of toy experiments but not real world datasets. It would be interesting to see the method applied in a bigger model and a bigger dataset (like an image dataset MNIST, CIFAR10).
-Structure is a bit odd since there are no conclusions and no discussion of limitations, future directions, societal impact. But again this is a theoretical work so societal impact is not applicable in this case. I would like to see the other stuff more in a separate paragraph though.

---

> ### Author Response · Authors · 2022-08-01
> **Response to Reviewer LMG3**
>
> Thank you for your efforts in reviewing our manuscript, and for your encouraging feedback! Below we provide a summary of the comments from the review, as well as a response to each.
>
> > “A lot of toy experiments but not real world datasets. It would be interesting to see the method applied in a bigger model and a bigger dataset…”
>
> > “My main question is regarding the scalability of the algorithm in deeper and bigger models…”
>
> We note that of the 15 data sets considered in the paper, 11 are based on real world data. Table 1 in the appendix summarizes all models considered in the paper.
>
> In terms of scalability, we have indeed considered large models, e.g., d = 10,395 parameters in the Phylogenetic model. This is certainly not a toy problem in the context of accurate characterization of posterior distributions due to multi-modality and/or weak identifiability. In this challenging phylogenetic Bayesian inference problem, our variational PT algorithm increased the median effective sample size by 70% compared to standard PT. This example is highlighted in Table 3.
>
> We have also considered large datasets, e.g., n = 77,828 in the Vaccines problem and n = 22,548 in the Phylogenetic problem (in addition to two n=100,000 synthetic datasets). Once again, in the context of accurate characterization of Bayesian posterior distributions, these are not small problems.
>
> Finally, as a related addition, we reran the Pollution example (n = 22,548 and d = 275) with 4 million parallel tempering MCMC scans and found that the global communication barrier decreases from 57 (fixed reference, see Table 1 in the appendix) to 7 (variational reference), which corresponds to a seven-fold increase in the asymptotic restart rate. This example is highlighted in Figure 18 of the updated appendix.
>
> We believe that the extensive simulations on such larger models in the appendix illustrate the utility of our method for difficult Bayesian inference problems.
>
>
> > “Structure is a bit odd...there are no conclusions and no discussion of limitations, future directions, societal impact.”
>
> > “You need a conclusions paragraph.”
>
> Thank you for pointing this out – we will be certain to include a summary, discussion of limitations, future work, and societal impact in the camera-ready version. For now, we will summarize the main details here in this response.
>
> This paper addressed sampling from a complex target distribution within the parallel tempering framework by constructing a generalized annealing path connecting the posterior to an adaptively tuned variational reference. Experiments in a wide range of realistic Bayesian inference scenarios demonstrate the large empirical gains achieved by our method. In terms of the limitations of our method, in this work we have only considered linear paths from the reference distribution to the target. Future extensions of the work could also consider non-linear paths that allow for more efficient parallel tempering communication (see “Parallel tempering on optimized paths” by Syed et al. 2021). Another inherent limitation of our method—which also occurs in most other standard MCMC methods—is that inference becomes more computationally expensive as the dataset size increases. Fortunately, because any MCMC kernel can be used within each PT chain, recently developed MCMC methods with sub-linear cost in the data size can be readily incorporated into the PT algorithm.
>
> In this work, we provide a new algorithm for Bayesian inference; this is a foundational algorithmic contribution which has little direct societal impact. Although the use of Bayesian models in data analysis itself does indeed have societal implications, we do not want to speculate about potential downstream uses of our inference method with particular models.
>
>
> > “I would put the figures closer to the references cause for the reader is very difficult to go back and forth in the text and try to find Figure and the corresponding text”
>
> We agree that the placement is a bit tricky; unfortunately, due to fairly limited space in the paper, we were forced to make hard decisions regarding arrangement of figures. However, we appreciate the comment and will do our best to clarify the arrangement in the camera ready (it will have an additional page, which should help!).
>
>
> > “Write more clear the advantages of the method in the introduction.”
>
> We agree that the advantages could be made more clear. We will clarify the advantages of our method in the introduction for the camera-ready version. Specifically, we will clarify that even when one is not in the large data setting, our method can provide large empirical gains compared to fixed-reference PT in a wide range of Bayesian inference scenarios. Our methodology is particularly useful in the common case when the target distribution is complex and the prior is far away from the posterior target.
>
> (Response continued in the next comment.)

---

> > ### Author Response · Authors · 2022-08-01
> > **Response to Reviewer LMG3 (Continued)**
> >
> > (The first part of the response is contained in the previous comment.)
> >
> > > “Line 21: gold standard methodology for distribution approximation -> I would avoid using phrases as gold standard”
> >
> > We agree; we will remove this terminology anywhere it appears throughout the manuscript.
> >
> >
> > > “Should have legend on the figures and use the description to describe the plots and what the reader should focus on.”
> >
> > We will add legends where necessary for the camera-ready version. Thank you for pointing this out!

---

### Meta-Review · Area_Chair_SWYB · 2022-08-26

**Recommendation:** Accept
**Confidence:** Less certain

**Metareview:**

The idea of this paper is to tune the reference distribution for parallel tempering to improve efficiency. The key idea is simple: Assume the reference distribution is in the exponential family and use sufficient statistics. Experimental results show that this typically helps in terms of metrics like effective sample size per iteration, though not necessarily in terms of effective samples per second. There are theoretical guarantees which each rely on a long list of assumptions which are deferred to the appendix. While I realize the limitations of space, I echo the reviewers that more discussion of the assumptions should be included in the paper of which should be considered more minor or major. Still this paper proposes a novel approach that is plausibly useful in at least some settings so I recommend acceptance.

A minor point: The font sizes are poorly chosen, to the point of being unreadable if the paper is printed. I had to resort to zooming into individual figures on the computer to reference which was quite tedious.

**Award:**

No

---

### Decision · Program_Chairs · 2022-09-14

Accept